# Physics-informed Neural Networks for Functional Differential Equations: Cylindrical Approximation and Its Convergence Guarantees

**Taiki Miyagawa\***
Independent Researcher
chopin.grande.valse.brillatnte@gmail.com

**Takeru Yokota\***
Interdisciplinary Theoretical and Mathematical Sciences Program (iTHEMS), RIKEN
Center for Quantum Computing, RIKEN
takeru.yokota@riken.jp

## Abstract

We propose the first learning scheme for functional differential equations (FDEs). FDEs play a fundamental role in physics, mathematics, and optimal control. However, the numerical analysis of FDEs has faced challenges due to its unrealistic computational costs and has been a long standing problem over decades. Thus, numerical approximations of FDEs have been developed, but they often oversimplify the solutions. To tackle these two issues, we propose a hybrid approach combining physics-informed neural networks (PINNs) with the *cylindrical approximation*. The cylindrical approximation expands functions and functional derivatives with an orthonormal basis and transforms FDEs into high-dimensional PDEs. To validate the reliability of the cylindrical approximation for FDE applications, we prove the convergence theorems of approximated functional derivatives and solutions. Then, the derived high-dimensional PDEs are numerically solved with PINNs. Through the capabilities of PINNs, our approach can handle a broader class of functional derivatives more efficiently than conventional discretization-based methods, improving the scalability of the cylindrical approximation. As a proof of concept, we conduct experiments on two FDEs and demonstrate that our model can successfully achieve typical $L^1$ relative error orders of PINNs $\sim 10^{-3}$. Overall, our work provides a strong backbone for physicists, mathematicians, and machine learning experts to analyze previously challenging FDEs, thereby democratizing their numerical analysis, which has received limited attention. Code is available at https://github.com/TaikiMiyagawa/FunctionalPINN.

## 1 Introduction

Functional differential equations (FDEs) appear in a wide variety of research areas [91, 92, 79]. FDEs are partial differential equations (PDEs) involving functional derivatives, where a functional $F$ is a function of an input function $\theta(x)$ to a real number, i.e., $F : \theta \mapsto F([\theta]) \in \mathbb{R}$, and a functional derivative is defined as the derivative of functional w.r.t. the input function at $x$, denoted by $\delta F([\theta])/\delta\theta(x)$. FDEs play a fundamental role in Fokker-Planck systems [27], turbulence theory [67], quantum field theory [71], mean-field games [16], mean-field optimal control [18, 81], and unnormalized optimal transport [32]. Major examples of FDEs include the Hopf functional equation

---

*These authors contributed equally to this work.

38th Conference on Neural Information Processing Systems (NeurIPS 2024).

in fluid mechanics, the Fokker-Planck functional equation in the theory of stochastic processes, and the functional Hamilton-Jacobi equation in optimal control problems in density spaces.

Despite their wide applicability, numerical analyses of FDEs are known to suffer from significant computational complexity; therefore, numerical approximation methods have been developed over decades. They include the functional power series expansion [67], the Reynolds number expansion [67], finite difference methods [18], finite element methods [79], tensor decomposition methods [91, 79], and the cylindrical approximation [7, 29, 34, 88].

However, they tend to oversimplify the solution of FDEs, prioritizing the reduction of computational costs. The functional power series expansion is applicable only to input functions close to the expansion center. Moreover, it has no convergence guarantees in general [67]. The Reynolds number expansion requires the Reynolds number to be close to zero, severely restricting its applicability, because the Reynolds number for turbulent flow can be $\gtrsim 1000$. Discretization-based methods such as the finite difference and element methods restrict spacetime resolution and/or the class of input functions and functional derivatives [79]. Existing methods relying on the cylindrical approximation, akin to the spectral method for PDEs, include tensor decomposition [91, 92] to reduce computational costs; however, they tend to significantly simplify the class of input functions and functional derivatives. For instance, their expressivity is limited to polynomials or Fourier series of a few degrees.

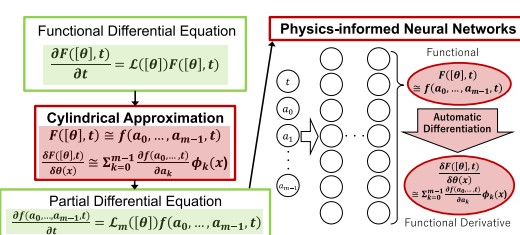

Figure 1: **Overall architecture.** An FDE is simplified to a high-dimensional PDE via the cylindrical approximation. The PDE is solved with a PINN. The approximated functional derivative can be efficiently computed with automatic differentiation.

To address the notorious computational complexity and limited approximation ability, we propose a hybrid approach combining physics-informed neural networks (PINNs) and the cylindrical approximation (Fig. 1). In the first stage, we expand the input function with orthonormal basis functions, thereby transforming a given FDE into a high-dimensional PDE of the expansion coefficients. This approximation is referred to as the *cylindrical approximation*. We prove the convergence of the approximated functional derivatives and FDE solutions, validating the reliability of the cylindrical approximation for FDE applications, which is our main theoretical contribution. In the second stage, the derived high-dimensional PDE is numerically solved with a PINN, which is known to be a universal approximator tailored to solve high-dimensional PDEs efficiently in a mesh-free manner.

A notable advantage of our approach is that, with the help of PINNs, it reduces the computational complexity by orders of magnitude, compared with previous discretization-based methods. In fact, it requires only $\mathcal{O}(m^r)$, where $m$ represents the "class size" of input functions and functional derivatives (e.g., the degree of polynomials), and $r$ ($\geq 1$) denotes the order of the functional derivative included in the target FDE (typically 1 or 2). This is a notable reduction from the state-of-the-art cylindrical approximation algorithm [91], which requires as large as $\mathcal{O}(m^6)$. Consequently, our approach substantially extends the class of input functions and functional derivatives that can be represented by the cylindrical approximation. For instance, our approach extends the degrees of polynomials or Fourier series used for the approximation from 6 [91] to 1000, showing unprecedented expressivity.

As a proof of concept, we conduct experiments on two FDEs: the functional transport equation and the Burgers-Hopf equation. The results show that our model accurately approximates not only the solutions but also their functional derivatives, successfully achieving typical $L^1$ relative error orders of PINNs $\sim 10^{-3}$ [19, 12, 20, 21, 47, 77, 84, 93, 94].

Our contribution is threefold. (1) We propose the first learning scheme for FDEs to address the significant computational complexity and limited approximation ability. Our model exponentially extends the class of input functions and functional derivatives that can be handled accurately and efficiently. (2) We prove the convergence of approximated functional derivatives and FDE solutions, ensuring the cylindrical approximation to be safely applied to FDEs. (3) Our experimental results show that our model accurately approximates not only the FDE solutions but also their functional derivatives.

## 2 Related Work

FDEs are prevalent across numerous research fields [27, 24, 32, 16, 18, 81, 67, 71]. Research on FDEs has mainly focused on their theoretical aspects and formal solutions, with very few algorithms available to numerically solve general FDEs [18, 79, 91, 104]. In [18], a numerical method specialized for the Hamilton-Jacobi functional equation for optimal control problems in density space is proposed, based on spacetime discretization. Similarly, [79] employs spacetime discretization with tensor decomposition. The state-of-the-art algorithm proposed in [91], the CP-ALS (Canonical Polyadic tensor expansion & Alternating Least Squares) algorithm, uses the cylindrical approximation along with the finite difference method and tensor decomposition, requiring $\mathcal{O}(m^6)$ (see App. F.2 for the derivation), whereas our model requires only $\mathcal{O}(m^r)$ ($r$ is 1 or 2 in most FDEs). Furthermore, our model does not require such discretization, making it mesh-free. See App. B for an additional introduction to FDEs and their approximations.

The cylindrical approximation originates from the theory of stochastic processes [34, 88]. It is reminiscent of the spectral method for PDEs [66] and is a generalization to FDEs. Convergence theorems of the cylindrical approximation are summarized in a recent seminal paper [92]. Note that the cylindrical approximation in this paper (Eq. (2)) is different from the one in [92], tailored for practical use. Consequently, our convergence theorems also differ from those in [92]. See App. C.4.2 for technical details. See App. A for more comparisons with other studies.

## 3 Proposed Approach

### 3.1 Step 1: Cylindrical Approximation

We first introduce the *cylindrical approximation of functionals, functional derivatives, and FDEs*, beginning with the expansion of input functions and culminating in the transformation of FDEs into high-dimensional PDEs. Additionally, we prove the convergence theorems for this approximation. The rigorous mathematical background is reviewed in App. C for interested readers.

Firstly, we define the *cylindrical approximation of functionals* [7, 29, 34, 88]. Any function $\theta$ in a real separable Hilbert space $H$ can be represented uniquely in terms of an orthonormal basis $\{\phi_k\}_{k=0}^{\infty}$ as $\theta(x) = \sum_{k=0}^{\infty} a_k \phi_k(x)$, where $a_k := (\theta, \phi_k)_H$ are the *coefficients* (or spectrum) of $\theta$ in terms of $\{\phi_k\}_{k\geq 0}$, and $(\cdot, \cdot)_H$ denotes the inner product of $H$. Substituting this expansion to functional $F([\theta])$, we can define a multivariable function $f(\{a_k\}_{k=0}^{\infty}) := F([\sum_{k=0}^{\infty} a_k \phi_k])$ for any functional $F : H \to \mathbb{R}$. Truncating $k$ at $m - 1 \in \mathbb{Z}_{\geq 0}$ gives the cylindrical approximation of functionals:

$$f(\{a_k\}_{k=0}^{m-1}) := F([P_m \theta]), \tag{1}$$

where $P_m$ is the projection operator s.t. $P_m \theta(x) := \sum_{k=0}^{m-1} a_k \phi_k(x)$, and $m$ is referred to as the *degree* of approximation. See Thm. C.19 and Thm. C.20 for the uniform convergence and convergence rate of this approximation, originally given by [75, 92].

Secondly, we define the *cylindrical approximation of functional derivatives*. The functional derivative of $F$ w.r.t. $\theta$ at $x$ is defined as $\frac{\delta F([\theta])}{\delta \theta(x)} := \lim_{\epsilon \to 0} \frac{F([\theta(y) + \epsilon \delta(x-y)]) - F([\theta(y)])}{\epsilon}$, where $\delta(x)$ denotes the Dirac delta function. This definition is impractical to simulate on computers with spacetime discretization; thus, we employ the expansion $\frac{\delta F([\theta])}{\delta \theta(x)} = \sum_{k=0}^{\infty} (\frac{\delta F([\theta])}{\delta \theta}, \phi_k)_H \phi_k(x)$. The expansion is possible because $\frac{\delta F([\theta])}{\delta \theta(x)}$ itself is a function of $x$ in $H$ and thus can be represented as an orthonormal basis expansion. Note that the expansion coefficients $(\frac{\delta F([\theta])}{\delta \theta}, \phi_k)_H$ are known to be equal to $\frac{\partial f}{\partial a_k}$ (see App. C.4.2 for the proof). Hence, truncating $\theta$ at $m - 1$ gives the cylindrical approximation of functional derivatives:

$$P_m \frac{\delta F([P_m \theta])}{\delta \theta(x)} = \sum_{k=0}^{m-1} \frac{\partial f}{\partial a_k} \phi_k(x). \tag{2}$$

Note that Eq. (2) is different from the cylindrical approximation adopted in [91, 92]. They do not apply $P_m$ to $\delta F([P_m \theta])/\delta \theta(x)$, and the emerging "tail term" $\Sigma_{k=m}^{\infty}(\delta F([\theta])/\delta \theta, \phi_k)\phi_k(x)$ is simply ignored without any rationale.

The first main theoretical contribution of our work is the following convergence theorem of Eq. (2).

**Theorem 3.1** (Pointwise convergence of approximated functional derivatives (informal)). *For arbitrary $\theta \in H$ and orthonormal basis $\{\phi_0, \phi_1, \ldots\}$, Eq. (2) converges to $\frac{\delta F([\theta])}{\delta\theta}$ as $m \to \infty$.*

The formal statement and proof are given in App. D.1. The convergence rate is the same as $\|\theta - P_m\theta\|$ if $\delta F([\theta])/\delta\theta(x) \in \text{span}\{\phi_0, \ldots, \phi_{m-1}\}$. A technical discussion when this is not the case is provided in App. D.

Finally, we define the *cylindrical approximation of FDEs* [92]. In this paper, we consider the *abstract evolution equation*, a class of FDEs having the following form:

$$\partial F([\theta], t)/\partial t = \mathcal{L}([\theta])F([\theta], t) \tag{3}$$

with $F([\theta], 0) = F_0([\theta])$, where $\mathcal{L}([\theta])$ is a linear functional operator, and $F_0$ is a given initial condition. The abstract evolution equation is a crucial class of FDEs in physics, mathematics, and engineering, including the Hopf functional equation, Fokker-Planck functional equation, and functional Hamilton-Jacobi equation. The cylindrical approximation of the abstract evolution equation is given by

$$\partial f(\boldsymbol{a}, t)/\partial t = \mathcal{L}_m([\theta])f(\boldsymbol{a}, t) \tag{4}$$

with $f(\boldsymbol{a}, 0) = f_0(\boldsymbol{a})$, where $\boldsymbol{a} := (a_0, \ldots, a_{m-1})^\top$, $f(\boldsymbol{a}, t) := F([P_m\theta], t)$, and $f_0(\boldsymbol{a}) := F_0([P_m\theta])$. The operator $\mathcal{L}_m([\theta])$ is the cylindrical approximation of $\mathcal{L}([\theta])$. Examples are given in Secs. 3.1.1 & 3.1.2.

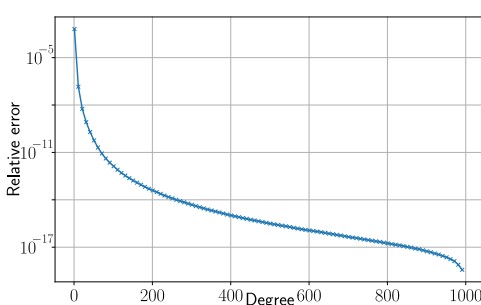

Figure 2: **Cylindrical approximation of FDE's solution.** The $L^1$ relative error, defined as $\frac{1}{N}\sum_{i=1}^{N}|F([\theta_i]) - F([P_m\theta_i])|/|F([\theta_i])|$, diminishes with increasing $m$. The Burgers-Hopf equation with the delta initial condition (Sec. 3.1.2) is considered. Note that PINN's training is not included.

The second main theoretical contribution of our work is the following convergence theorem of solutions:

**Theorem 3.2** (Convergence of approximated solutions (informal)). *Under the cylindrical approximation (Eq. (2)), if the FDE depends on functional derivatives only in the form of the inner product $(v, \frac{\delta F([\theta])}{\delta\theta})_H$ ($v \in H$), then, the solution of the approximated abstract evolution equation $(F([P_m\theta], t))$ converges to the solution of the original one $(F([\theta], t))$ as $m \to \infty$.*

The proof is given in App. E. The convergence is visualized in Fig. 2. Similar theorems for the FDEs with the second or higher-order functional derivatives can be shown in a similar way. The inner-product assumption in Thm. 3.2 is satisfied by major FDEs, such as the Hopf functional equation, Fokker-Planck functional equation, and functional Hamilton-Jacobi equation.

In the following, we apply the cylindrical approximation to two FDEs: the functional transport equation (FTE) and the Burgers-Hopf equation (BHE).

### 3.1.1 Application 1: Functional Transport Equation

We first construct a simple FDE, the *functional transport equation* (FTE), which is a generalization of the transport equation (the continuity equation) [53]. The FTE is provided by

$$\frac{\partial}{\partial t}F([\theta], t) = -\int dx\, u(x)\frac{\delta F([\theta], t)}{\delta\theta(x)}, \tag{5}$$

with the initial condition $F([\theta], 0) = F_0([\theta])$, where $x \in [-1, 1]$, and $u(x)$ is a given function. Specifically, we use the initial condition $F([\theta], 0) = \rho_0\int dx\, u(x)\theta(x)$ with $\rho_0$ a constant. The exact solution is $F([\theta], t) = F_0([\theta - ut]) = \rho_0\int dx\, u(x)(\theta(x) - u(x)t)$, as can be seen by substituting this into Eq. (5). More details and motivations of the FTE are provided in App. E.1.

The cylindrical approximation of the FTE is given by

$$\frac{\partial}{\partial t}f(\boldsymbol{a}, t) = -\sum_{k=0}^{m-1} u_k\frac{\partial}{\partial a_k}f(\boldsymbol{a}, t) \tag{6}$$

with the initial condition $f(\boldsymbol{a}, 0) = f_0(\boldsymbol{a})$, where $\theta(x) = \sum_{k=0}^{m-1} a_k \phi_k(x)$, and $f_0(\boldsymbol{a}) = F_0([P_m \theta]) = \rho_0 \Sigma_{k=0}^{m-1} u_k a_k$ with $u_k := (u, \phi_k)_{L^2([-1,1])} = \int dx u(x) \phi_k(x)$. We use the Legendre polynomials as the orthonormal basis $\{\phi_k\}_{k \geq 0}$. The exact solution of Eq. (6) is $f(\boldsymbol{a}, t) = \rho_0 \sum_{k=0}^{m-1} u_k(a_k - u_k t)$.

In our experiments in Sec. 4, we consider two types of FTEs: (i) $u_k := v_0$ for $k = 1$ (0 otherwise) and (ii) $u_k := v_0$ for $k \leq 14$ (0 otherwise), where $v_0$ is a constant. For convenience, we call them the *linear* and *nonlinear initial conditions*, respectively. The high-dimensional PDE thus obtained (Eq. (6)) is solved with PINNs (Sec. 3.2).

### 3.1.2 Application 2: Burgers-Hopf Equation

The second FDE is the *Burgers-Hopf equation* (BHE), a crucial equation in turbulence theory:

$$\frac{\partial F([\theta], t)}{\partial t} = \int dx \theta(x) \frac{\partial^2}{\partial x^2} \frac{\delta F([\theta], t)}{\delta \theta(x)} \tag{7}$$

with the initial condition $F([\theta], 0) = F_0([\theta])$, where $x \in [-1/2, 1/2]$. Specifically, we use the Gaussian initial condition $F_0([\theta]) = -\overline{\mu} \int dx \theta(x) + \frac{1}{2} \int dx \int dx' C(x, x') \theta(x) \theta(x')$, where $\overline{\mu}$ is a constant, and $C(x, x')$ is the infinite-dimensional covariance matrix. The exact solution is $F([\theta], t) = F_0([\Theta])$, where $\Theta([\theta], x, t) := \frac{1}{\sqrt{4\pi t}} \int_{-\infty}^{\infty} dx' e^{-\frac{1}{4t}(x-x')^2} \theta(x')$. The derivation is provided in App. D.2.1. Strictly speaking, Eq. (7) is a modification of the original BHE. The modification includes making the BHE dimensionless and neglecting the advection term. For more technical details, see App. E.2.

The cylindrical approximation of the BHE is given by

$$\frac{\partial}{\partial t} f(\boldsymbol{a}, t) = \sum_{k=0}^{m-1} \sum_{l=0}^{m-1} \int dx \frac{\partial \phi_k(x)}{\partial x^2} \phi_l(x) a_k \frac{\partial}{\partial a_l} f(\boldsymbol{a}, t) \tag{8}$$

with the initial condition $f(\boldsymbol{a}, 0) = f_0(\boldsymbol{a})$, where $\theta(x) = \sum_{k=0}^{m-1} a_k \phi_k(x)$, $f_0(\boldsymbol{a}) = F_0([P_m \theta]) = -\bar{\mu} a_0 + \frac{1}{2} \Sigma_{k,l=0}^{m-1} \tilde{C}_{kl} a_k a_l$ with $\tilde{C}_{kl} := \int dx \int dx' C(x, x') \phi_k(x) \phi_l(x')$. We use the Fourier series as the orthonormal basis: $\{\phi_k(x)\}_{k \geq 0} = \{1, \sqrt{2} \sin(\pi k x), \sqrt{2} \cos(\pi k x)\}_{k \geq 1}$. Then, the exact solution under the cylindrical approximation is given by

$$f(\boldsymbol{a}, t) = -\overline{\mu} a_0 + \Sigma_{k,l=0}^{M-1} \big( e^{-4\pi^2(k^2+l^2)t} \tilde{C}_{2k,2l} a_{2k} a_{2l}$$
$$+ e^{-4\pi^2(k^2+(l+1)^2)t} \tilde{C}_{2k,2l+1} a_{2k} a_{2l+1} + e^{-4\pi^2((k+1)^2+l^2)t} \tilde{C}_{2k+1,2l} a_{2k+1} a_{2l}$$
$$+ e^{-4\pi^2((k+1)^2+(l+1)^2)t} \tilde{C}_{2k+1,2l+1} a_{2k+1} a_{2l+1} \big)/2, \tag{9}$$

where $m = 2M$ ($M \in \mathbb{N}$). The derivation is given in App. D.2.2.

In our experiments in Sec. 4, we adopt two types of the covariance matrices: (i) $\tilde{C}_{ij} = \sigma^2$ for all $i = j \geq 0$ (0 otherwise) and (ii) $\tilde{C}_{ij} = \sigma^2$ for $i = j = 0$ (0 otherwise), where $\sigma^2$ is a constant. Substituting (i) and (ii) into $f_0$, we have two types of initial conditions, which we call the *delta and constant initial conditions*, respectively. Again, the high-dimensional PDE thus obtained (Eq. (8)) is solved with PINNs (Sec. 3.2).

### 3.2 Step 2: Solving Approximated FDEs with PINNs

We briefly introduce the foundation of PINNs [77]. PINNs are universal approximators and can solve PDEs. Let us consider a PDE $\partial f(t, x)/\partial t = \mathcal{N}[f]$ with an initial condition $\mathcal{B}[f]|_{t=0} = 0$, where $t \in [0, 1]$, $x \in [-1, 1]$. $\mathcal{N}$ and $\mathcal{B}$ are operators defining the PDE and the initial condition, respectively. The PINN aims to approximate the solution $f(t, x)$. Thus, the inputs to the PINN are $t$ and $x$, randomly sampled from $[0, 1]$ and $[-1, 1]$, respectively. Note that $(t = 0, x)$ with $x \in [-1, 1]$ are also input

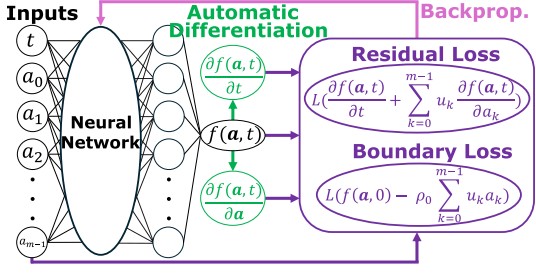

Figure 3: **PINN's architecture.**

to the PINN to compute the boundary loss. The inputs are transformed through linear layers and activation functions. The final output of the PINN is an approximation of $f(t, x)$, denoted by $\hat{f}(t, x)$. The loss function is the weighted sum of the residual loss $\|\partial \hat{f}(t, x)/\partial t - \mathcal{N}[\hat{f}]\|$ and the boundary loss $\|B[\hat{f}]|_{t=0}\|$, where $\|\cdot\|$ is a certain norm. The partial derivatives in the loss function can be computed via automatic differentiation of the PINN's output $\hat{f}$ w.r.t. the inputs $(t, x)$. Finally, the weight parameters of the PINN are minimized through backpropagation.

Next, we explain how to apply PINNs to our high-dimensional PDEs: see Fig. 3. For concreteness, consider the FTE (Eq. (6)) with the linear initial condition. The inputs to the PINN are $(t, a_0, a_1, \ldots, a_{m-1})$ and $(0, a_0, a_1, \ldots, a_{m-1})$, randomly sampled from finite intervals, and the outputs are $\hat{f}(\boldsymbol{a}, t)$ and $\hat{f}(\boldsymbol{a}, 0)$, respectively. Then, $\partial \hat{f}(\boldsymbol{a}, t)/\partial t$ and $\partial \hat{f}(\boldsymbol{a}, t)/\partial \boldsymbol{a}$ are computed via automatic differentiation, and we obtain the residual loss $\|\partial \hat{f}(\boldsymbol{a}, t)/\partial t + \Sigma_{k=0}^{m-1} u_k \partial \hat{f}(\boldsymbol{a}, t)/\partial a_k\|$ and the boundary loss $\|\hat{f}_0(\boldsymbol{a}) - \rho_0 \Sigma_{k=0}^{m-1} u_k a_k\|$, where $\hat{f}_0(\boldsymbol{a}) := \hat{f}(\boldsymbol{a}, 0)$. These losses are minimized via mini-batch optimization.

**Computational Complexity** The total computational complexity w.r.t. $m$ up to the computation of functional derivatives is given by $\mathcal{O}(m) + \mathcal{O}(m^r) = \mathcal{O}(m^r)$, where $r \geq 1$ is the order of the functional derivative included in the target FDE (typically 1 or 2). The first term $\mathcal{O}(m)$ comes from the input layer of the PINN. The second term $\mathcal{O}(m^r)$ comes from the computation of functional derivatives under the cylindrical approximation (Eq. (2)). See App. F.4 for more detailed discussions on computational complexity.

This is a notable reduction from discretization-based methods such as finite difference and element methods, which typically require exponentially large computational complexity w.r.t. the dimension of PDE $m$. Also, $\mathcal{O}(m^r)$ is significantly smaller than the state-of-the-art cylindrical approximation algorithm, the CP-ALS [91], which requires $\mathcal{O}(m^6)$ (the derivation is given in App. F.2). Consequently, given that $m$ represents the "class size" of input functions and functional derivatives (Eqs. (1) & (2)), our approach significantly extends the range of input functions and functional derivatives that can be represented via the cylindrical approximation. In fact, our approach extends the degrees of polynomials or Fourier series used for the approximation from 6 [91] to 1000 (Sec. 4).

Finally, we note that the selection of basis functions influences computational efficiency. The choice depends on the specific FDE, boundary conditions, symmetry, function spaces, and numerical stability. For further discussions, see App. F.3).

Table 1: **Mean relative and absolute errors of models trained on FTEs under linear (top two) and nonlinear (bottom two) initial conditions.** I.C. is short for "initial condition". The error bars are the standard deviation over 10 training runs with different random seeds. Note that this table is not for the assessment of the theoretical convergence of the cylindrical approximation (see Fig. 2, App. H.6, and the footnote in Sec. 4.1 instead).

| DEGREE | RELATIVE ERROR (LINEAR I.C.) |
|---:|---|
| 4 | $(1.26820 \pm 0.31421) \times 10^{-3}$ |
| 20 | $(2.01716 \pm 0.21742) \times 10^{-3}$ |
| 100 | $(6.24740 \pm 0.33492) \times 10^{-3}$ |

| DEGREE | ABSOLUTE ERROR (LINEAR I.C.) |
|---:|---|
| 4 | $(1.32203 \pm 0.44061) \times 10^{-3}$ |
| 20 | $(2.29632 \pm 0.16459) \times 10^{-3}$ |
| 100 | $(1.23312 \pm 0.18931) \times 10^{-3}$ |

| DEGREE | RELATIVE ERROR (NONLINEAR I.C.) |
|---:|---|
| 4 | $(1.79295 \pm 0.28535) \times 10^{-3}$ |
| 100 | $(7.63769 \pm 0.90872) \times 10^{-3}$ |
| 1000 | $(8.27096 \pm 1.19378) \times 10^{-3}$ |

| DEGREE | ABSOLUTE ERROR (NONLINEAR I.C.) |
|---:|---|
| 4 | $(2.37627 \pm 0.15278) \times 10^{-4}$ |
| 100 | $(1.84506 \pm 0.15765) \times 10^{-3}$ |
| 1000 | $(1.76470 \pm 0.36885) \times 10^{-3}$ |

In summary, our proposed approach transforms an FDE into a high-dimensional PDE using cylindrical approximation and then solves it with a PINN, which serves as a universal approximator of the solution (Figs. 1 & 3). It is important to note that our model employs the basic PINN framework, allowing for seamless integration with any techniques developed within the PINN community.

## 4 Experiment

Table 2: **Mean relative (top) and absolute (bottom) errors.** The models are trained on the BHE. The error bars are the standard deviation over 10 training runs with different random seeds. Note that this table is not for the assessment of the theoretical convergence of the cylindrical approximation (see Fig. 2, App. H.6, and the footnote in Sec. 4.1 instead).

| | INITIAL CONDITIONS | |
| --- | --- | --- |
| DEGREE | DELTA | CONSTANT |
| 4 | $(2.93905 \pm 0.17403) \times 10^{-4}$ | $(12.1782 \pm 8.54758) \times 10^{-5}$ |
| 20 | $(2.20842 \pm 0.28531) \times 10^{-4}$ | $(6.14352 \pm 1.28641) \times 10^{-5}$ |
| 100 | $(2.41667 \pm 0.25264) \times 10^{-4}$ | $(5.50375 \pm 1.75507) \times 10^{-5}$ |

| | INITIAL CONDITIONS | |
| --- | --- | --- |
| DEGREE | DELTA | CONSTANT |
| 4 | $(1.66451 \pm 0.47547) \times 10^{-5}$ | $(1.62849 \pm 0.66896) \times 10^{-5}$ |
| 20 | $(1.34640 \pm 0.12105) \times 10^{-5}$ | $(1.12371 \pm 0.32367) \times 10^{-5}$ |
| 100 | $(1.60980 \pm 0.08952) \times 10^{-5}$ | $(1.05558 \pm 0.16673) \times 10^{-5}$ |

As a proof of concept for our approach, we numerically solve the FTE and BHE.[1] These two FDEs are suitable for numerical experiments because their analytic solutions are available, allowing for the computation of absolute and relative errors, major metrics in the numerical analysis of PDEs and FDEs. Note that the analytic solutions for most FDEs are currently unknown due to their mathematical complexity.

**Setups.** We use a 4-layer PINN with $3\times$ (linear + sin activation + layer normalization [6]) + last linear layer. The total loss function is the smooth $L^1$ loss or the sum of $L^1$ and $L^\infty$ losses. Softmax loss-reweighting is employed. The optimizer is AdamW [59]. The learning rate scheduler is the linear warmup with cosine annealing with warmup [58]. Latin hypercube sampling [64, 39] is used for the training, validation, and test sets. For the BHE, the sampling range is decayed quadratically in terms of $k \in \{0, 1, \dots, m-1\}$ to stabilize the training. We use $L^1$ relative and absolute errors, standard performance metrics for numerical analysis of PDEs and FDEs. Absolute error $\frac{1}{N}\Sigma_{i=1}^{N}|f(\boldsymbol{a}_i, t_i) - \hat{f}(\boldsymbol{a}_i, t_i)|$ is used instead of relative error $\frac{1}{N}\Sigma_{i=1}^{N}|f(\boldsymbol{a}_i, t_i) - \hat{f}(\boldsymbol{a}_i, t_i)|/|f(\boldsymbol{a}_i, t_i)|$ when the analytic solution is close to zero because relative error in such a region blows up by definition, regardless of the model's prediction. $\upsilon_0$, $\rho_0$, $\bar{\mu}$, and $\sigma^2$ are set to $1, 1, 8$, and $10$, respectively. In App. G, we provide more detailed setups for reproducibility, including the range of $a_k$.

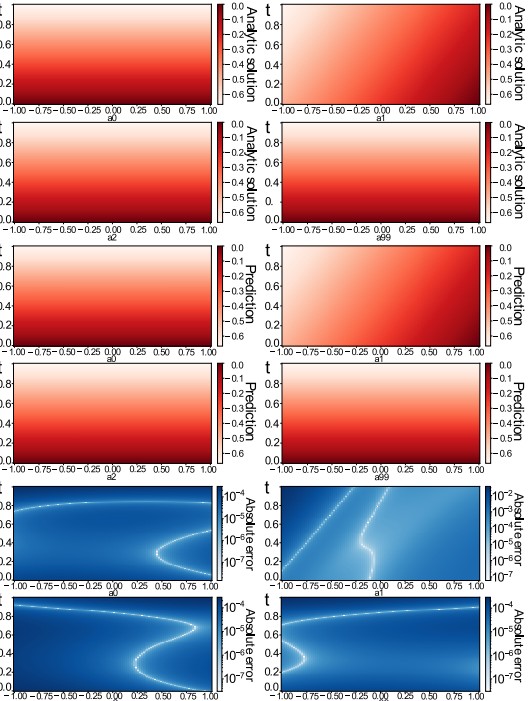

Figure 4: **Analytic solution (top four panels), prediction (second four panels), and absolute error (bottom four panels) of FTE with degree 100 under linear initial condition.** The horizontal axes represent $a_k$ for $k = 0, 1, 2, 99$, with all the other coefficients set to 0. Our model successfully learns the FTE.

---

[1]Code: `https://github.com/TaikiMiyagawa/FunctionalPINN`.

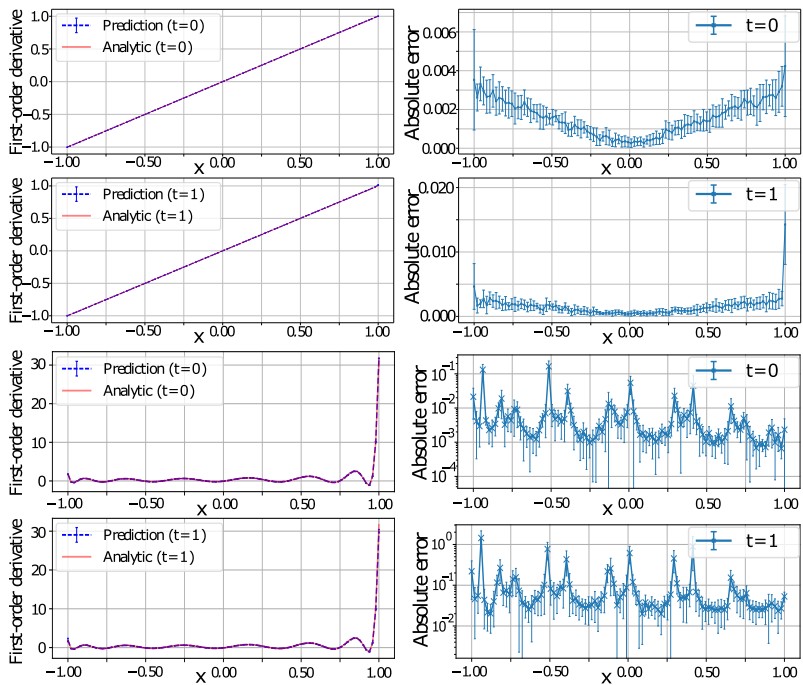

Figure 5: **Absolute error of first-order functional derivative of FTE with degree 100 (top) and 1000 (bottom) under linear (top) and nonlinear (bottom) initial conditions.** The error bars represent the standard deviation over 10 runs with different random seeds.

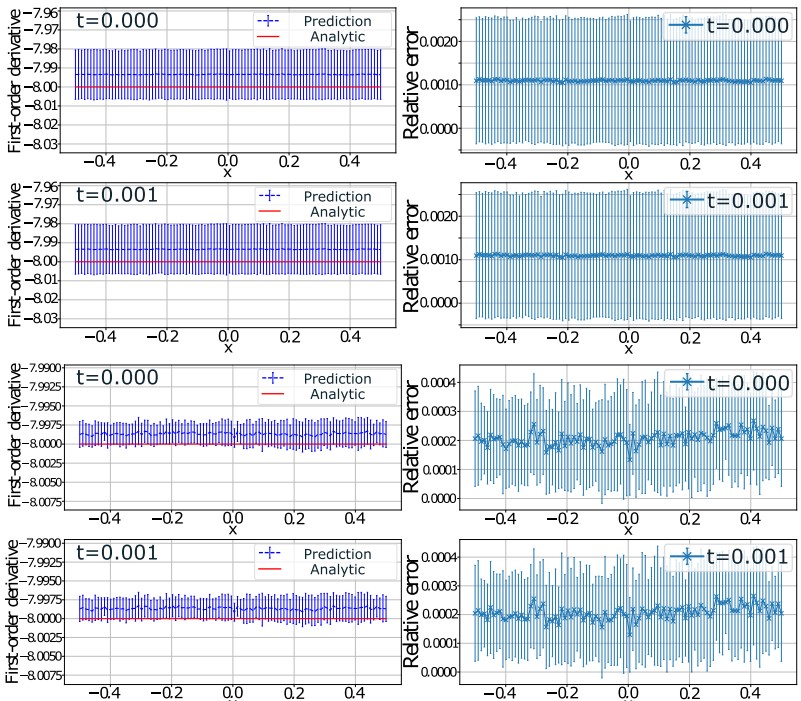

Figure 6: **Relative error of first-order functional derivative of BHE with degree 100 under delta initial condition.** The error bars represent the standard deviation over 10 runs. The top/bottom four panels show the results with/without the loss function $\|W([0], t)\|$, respectively. With this loss function, the error reduces by $10^{-1}$.

## 4.1 Result : Functional Transport Equation

**Tab. 1** shows the main result.[2] Our model achieves typical relative error orders of PINNs $\sim 10^{-3}$, even when the degree is as large as 1000, which means our model's capability of representing $\theta$ and $\delta F([\theta], t)/\delta\theta(x)$ as polynomials of degree 1000. This is a notable improvement from the state-of-the-art algorithm [91], which can handle $m \leq 6$.

**Fig. 4** visualizes the analytic solution, prediction, and absolute error of a model trained on the FTE with degree 100 under the linear initial condition. Note that some of the collocation points used for plotting Fig. 4 are not in the training set, as can be seen from Figs. 11–16 in App. H.3. This aspect highlights the model's ability to extrapolate beyond its training data ($a_k \sim 0$).

**Fig. 5** shows the absolute error of the first-order functional derivative estimated at $t = 1$ and $\theta = 0$. Again, $\theta = 0$ is not included in the training set. The errors in the top four panels increase at the edges of intervals ($x = \pm 1$) due to Runge's phenomenon [80].

## 4.2 Result: Burgers-Hopf Equation

**Tab. 2** shows the main result. Again, our model successfully achieves $\sim 10^{-3}$, the typical order of relative error of PINNs. See Fig. 2, App. H.6, and the footnote in Sec. 4.1 for the assessment of the theoretical convergence of the cylindrical approximation.

**Fig. 6** shows the relative error of first-order functional derivatives at $\theta = 0$. Note again that some of the collocation points used for this figure are not included in the training dataset, highlighting the model's ability to extrapolate beyond its training dataset ($a_k \sim 0$). Additionally, the error is reduced by a factor of $10^{-1}$ by incorporating a loss term corresponding to the identity $W([0], t) = 0$ (bottom four panels).

**Fig. 7** visualizes the analytic solution, prediction, and absolute error of a model trained on the BHE with degree 20 under the delta initial condition. The absolute error w.r.t. $a_0$ is $10^{-4}$ times smaller than the scale of the solution; i.e., the model learns $\theta$ well in the direction of $a_0$, which dominates the analytic solution. Conversely, the absolute error w.r.t. $a_{19}$ is on par with the scale of the solution. This result is anticipated because the dependence of the solution on $a_{19}$ is much smaller than $a_0$. This relationship is evident from Eq. (9), which indicates that the higher degree terms decay exponentially in terms of $k$, $l$, and $t$, and the solution is dominated by $a_k$ with $k \lesssim 1$. Therefore, optimizing the model in the direction of $a_{19}$ has only a negligible effect on minimizing the loss function.

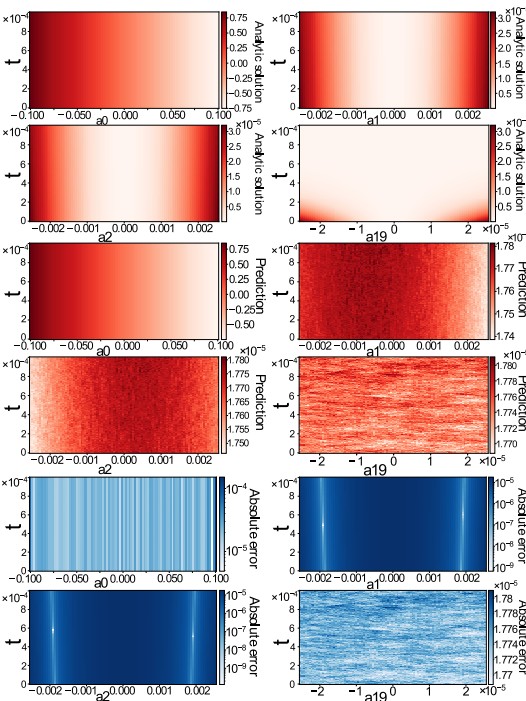

Figure 7: **Analytic solution (top four panels), prediction (second four panels), and absolute error (last four panels) of BHE with degree 20 under delta initial condition.** The horizontal axes are $a_k$ ($k = 0, 1, 2,$ or $19$). The other coefficients are kept $0$. Our model successfully learns the BHE.

Finally, many additional experimental results are provided in App. H, including a comparison with the CP-ALS algorithm.

---

[2]Note that Tabs. 1 & 2 are not for the assessment of the theoretical convergence of the cylindrical approximation, unlike Fig. 2, because the analytic solutions used for error computation vary across each row, depending on the degree. See App. H.6 instead, where we additionally perform a cross-degree evaluation.

# 5    Limitations

One limitation of our work is that the spacetime dimension is limited to $1 + 1$ ($t$ and $x$) in our experiments. However, generalization to $1 + d$ dimensions is feasible, albeit with additional computational costs. For $d > 1$ dimensional spaces, we have several options for expansion bases [79].

Another limitation is that the orders of functional derivatives in FDEs in our experiments are limited to $r = 1$. However, extending to $r \geq 2$ is straightforward. For instance, the cylindrical approximation of the second-order functional derivative is expressed as $\delta^2 F([\theta], t)/\delta\theta(x)\delta\theta(y) \approx \Sigma_{k,l=0}^{m-1} \partial^2 f(\boldsymbol{a}, t)/\partial a_k \partial a_l \phi_k(x)\phi_l(y)$, which can be computed via backpropagation twice.

Furthermore, this paper focuses exclusively on the abstract evolution equation. While this includes important FDEs (see Sec. 3), it does not cover certain equations, such as the Schwinger-Dyson equation or the Wetterich equation. Nonetheless, the mathematical foundations regarding the existence and uniqueness of these FDEs remain unestablished, which is beyond the scope of our paper. Once these foundations are defined, applying our model to these equations would be straightforward. More discussions on limitations, including technical ones, are provided in App. F.1.

## Acknowledgments and Disclosure of Funding

Taiki Miyagawa, an employee at NEC Corporation, contributed to this paper independently of his company role. He undertook this research as an independent researcher under the permission by the company. Takeru Yokota was supported by the RIKEN Special Doctoral Researchers Program.

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

# Appendix

# Contents

# A   Supplementary Related Work

**Note on [91] & [92].**   Most of the numerical results presented in [91] and [92] are derived from simulations based on analytically specified functions and functionals, without involving the numerical integration of FDEs. For instance, Fig. 2 in [92] does not depict the result of numerically solving an FDE. Instead, it illustrates the approximation error of the cylindrical approximation of an analytically given functional (refer to Eq. (127) in [92]). The performance of numerical integration of FDEs using CPU-based algorithms is given in Fig. 38 in [91], which shows the application of the CP-ALS and hierarchical Tucker (HT) methods for cases with $m \leq 6$. Note that the HT is reported to be slower than the CP-ALS [91].

**Applications of functionals.**   Functionals play a fundamental role in stochastic systems [101, 27, 49], Fokker-Planck equations [27], the statistical theory of turbulence [43, 3, 67], the theory of superfluidity [13, 83], quantum field theories [90, 91], mean-field games [16], many-body Schrödinger equations [42], mean-field optimal control [99, 81], and unnormalized optimal transport [32].

**Examples of FDEs.**   Examples of FDEs include the Schwinger-Dyson equation in quantum field theory [46], the Hopf characteristic functional equation in fluid mechanics and random processes [11, 49, 67], probability density functional equations, and effective Fokker-Planck systems [91].

**Operator learning.**   Functionals can be seen as operators that map a function to a scalar; thus, operator learning [4, 54, 60, 61] appears to be a promising approach to learning functionals. However, this method requires simulation or observation data unless PINNs are used simultaneously [55, 95]. In other words, operator learning methods solve inverse problems, while we focus on forward problems, where only the equation to be solved is given.

**Other approximation methods for FDEs.**   A common class of solvers for FDEs is based on truncating power series expansions at a finite order [67]. This includes the functional Taylor expansion, which expands a functional in terms of its argument functions. However, its applicability is limited because solutions obtained from the functional Taylor expansion can only be used for input functions close to the expansion center.

Another type of expansion used in the theory of functional renormalization group is the derivative expansion [23]. It is an expansion in terms of derivatives of the input functions. However, solutions are only feasible for inputs close to constants. For example, in the three-dimensional $\mathcal{O}(N)$ statistical model, derivative expansions up to the sixth order have been executed [23], but they are limited to calculations in uniform states and cannot handle non-uniform states.

Yet another expansion method uses the Reynolds number to distinguish between laminar and turbulent flow and has been applied to the Hopf equation [67]. However, increasing the truncation order poses a significant challenge. Specifically, calculating each expansion coefficient requires spatial integrals, leading to an exponential increase in computational costs.

Influence functions can be used for approximating Gateaux derivatives. In [48], the proposed approach is based on a finite-difference approximation of Gateaux derivatives, which requires a computational mesh for input function space when solving FDEs. Such an approach is infeasible because the number of mesh points grows exponentially with respect to the size of the input-function space.

**High-dimensional PDEs.**   In our experiments, with the help of PINNs, we numerically solved 1000-dimensional PDEs, which are impossible to handle with discretization-based methods, such as the finite element method. Numerical computation of high-dimensional PDEs is known to suffer from the curse of dimensionality, making PDEs with dimensions $d \geq 40$ particularly challenging [98].

However, rapid development in this field, especially in PINNs, has enabled solving much higher-dimensional PDEs. For example, a 100,000-dimensional PDE is now tractable [44], which can be combined with our model. Nevertheless, $d \sim \mathcal{O}(100)$ is typically sufficient in practice as long as input functions are regular. See also Fig. 10.

**PINNs.**   PINNs are a type of mesh-free universal approximators of PDE solutions [77]. There are several machine learning-based mesh-free solvers, e.g., [17].

**Automatic functional differentiation of higher-order functions.** Automatic differentiation of higher-order functions has a long history in theoretical computer science (see [68, 10] and the references therein)." Most studies focus on the mathematical aspects of programming languages, particularly how to implement reliable automatic functional differentiation, which is beyond our scope. We cite two recent papers that explicitly mention functional derivatives. Di et al. [22] develop a language to compute automatic functional derivatives; however, the implementation is not available. Lin [56] provides a JAX implementation of functional derivatives w.r.t. parameterized input functions. However, it supports only local, semilocal, and several nonlocal operators, limiting the functional space. In contrast, our model extends its approximability as $m$ increases.

**Density functional theory (DFT).** An alternative neural network-based approach to functional analysis utilizes finite element methods for spacetime grid approximation, commonly employed in first-principles computations of density functional theory (DFT) [26]. Examples include a neural network, $\hat{F}(\{y_j := f(\mathbf{r}_j)\}_j)$, approximating a target functional $F([f])$ by evaluating $f$ at specific grid points $\{\mathbf{r}_j\}_j$. Functional derivatives at each grid point can be computed using automatic differentiation: $\{\frac{\delta F([f])}{\delta f(\mathbf{r})}\}_{\mathbf{r}} \doteqdot \{\frac{\partial \hat{F}(\{y_j\}_j)}{\partial y_i}\}_i$. However, the central focus of the machine learning studies for DFT does not include solving PDEs, let alone FDEs.

**Neural functional networks.** Implicit Neural Representations (INRs) is another strategy to handle functions as the inputs to neural networks [107, 9, 62, 28]. However, this method requires a large number of weight parameters, resulting in substantial computational demands.

# B  Additional Introduction to FDEs

## B.1  Background

Functional differential equations (FDEs) are prevalent across various scientific fields, including the Hopf equation in statistical turbulence theory [43], the Schwinger-Dyson equation in quantum field theory [71], the functional renormalization group equation [97, 102, 72, 100], the Fokker-Planck equation in statistical mechanics [27], and equations for the energy density functional in DFT [73, 82]. The strength of FDEs lies in their comprehensive nature, enabling the derivation of various statistical properties of physical systems. For instance, the Hopf equation yields the characteristic functional, encompassing all information about simultaneous correlations of velocities at different positions, a crucial quantity in turbulence theory. Therefore, a highly accurate, efficient, and universal method for solving FDEs significantly impacts a broad range of scientific research but is yet to be explored.

A common class of solvers for FDEs is based on truncating power series expansions at a finite order [67]. This includes the functional Taylor expansion, which expands a functional in terms of its argument functions. However, its applicability is limited because solutions obtained from the functional Taylor expansion can only be used for input functions close to the expansion center.

Another type of expansion used in the functional renormalization group theory is the derivative expansion [23]. It is an expansion in terms of derivatives of the input functions. However, solutions are only feasible for inputs close to constants. For example, in the three-dimensional $\mathcal{O}(N)$ statistical model, derivative expansions up to the sixth order have been executed [23], but they are limited to calculations in uniform states and cannot treat non-uniform states.

Yet another expansion method uses the Reynolds number, distinguishing between laminar and turbulent flow, and has been applied to the Hopf equation [67]. However, increasing the truncation order poses a significant challenge. Specifically, calculating each expansion coefficient requires spatial integrals, leading to an exponential increase in computational costs.

An alternative to solving FDEs is the cylindrical approximation [91]. In this method, the input function is expanded using a set of basis functions truncated at a finite degree. The cylindrical approximation transforms an FDE into a high-dimensional PDE, and discretization-based methods are often used together. To address its high computational cost, tensor decomposition methods are also used. Canonical Polyadic (CP) tensor expansion with the Alternating Least Squares (ALS) method [91] is the state-of-the-art algorithm in this class. The reported results to date are limited to cases with six or fewer basis functions. The computational cost related to $m$ is at least $\mathcal{O}(m^6)$, presenting a challenge in increasing the number of bases $m$. In contrast, our model scales $\sim \mathcal{O}(m) + \mathcal{O}(m^r)$, where $r$ is the order of the functional derivative included in the target FDE. $r$ is typically 1 or 2, and thus the dependence on $m$ is $\sim \mathcal{O}(m)$ or $\mathcal{O}(m^2)$.

An alternative to solving FDEs is the cylindrical approximation [91]. In this method, the input function is expanded using a set of basis functions truncated at a finite degree. The cylindrical approximation transforms an FDE into a high-dimensional PDE, often solved using discretization-based methods. To address the high computational cost, tensor decomposition methods are also used. The CP-ALS method [91] is the state-of-the-art algorithm in this class. The reported results to date are limited to cases with six or fewer basis functions. The computational cost related to $m$ is at least $\mathcal{O}(m^6)$, presenting a challenge in increasing the number of basis functions $m$. In contrast, our model scales as $\mathcal{O}(m^r)$, where $r$ is the order of the functional derivative in the target FDE. Typically, $r$ is 1 or 2.

Below, we provide examples from the fields of turbulence, quantum field theory, and density functional theory.

## B.2  Turbulence

Turbulence appears everywhere, from natural systems (e.g., river flows and wind currents) to artificial systems (e.g., water flow in pipes and airflow over airplane surfaces). Understanding its properties is important both in natural sciences and in engineering. However, turbulence is a very complex system involving many degrees of freedom, and the only way to theoretically represent the properties of turbulence is through statistical methods. The Hopf equation [43] is an FDE that comprehensively describes the properties of turbulence. For example, the Hopf equation for a fluid following the

Navier-Stokes equations is written as follows:

$$\frac{\partial \Phi([\boldsymbol{\theta}], t)}{\partial t} = \sum_{k=1}^{3} \int_V d\boldsymbol{x}\, \theta_k(\boldsymbol{x}) \left( \frac{i}{2} \sum_{j=1}^{3} \frac{\partial}{\partial x_j} \frac{\delta^2 \Phi([\boldsymbol{\theta}], t)}{\delta \theta_j(\boldsymbol{x}) \delta \theta_k(\boldsymbol{x})} + \nu \nabla^2 \frac{\delta \Phi([\boldsymbol{\theta}], t)}{\delta \theta_k(\boldsymbol{x})} \right). \tag{10}$$

Here, $\Phi([\boldsymbol{\theta}], t)$ is a characteristic functional of a test function $\boldsymbol{\theta}(\boldsymbol{x}) = (\theta_1(\boldsymbol{x}), \theta_2(\boldsymbol{x}), \theta_3(\boldsymbol{x}))$, and $\nu$ is the kinematic viscosity. The characteristic functional contains all the information about the correlations of velocities at different locations at the same time. Indeed, the moments of velocity can be obtained from the derivatives of the characteristic functional as follows:

$$\langle u_{i_1}(\boldsymbol{x}_1, t) \cdots u_{i_n}(\boldsymbol{x}_n, t) \rangle_{\mathrm{m}} = (-i)^n \left. \frac{\delta^n \Phi([\boldsymbol{\theta}], t)}{\delta \theta_{i_1}(\boldsymbol{x}_1) \cdots \delta \theta_{i_n}(\boldsymbol{x}_n)} \right|_{\boldsymbol{\theta}=0}, \tag{11}$$

where $u_i(\boldsymbol{x}, t)$ is the $i$the component of the velocity at $(\boldsymbol{x}, t)$. In particular, the Fourier transform of the second-order cumulant is known as the energy spectrum, which represents the contribution of eddies at various momentum scales to turbulence. The search for a comprehensive method to describe the behavior of the energy spectrum across a wide range of momentum scales continues, and for this reason, developing methods to solve the Hopf equation holds significant importance.

More specifically, let us consider the fluid mechanics of aircraft or pipeline flow. For such systems, the second-order functional derivative of the solution of the Hopf characteristic functional gives the two-point correlation function of the velocity field at arbitrary two positions $x$ and $y$ and an arbitrary time $t$. The Fourier transform of it w.r.t. $x$ and $y$ is called the energy spectrum, which indicates which scales of motion are most energetic in the fluid flow. The energy spectrum is used to model and predict the behavior of turbulence, e.g., constructing safe and efficient shapes of airplanes or pipelines [69].

## B.3 Quantum Field Theory

Quantum mechanics tells us that physical quantities in the microscopic world do not always have deterministic values but fluctuate. In quantum field theory (QFT), which is a branch of quantum mechanics and forms the basis of modern particle physics, particles are described as fluctuating *fields* spreading throughout spacetime. QFT allows us to describe the properties of elementary particles in a statistical way, i.e., the correlation functions of fields at different points in spacetime. Therefore, developing methods to calculate correlation functions is very important for understanding the properties of elementary particles.

Several FDEs provide information on the correlation function of fields. A well-known example is the Schwinger-Dyson equation [71]. In the statistical model known as the $\phi^4$ model, which is described by the following action

$$S([\phi]) = \int d\boldsymbol{x} \left( \frac{1}{2} \phi(\boldsymbol{x}) \left( -\nabla^2 + m^2 \right) \phi(\boldsymbol{x}) + \frac{\lambda}{4!} \phi(\boldsymbol{x})^4 \right), \tag{12}$$

the Schwinger-Dyson equation is given as follows:

$$-\nabla^2 \frac{\delta Z([J])}{\delta J(\boldsymbol{x})} + m^2 \frac{\delta Z([J])}{\delta J(\boldsymbol{x})} - \frac{\lambda}{3!} \frac{\delta^3 Z([J])}{\delta J(\boldsymbol{x})^3} - iJ(\boldsymbol{x})Z([J]) = 0. \tag{13}$$

Here, $Z([J])$ is a quantity known as the partition function, and by functionally differentiating this quantity w.r.t. $J(\boldsymbol{x})$, all correlation functions for the field $\phi(\boldsymbol{x})$ can be obtained. Another method to describe the correlation functions is the functional renormalization group [97, 102, 72, 100]. The renormalization group is a method of analyzing physical systems based on the operation of reducing spacetime resolution. Under such operations, we can define an FDE for the effective action $\Gamma([\phi])$, which is calculated by the Legendre transformation of $\ln Z([J])$. $\Gamma([\phi])$ contains all the information of the correlation functions, similarly to $Z([J])$. $\Gamma([\phi])$ satisfies the following FDE [100]:

$$\partial_k \Gamma_k([\phi]) = \int d\boldsymbol{x} \int d\boldsymbol{x}' \partial R_k(\boldsymbol{x} - \boldsymbol{x}') G_k([\phi], \boldsymbol{x}', \boldsymbol{x}). \tag{14}$$

$k$ represents the momentum scale that specifies the resolution at which spacetime is observed, $R_k(\boldsymbol{x})$ is a function manually provided to realize the operation of the renormalization group, and $G_k([\phi], \boldsymbol{x}', \boldsymbol{x})$ is the regulated propagator defined as

$$\int d\boldsymbol{x}' \left( \frac{\delta^2 \Gamma_k([\phi])}{\delta \phi(\boldsymbol{x}) \delta \phi(\boldsymbol{x}')} + R_k(\boldsymbol{x} - \boldsymbol{x}') \right) G_k([\phi], \boldsymbol{x}', \boldsymbol{x}'') = \delta(\boldsymbol{x} - \boldsymbol{x}''). \tag{15}$$

## B.4 Density Functional Theory

Density functional theory (DFT) is widely used in material science, quantum chemistry, and nuclear physics. The properties of materials and molecules are determined by the state of electrons, which follows the Schrödinger equation. Solving the Schrödinger equation becomes challenging especially when the material contains many electrons. Hohenberg and Kohn demonstrated that it is possible to determine the state of electrons by solving a variational equation for the density [42], instead of solving the Schrödinger equation directly. This is a density-based variational equation and has been shown to be easier to solve than the Schrödinger equation.

However, in DFT, exact calculations are usually not possible. The reason is that the Energy Density Functional (EDF), which provides the variational equation for density, cannot be precisely determined. Whether a good approximation for the EDF can be provided or not significantly influences the success of DFT calculations. There has been a lot of research on finding EDFs, including empirical approaches, for a long time. One recent approach is based on FDEs. Specifically, several FDEs are known to describe the evolution of the EDF when the two-body interaction $U(\boldsymbol{x} - \boldsymbol{x}')$, e.g., the Coulomb interaction between electrons, gradually increases [73, 82]. When the interaction is replaced with $\lambda U(\boldsymbol{x} - \boldsymbol{x}')$, and when $\lambda$ gradually increases, the FDE becomes:

$$\partial_\lambda \Gamma_\lambda([n]) = \frac{1}{2} \int d\tau \int d\boldsymbol{x} \int d\boldsymbol{x}' U(\boldsymbol{x} - \boldsymbol{x}') \left[ n(\boldsymbol{x}, \tau) n(\boldsymbol{x}', \tau) + G_\lambda([n], \boldsymbol{x}, \tau, \boldsymbol{x}', \tau') - n(\boldsymbol{x}, \tau) \delta(\boldsymbol{x} - \boldsymbol{x}') \right],$$
(16)

$$\int d\boldsymbol{x}' \int d\tau' \frac{\delta^2 \Gamma_\lambda([n])}{\delta n(\boldsymbol{x}, \tau) \delta n(\boldsymbol{x}', \tau')} G_\lambda([n], \boldsymbol{x}', \tau', \boldsymbol{x}'', \tau'') = \delta(\boldsymbol{x} - \boldsymbol{x}'') \delta(\tau - \tau'').$$
(17)

Here, $n(\boldsymbol{x}, \tau)$ is the density of electrons, and $\Gamma_\lambda([n])$ represents an effective action, which is an extension of the EDF [31, 87]. In addition to the coordinates $\boldsymbol{x}$ and $\boldsymbol{x}'$, a virtual dimension, known as the imaginary time $\tau$, is introduced. This FDE is expected to provide a new method for constructing the EDF [73, 82]. For example, the EDF of the three-dimensional electron system is derived from Eqs. (16–17) based on the functional Taylor expansion [105].

# C  Theoretical Background of Cylindrical Approximation and Convergence

The mathematical background of our theoretical contribution is provided to make our paper self-contained. This section is based on the recent development of the theory of cylindrical approximation [92] and the classical spectrum theory [41, 85]. The differentiability of functionals is discussed in App. C.3, showing that the solutions of the FDEs in our experiment are differentiable. The equivalence and difference between the functional, Fréchet, and Gâteaux derivative are summarized in App. C.3. They are equivalent in practical settings, and we do not distinguish them in this paper.

## C.1  Continuity and Compactness of Functionals

Let $X$ be a Banach space unless otherwise stated. In this paper, a functional on $X$ is defined as a map $F$ from $D(F) \subset X$ to $\mathbb{R}$, where $D(F)$ is the domain of $F$. Note that $\mathbb{R}$ cannot be replaced with $\mathbb{Q}$ in all the statements below. We first define pointwise continuity, uniform continuity, compactness, and complete continuity of functionals.

**Definition C.1** (Pointwise continuity of $F$). A functional $F : X \supset D(F) \to \mathbb{R}$ is continuous at $\theta \in D(F) \subset X$ if for any Cauchy sequence $\{\theta_n\}_n \subset D(F)$,

$$\lim_{n \to \infty} ||\theta_n - \theta||_X = 0 \Rightarrow \lim_{n \to \infty} |F([\theta_n]) - F([\theta])| = 0, \tag{18}$$

where $|| \cdot ||_X$ is the norm induced by $X$.

**Definition C.2** (Uniform continuity of $F$). A functional $F : X \supset D(F) \to \mathbb{R}$ is uniformly continuous on $D(F)$ if

$$\forall \epsilon > 0, \ \exists \delta > 0 \ s.t.$$
$$\forall \theta_1, \theta_2 \in D(F) \text{ satisfying} ||\theta_1 - \theta_2||_X \leq \delta,$$
$$|F([\theta_1]) - F([\theta_2])| < \epsilon \text{ holds}, \tag{19}$$

where $|| \cdot ||_X$ is the norm induced by $X$.

We simply say "continuous" in the following.

**Definition C.3** (Compactness of $F$). A functional $F : X \supset D(F) \to \mathbb{R}$ is compact on $D(F)$ if $F$ maps any bounded subset of $D(F)$ into a pre-compact subset of $\mathbb{R}$.

Recall that $A \subset \mathbb{R}$ is a pre-compact subset if the closure of $A$, denoted by $\bar{A}$, is compact.

**Definition C.4** (Complete continuity of $F$). A functional $F : X \supset D(F) \to \mathbb{R}$ is completely continuous on $D(F)$ if $F$ is continuous and compact on $D(F)$.

Based on these concepts, functional derivatives are defined.

## C.2  Boundedness, Closedness, Compactness, and Pre-compactness of Metric Space of Functions

Next, we define boundedness, closedness, compactness, and pre-compactness of a metric space of functions.

**Definition C.5** (Boundedness of metric space of functions). Let $X$ be a metric space of functions. $K \subset X$ is bounded if $\exists M \in \mathbb{R}$ s.t. $\forall \theta \in K, ||\theta||_X < M$.

**Definition C.6** (Closedness of metric space of functions). Let $X$ be a metric space of functions. $K \subset X$ is closed if any convergent sequence in $K$ has a limit in $K$.

**Definition C.7** (Compactness of metric space of functions). Let $X$ be a metric space of functions. $K \subset X$ is compact if any open cover of $K$ has a finite subcover. Equivalently, $K$ is compact if and only if any sequence in $K$ is a bounded subsequence whose limit is in $K$.

**Definition C.8** (Pre-compactness of metric space of functions). Let $X$ be a metric space of functions. $K \subset X$ is pre-compact if its closure $\bar{K}$ is compact. Equivalently, $K$ is pre-compact if and only if any sequence in $K$ has a convergent subsequence whose limit is in $X$.

A critical characteristic of pre-compactness is given by the following theorem (a necessary and sufficient condition for the pre-compactness of metric spaces).

**Theorem C.9** (Necessary and sufficient condition of pre-compactness of metric space [65])**.** *A subset E of a real separable Hilbert space $H$ is pre-compact if and only if $E$ is (i) bounded, (ii) closed, and (iii) for any orthonormal basis $\{\phi_0, \phi_1, \ldots\}$ of $H$ and for any $\epsilon > 0$, there exists $m \in \mathbb{N}$ s.t.*

$$\forall \theta \in E, \quad \sum_{k=m+1}^{\infty} |(\theta, \phi_k)_H|^2 \leq \epsilon, \tag{20}$$

*where $(\cdot, \cdot)_H$ is the inner product defined on $H$.*

Eq. (20) is known as the equi-small tail condition. Thm. C.9 characterizes the domain of functions $D(F)$ to which the cylindrical approximation can be applied. Note that boundedness is not a necessary nor sufficient condition for the equi-small tail condition.

## C.3 Differentiation of Functionals

Next, we define the Gâteaux differential, Fréchet differential, and functional derivative. Higher-order functional derivatives can be defined in a similar way [91].

**Definition C.10** (Gâteaux differential)**.** A functional $F : X \overset{\text{open}}{\supset} D(F) \to \mathbb{R}$ is Gâteaux differentiable at $\theta \in D(F)$ if

$$dF_\eta^{\text{G}}([\theta]) := \lim_{\epsilon \to 0} \frac{F([\theta + \epsilon\eta]) - F([\theta])}{\epsilon} \tag{21}$$

exists and is finite for all $\eta \in D(F)$, where $dF_\eta^{\text{G}}([\theta])$ is called the Gâteaux differential of $F$ in the direction $\eta$.

There are some patterns of differentiability conditions. One of them is:

**Theorem C.11** (Gâteaux differentiability of Lipschitz functionals [63, 5, 57])**.** *Let Banach space $X$ be separable. Then, any Lipschitz functional $F : X \overset{\text{open}}{\supset} D(F) \to \mathbb{R}$ is Gâteaux differentiable outside a Gauss-null set.*

Note that a Gauss-null set is a Borel set $A \subset X$ s.t. $\forall$ non-degenerate Gaussian measure $\mu$ on $X$, $\mu(A)$ is equal to 0. In this theorem, there is no guarantee for non-Lipschitz functionals, e.g., $F([\theta]) = \int dx \sqrt{|\theta(x)|}$, where $\theta(0) = 0$.

Under mild conditions, the Gâteaux derivative is defined, based on the Gâteaux differential.

**Theorem C.12** (Gâteaux derivative [86])**.** *If the following two conditions are satisfied, then the Gâteaux differential $dF_\eta([\theta])$ of functional $F : X \overset{\text{open}}{\supset} D(F) \to \mathbb{R}$ at $\theta \in D(F)$ in the direction $\eta \in D(F)$ can be represented as a linear operator acting on $\eta$, denoted by $F'([\theta])$, s.t.*

$$dF_\eta^{\text{G}}([\theta]) = F'([\theta])\eta, \tag{22}$$

*where $F'([\theta]) : D(F) \to \mathbb{R}$ is a linear operator, or a linear functional, depending on $\theta$ and is called the Gâteaux derivative of $F$ at $\theta$.*

1. *$dF_\eta^{\text{G}}([\theta])$ exists in some neighborhood of $\theta_0 \in D(F)$ and is continuous w.r.t. $\theta$ at $\theta_0$.*

2. *$dF_\eta^{\text{G}}([\theta_0])$ is continuous w.r.t. $\eta$ at $\eta = \eta_0$, where $||\eta_0||_X = 0$.*

Next, we define the second type of differentials, Fréchet differential.

**Definition C.13** (Fréchet differential)**.** A functional $F : X \overset{\text{open}}{\supset} D(F) \to \mathbb{R}$ is Fréchet differentiable at $\theta \in D(F)$ if $dF_\eta^{\text{F}}([\theta]) \in \mathbb{R}$ s.t.

$$\lim_{\epsilon \to 0} \frac{|F([\theta + \epsilon\eta]) - F([\theta]) - dF_\eta^{\text{F}}([\theta])|}{\epsilon} = 0 \tag{23}$$

exists and is finite for all $\eta \in D(F)$, where $dF_\eta^{\text{F}}([\theta])$ is called the Fréchet differential of $F$ in the direction $\eta$.

There are also some patterns of differentiability conditions. One of them is:

**Theorem C.14** (Fréchet differentiability of Lipschitz functionals [74])**.** *Let $K$ be a compact subset of a Hilbert space $H$. Then, any locally-Lipschitz functional $F : K \overset{\text{open}}{\supset} D(F) \to \mathbb{R}$ is Fréchet differentiable on a dense subset of $K$.*

An example functional that is Gâteaux differentiable but Fréchet non-differentiable is $F([\theta]) = \frac{a_k^3}{a_k^2 + a_l^2}$, where $\theta(x) = \sum_{i=0}^{\infty} a_i \phi_i(x)$ and $k, l \in \mathbb{N}$.

**Relationship between Gâteaux and Fréchet differential.** If $F$ has a continuous Gâteaux derivative on $D(F)$, then $F$ is Fréchet differentiable on $D(F)$, and these two derivatives are equal [86]. The Gâteaux derivative of the aforementioned example, $F([\theta]) = \frac{a_k^3}{a_k^2 + a_l^2}$, is not continuous, and thus, the Fréchet differentiability is not guaranteed. In the following, we consider functionals $F$ that are continuously Gâteaux differentiable in $D(F)$; therefore, we do not care about differentiability and do not distinguish Gâteaux derivatives from Fréchet derivatives. Hereafter, we write $dF_\eta([\theta]) := dF_\eta^{\mathrm{G}}([\theta]) = dF_\eta^{\mathrm{F}}([\theta]) = F'([\theta])\eta$

Next, we define the third type of derivatives.

**Definition C.15** (Functional derivative)**.** The functional derivative of a functional $F : X \overset{\text{open}}{\supset} D(F) \to \mathbb{R}$ w.r.t. $\theta(x)$ is defined as

$$\frac{\delta F([\theta])}{\delta \theta(x)} := \lim_{\epsilon \to 0} \frac{F([\theta(y) + \epsilon \delta(x - y)]) - F([\theta(y)])}{\epsilon}, \tag{24}$$

if it exists and is finite, where $\delta(x)$ is the Dirac delta.

Strictly speaking, this definition may be informal because $\theta(y)$ is a function, while $\delta(x - y)$ is a distribution. The representation theorem below (Thm. C.17) is sometimes regarded as the definition of the functional derivative.

Lem. C.16 and the Riesz representation theorem prove the following relation between the Fréchet derivative and the functional derivative.

**Lemma C.16** (Compactness of Fréchet derivative)**.** *Let $K$ be a compact subset of a real separable Hilbert space $H$. Let $F$ be a continuous functional on $H$. If the Fréchet derivative $F'([\theta])$ exists at $\theta \in K$, then it is a compact linear operator.*

**Theorem C.17** (Representation theorem of Fréchet derivative)**.** *Let $K$ be a compact subset of a real separable Hilbert space $H$. Let $F$ be a continuous functional on $H$. If the Fréchet derivative $F'([\theta])$ exists at $\theta \in K$, then the following unique integral representation of the Fréchet derivative holds:*

$$\forall \eta \in H, \ F'([\theta])\eta = \left( \frac{\delta F([\theta])}{\delta \theta}, \eta \right)_H, \tag{25}$$

*where $\frac{\delta F([\theta])}{\delta \theta(x)} \in H$.*

The representation theorem C.17 is the foundation of the cylindrical approximation of functional derivatives, which is shown below.

## C.4 Cylindrical Approximation

### C.4.1 Functionals

Let $H$ be a real separable Hilbert space. The cylindrical approximation is based on the fact that any $\theta \in H$ can be represented uniquely in terms of an orthonormal basis $\{\phi_0, \phi_2, \ldots\}$ as $\theta(x) = \sum_{k=0}^{\infty} (\theta, \phi_k)_H \phi_k(x)$.

Thus, we can define

$$f(a_0, a_1, \ldots) := F([\sum_{k=0}^{\infty} a_k \phi_k]), \tag{26}$$

where $a_k := (\theta, \phi_k)_H$. Truncating $k \leq m - 1$ ($m \in \mathbb{N}$) gives the cylindrical approximation of functionals:

**Definition C.18** (Cylindrical approximation of functionals [29, 7, 34, 88])**.** Let $P_m$ be the projection operator s.t. $P_m\theta := \sum_{k=0}^{m-1}(\theta,\phi_k)_H\phi_k$. Let $D_m$ be the finite-dimensional space induced by $P_m$; i.e., $D_m := \text{span}\{\phi_0,\phi_1,\ldots,\phi_{m-1}\}$. The cylindrical approximation of a functional $F([\theta])$ on $H$ is the $m$-dimensional multivariable function

$$f(a_0, a_1, \ldots, a_{m-1}) := F([P_m\theta]) = F([\sum_{k=0}^{m-1} a_k\phi_k(x)]), \tag{27}$$

where $a_k := (\theta, \phi_k)_H$. In short, $F([\theta]) \sim f(a_0, \ldots, a_{m-1})$.

The cylindrical approximation of functionals is uniform:

**Theorem C.19** (Uniform convergence of cylindrical approximation of functionals [75])**.** *Let $K$ be a compact subset of a real separable Hilbert space $H$. Let $F$ be a continuous functional on $H$. Then,*

$$\forall\epsilon > 0, \; \exists M \in \mathbb{N} \; s.t. \; \forall m \geq M, \; \forall\theta \in K, \; |F([\theta]) - F([P_m\theta])| < \epsilon \tag{28}$$

*holds; i.e., $F([P_m\theta])$ converges uniformly to $F([\theta])$ on $K$.*

The convergence rate is given by the mean value theorem of functionals:

**Theorem C.20** (Convergence rate of cylindrical approximation of functionals [92])**.** *Let $K$ be a compact and convex subset of a real separable Hilbert space $H$. Let $F$ be a continuously differentiable functional on $K$. Then,*

$$\forall\theta \in K, \; |F([\theta]) - F([P_m\theta])| \leq \sup_{\eta\in K}\|F'([\eta])\| \, \|\theta - P_m\theta\|_H \; . \tag{29}$$

$\|F'([\eta])\|$ is the operator norm of the linear operator $F'([\eta])$; i.e., $\|F'([\theta])\| := \sup_{\eta(\neq 0)\in H} \frac{|F'([\theta])\eta|}{\|\eta\|_H}$. The convergence rate of $\mathcal{O}(\|\theta - P_m\theta\|_H)$ depends on the basis and is provided in [41] (Chapters 4 & 6) and [85] (Sec. 3.5) for several bases.

### C.4.2 Functional Derivatives

The cylindrical approximation of functional derivatives is motivated by the representation theorem C.17, which states that (i) $\frac{\delta F([\theta])}{\delta\theta(x)} \in H$ and (ii) $F'([\theta])\eta = (\frac{\delta F([\theta])}{\delta\theta(x)}, \eta)_H$. Statement (i) means that $\frac{\delta F([\theta])}{\delta\theta(x)}$ can be represented in terms of an orthogonal basis as

$$\frac{\delta F([\theta])}{\delta\theta(x)} = \sum_{k=0}^{\infty}(\frac{\delta F([\theta])}{\delta\theta(x)}, \phi_k)_H\phi_k(x) \; . \tag{30}$$

Statement (ii) means that

$$\frac{\partial f}{\partial a_k} = \lim_{\epsilon\to 0}\frac{f(a_0, \ldots, a_k + \epsilon, \ldots) - f(a_0, \ldots, a_k, \ldots)}{\epsilon} \tag{31}$$

$$= \lim_{\epsilon\to 0}\frac{1}{\epsilon}\left[F([\sum_{l=0}^{\infty} a_l\phi_l + \epsilon\phi_k]) - F([\sum_{l=0}^{\infty} a_l\phi_l])\right] \quad (\because \text{Eq. (26)}) \tag{32}$$

$$= F'([\sum_{l=0}^{\infty} a_l\phi_l])\phi_k \quad (\because \text{Eqs. (21–22)}) \tag{33}$$

$$= (\frac{\delta F([\theta])}{\delta\phi_k}, \phi_k)_H \quad (\because \text{Eq. (25)}), \tag{34}$$

where $\theta(x) = \sum_{l=0}^{\infty} a_l\phi_l(x)$ with $a_l = (\theta, \phi_l)_H$. Eqs. (30) and (34) gives

$$\frac{\delta F([\theta])}{\delta\theta(x)} = \sum_{k=0}^{\infty}\frac{\partial f}{\partial a_k}\phi_k(x) \; . \tag{35}$$

Therefore, truncating $k \leq m - 1$ ($m \in \mathbb{N}$) gives the cylindrical approximation of functional derivatives:

**Definition C.21** (Cylindrical approximation of functional derivatives [91, 92])**.** The cylindrical approximation of a functional derivative $\frac{\delta F([\theta])}{\delta \theta(x)} \in H$ is defined as

$$\frac{\delta F([P_m \theta])}{\delta \theta(x)} = \sum_{k=0}^{m-1} \frac{\partial f}{\partial a_k} \phi_k(x) + \sum_{k=m}^{\infty} \left( \frac{\delta F([P_m \theta])}{\delta \theta}, \phi_k \right)_H \phi_k(x) .$$

If $\frac{\delta F([P_m \theta])}{\delta \theta(x)} \in D_m$, the second term on the left-hand side is equal to zero. $\sum_{k=m}^{\infty} \left( \frac{\delta F([P_m \theta])}{\delta \theta}, \phi_k \right)_H \phi_k(x)$ is called the *tail term* in this paper.

In short, $\frac{\delta F([\theta])}{\delta \theta(x)} \sim \sum_{k=0}^{m-1} \frac{\partial f}{\partial a_k} \phi_k(x)$ for $\frac{\delta F([\theta])}{\delta \theta(x)}$ that satisfies the equi-small tail condition.

Note that this version of the cylindrical approximation of functional derivatives is different from ours (Eq. 2). Specifically, in [91, 92], $P_m$ is not applied to $\delta F([\theta])/\delta \theta(x)$, and the emerging tail term $\Sigma_{k=m}^{\infty}(\delta F([\theta])/\delta \theta, \phi_k)\phi_k(x)$ is simply ignored without any rationale.

The cylindrical approximation of functional derivatives is uniform:

**Theorem C.22** (Uniform convergence of cylindrical approximation of functional derivatives [92])**.** *Let $K$ be a compact subset of a real separable Hilbert space $H$. Let $F$ be a continuously differentiable functional on $K$. Then,*

$$\forall \epsilon > 0, \ \exists M \in \mathbb{N} \ s.t. \ \forall m \geq M, \ \forall \theta \in K, \ \left\| \frac{\delta F([\theta])}{\delta \theta} - \frac{\delta F([P_m \theta])}{\delta \theta} \right\|_H < \epsilon \qquad (36)$$

*holds; i.e., $\frac{\delta F([P_m \theta])}{\delta \theta}$ converges uniformly to $\frac{\delta F([\theta])}{\delta \theta}$.*

Note that this theorem proves the *uniform* convergence, while ours (Thm. 3.1) proves the *pointwise* convergence. The difference comes from that the uniform convergence of the tail term, which is assumed in the above theorem, can be violated when our cylindrical approximation Eq. (2) is used because the tail term is absent in Eq. (2).

Similarly, the cylindrical approximation of Fréchet derivatives is uniform:

**Theorem C.23** (Uniform convergence of cylindrical approximation of Fréchet derivatives [92])**.** *Let $K$ be a compact subset of a real separable Hilbert space $H$. Let $F$ be a continuously differentiable functional on $K$. Then,*

$$\forall \epsilon > 0, \ \exists M \in \mathbb{N} \ s.t. \ \forall m \geq M, \ \forall \theta \in K, \ \|F'([\theta]) - F'([P_m \theta])\| < \epsilon \qquad (37)$$

*holds; i.e., $F'([P_m \theta])$ converges uniformly to $F'([\theta])$.*

The convergence rate is given by:

**Theorem C.24** (Convergence rate of cylindrical approximation of Fréchet derivatives [92])**.** *Let $K$ be a compact and convex subset of a real separable Hilbert space $H$. Let $F$ be a differentiable functional on $K$ with continuous first- and second-order Fréchet derivatives. Then,*

$$\forall \theta \in K, \ \|F'([\theta]) - F'([P_m \theta])\| \leq \sup_{\eta \in K} \|F''([\eta])\| \|\theta - P_m \theta\|_H , \qquad (38)$$

*where $\|F''([\eta])\| := \sup_{\zeta, \xi \in H, \zeta \neq 0, \xi \neq 0} \frac{|F''([\eta])\zeta\xi|}{\|\zeta\|_H \|\xi\|_H}$.*

In terms of a functional derivative, this is rewritten as

$$\left\| \frac{\delta F([\theta])}{\delta \theta} - \frac{\delta F([P_m \theta])}{\delta \theta} \right\|_H \leq \sup_{\zeta \in K} \left( \left\| \frac{\delta^2 F([\zeta])}{\delta \theta \delta \theta} \right\| \right) \|\theta - P_m \theta\|_H , \qquad (39)$$

where $\left\| \frac{\delta^2 F([\zeta])}{\delta \theta \delta \theta} \right\| := \sup_{\xi, \xi' \in H, \xi \neq 0, \xi' \neq 0} \frac{\left| \frac{\delta^2 F([\zeta])}{\delta \theta \delta \theta} \xi \xi' \right|}{\|\xi\|_H \|\xi'\|_H} := \sup_{\xi, \xi' \in H, \xi \neq 0, \xi' \neq 0} \frac{\left| \left( \frac{\delta}{\delta \theta} \left( \frac{\delta F([\zeta])}{\delta \theta}, \xi \right)_H, \xi' \right)_H \right|}{\|\xi\|_H \|\xi'\|_H}$;

i.e., we regard $\frac{\delta^2 F([\zeta])}{\delta \theta \delta \theta}$ as an operator acting on $H \times H$.

### C.4.3 Pointwise Convergence of Functional Derivatives under Cylindrical Approximation

We prove the pointwise convergence of the functional derivative under the cylindrical approximation. We already noted that the cylindrical approximation of functional derivatives (36) uniformly converges. Below, we show that the convergence becomes pointwise if we omit the second term of the r.h.s. of Eq. (36); i.e., we use

$$P_m \frac{\delta F([P_m\theta])}{\delta\theta(x)} = \sum_{k=0}^{m-1} \frac{\partial f}{\partial a_k} \phi_k(x) \quad \text{(Eq.(2))}$$

as the approximated functional derivative.

**Theorem C.25** (Thm. 3.1. Pointwise convergence of cylindrical approximation). *Let $K$ be a compact subset of a real separable Hilbert space $H$. Let $F$ be a continuously differentiable functional on $K$. Then,*

$$\forall \text{orthonormal basis } \{\phi_0, \phi_1, \ldots\}, \ \forall\epsilon > 0, \ \forall\theta \in K, \ \exists M \in \mathbb{N} \text{ s.t. } \forall m \geq M, \ \left\| \frac{\delta F([\theta])}{\delta\theta} - P_m \frac{\delta F([P_m\theta])}{\delta\theta} \right\|_H < \epsilon,$$
$$(40)$$

*where $P_m$ is the projection onto $\text{span}\{\phi_0, \ldots, \phi_{m-1}\}$.*

Now, the convergence becomes pointwise. Technically, this is because the set $K' = \{\frac{\delta F([\theta])}{\delta\theta} : \theta \in K\}$ is not guaranteed to satisfy the equi-small tail condition

$$\forall \text{ orthonormal basis } \{\phi_0, \phi_1, \ldots\}, \ \forall\epsilon' > 0, \ \exists M \in \mathbb{N} \text{ s.t. } \forall m \geq M, \ \forall\theta \in K, \ \sum_{k=m}^{\infty} \left| \left( \frac{\delta F([\theta])}{\delta\theta}, \phi_k \right)_H \right|^2 < \epsilon,$$
$$(41)$$

while its boundedness $\sup_{\theta \in K} \| \frac{\delta F([\theta])}{\delta\theta} \|_H < \infty$ holds according to Thm. C.16. An example that converges pointwisely but not uniformly is

$$F([\theta]) = \begin{cases} \sum_{k=1}^{\infty} e^{-(k-\tan a_0)^2} a_k & (0 \leq a_0 < \pi/2) \\ 0 & (a_0 = \pi/2) \end{cases}, \ a_k = (\theta, \phi_k)_H,$$
$$(42)$$

defined on a compact subset

$$K = \left\{ \theta : a_k = (\theta, \phi_k)_H, \ 0 \leq a_k \leq \frac{\pi}{2(k+1)} \text{ for } k = 0, 1, \ldots \right\}.$$
$$(43)$$

Anyways, we have to use large $m$ in either case (Eq. (36) or (2)) when we want to approximate a complicated functional derivative, and the degree $m$ that is required for a sufficiently small approximation error depends on the smoothness, or spectral tail, of $\frac{\delta F([P_m\theta])}{\delta\theta(x)}$. As discussed in App. C.4.4, while the lack of uniform convergence may affect the convergence of the cylindrical approximation for linear FDEs in general, this is not problematic in our experiment. We use Eq. (2) as the approximated functional derivative in our experiment.

### C.4.4 Abstract Evolution Equations

We first provide related theorems to the convergence of equations (*consistency*) [92] and then those to the convergence of solutions (*stability*) [25, 36, 92].

**Definitions.** Let $\mathcal{F}(H)$ be a Banach space of functionals from a real separable Hilbert space $H$ into $\mathbb{R}$. We consider an abstract evolution equation [36]

$$\frac{\partial F([\theta], t)}{\partial t} = \mathcal{L}([\theta])F([\theta], t) \text{ with } F([\theta], 0) = F_0([\theta]),$$
$$(44)$$

where $F$ is in $\mathcal{F}(H)$, and $\mathcal{L}([\theta])$ is a linear operator in the set of closed, densely-defined, and continuous linear operators on $\mathcal{F}(H)$ denoted by $\mathcal{C}(\mathcal{F})$. Let $D(\mathcal{L})$ denote the domain of operator $\mathcal{L}$. Let $\mathcal{F}_m(H)$ be the Banach space of functionals on $H$ such that $\mathcal{F}_m(H) := \{F_m \mid F_m([\theta], t) = F([P_m\theta], t)\}$; in other words, $\mathcal{F}_m(H)$ is the set of cylindrically approximated functionals. Using $\mathcal{F}(H)$ and $\mathcal{F}_m(H)$, we define a continuous linear operator $B_m : \mathcal{F}(H) \ni F([\theta], t) \mapsto F([P_m\theta], t) \in$

$\mathcal{F}_m(H)$, which represents the cylindrical approximation of functionals. $B_m$ allows us to decompose the right-hand side of the abstract evolution equation:

$$B_m(\mathcal{L}([\theta])F([\theta],t)) = \mathcal{L}_m([\theta])F_m([\theta],t) + R_m([\theta],t), \tag{45}$$

where $\mathcal{L}_m([\theta])$ is a linear operator acting on the $m$-dimensional multivariable function $f(a_0,\ldots,a_{m-1},t) = F_m([\theta],t)$ (note that $\mathcal{L}_m([\theta])$ have nothing to do with $t$), and $R_m([\theta],t)$ is the functional residual.

**Convergence of equations (consistency).** Now, we can show that $|\mathcal{L}([\theta])F([\theta]) - \mathcal{L}_m([\theta])F_m([\theta])| = \mathcal{O}(\|\theta - P_m\theta\|_H)$ [92].

**Definition C.26** (Consistency of sequence of operators). A sequence of linear operators $\{\mathcal{L}_m\} \in \mathcal{C}(\mathcal{F}_m)$ is consistent with a linear operator $\mathcal{L} \in \mathcal{C}(\mathcal{F})$ if $\forall F \in D(\mathcal{L})$, $\exists$ a sequence $F_m \in D(\mathcal{L}_m)$ s.t. $\|F - F_m\| \xrightarrow{m\to\infty} 0$ and $\|\mathcal{L}F - \mathcal{L}_mF_m\| \xrightarrow{m\to\infty} 0$. If $\|\mathcal{L}F - \mathcal{L}_mF_m\| \xrightarrow{m\to\infty} \mathcal{O}(m^{-p})$, $\{\mathcal{L}_m\}$ is consistent with $\mathcal{L}$ to order $p(> 0)$.

**Theorem C.27** (Consistency of FDEs under cylindrical approximation [92]). *Let $H$ be a real separable Hilbert space. Let $F \in \mathcal{F}(H)$ and $\mathcal{L} \in \mathcal{L}(\mathcal{F})$. If $\mathcal{L}([\theta])F([\theta])$ is continuous in $\theta$, then the sequence of operators $\{\mathcal{L}_m\}$ is consistent with $\mathcal{L}$ on arbitrary compact subset $K$ of $H$, provided that $\forall \theta \in K$, $|R_m([\theta])| \xrightarrow{m\to\infty} 0$.*

**Corollary C.28** (Convergence of cylindrical approximation of FDEs [92]). *Let $H$ be a real separable Hilbert space. Let $F \in \mathcal{F}(H)$ and $\mathcal{L} \in \mathcal{L}(\mathcal{F})$. If $\mathcal{L}([\theta])F([\theta])$ is continuous in $\theta$, then the sequence of operators $\{\mathcal{L}_m\}$ is consistent with $\mathcal{L}$ to the same order as $\|\theta - P_m\theta\|_H$ on arbitrary compact, convex subset $K$ of $H$, provided that $\forall \theta \in K$, $|R_m([\theta])| = \mathcal{O}(\|\theta - P_m\theta\|_H)$ as $m \to \infty$ and that $\mathcal{L}([\theta])F([\theta])$ is continuously Fréchet differentiable in $K$. In short, $|\mathcal{L}([\theta])F([\theta]) - \mathcal{L}_m([\theta])F_m([\theta])| = \mathcal{O}(\|\theta - P_m\theta\|_H)$.*

**Convergence of solutions (stability).** Next, we show that $F_m([\theta],t) \to F([\theta],t)$ if and only if the cylindrical approximation is stable and consistent. Let us consider the approximated abstract evolution equation $\frac{\partial F_m([\theta],t)}{\partial t} = \mathcal{L}_m([\theta])F_m([\theta],t)$ with $F_m([\theta],0) = B_mF_0([\theta])$. It is said to be *consistent* with the original abstract evolution equation if Thm. C.27 holds.

**Definition C.29** (Stability of approximated equation). Suppose that $\mathcal{L}_m$ of the approximated abstract evolution equation generates a strongly continuous semigroup $e^{t\mathcal{L}_m}$. The approximated abstract evolution equation is stable if $\exists M, \omega$ s.t. $\|e^{t\mathcal{L}_m}\| \leq Me^{\omega t}$, where $M$ and $\omega$ are independent of $m$.

**Theorem C.30** (Convergence of solutions under cylindrical approximation [25, 36, 92]). *Let $K$ be a compact subset of a real separable Hilbert space $H$. Suppose that the approximated abstract evolution equation is well-posed, in the sense of an initial value problem, in the time interval $[0,T]$ with a finite $T$. Suppose also that $\mathcal{L}([\theta]) \in \mathcal{C}(\mathcal{F})$ generates a strongly continuous semigroup in $[0,T]$. Then, the approximated abstract evolution equation is stable and consistent in $K$ if and only if $\max_{t \in [0,T]} \max_{\theta \in K} |F_m([\theta],t) - F([\theta],t)| \xrightarrow{m\to\infty} 0$, provided that $F_m([\theta],0) \xrightarrow{m\to\infty} F_0([\theta])$. In short, $F_m([\theta],t) \to F([\theta],t)$ if and only if the cylindrical approximation is stable and consistent.*

For example, the cylindrical approximation of the following initial value problem is stable and consistent [92]: $\frac{\partial F([\theta],t)}{\partial t} = \int_0^{2\pi} \theta(x)\frac{\partial}{\partial x}\frac{\delta F([\theta],t)}{\delta\theta(x)}dx$ with $F([\theta],0) = F_0([\theta])$. To our knowledge, the convergence rate has been unknown so far.

**Remark 1.** Most of the approximation results for compact subsets of real separable Hilbert spaces hold also in compact subsets of Banach spaces admitting a basis. We refer the readers to Sec. 8 in [92].

**Remark 2.** Finally, we comment on how the lack of uniform convergence of $\frac{\delta F([\theta])}{\delta\theta} - P_m\frac{\delta F([P_m\theta])}{\delta\theta}$ affects the cylindrical approximation of linear FDEs. The difference $\frac{\delta F([\theta])}{\delta\theta} - P_m\frac{\delta F([P_m\theta])}{\delta\theta}$ is manifested in the functional residual $R_m([\theta],t)$ in Eq. (45). The lack of uniform convergence may have a negative effect on the consistency of the cylindrical approximation, given that the convergence of $R_m([\theta],t)$ is required in Thm. C.27. This issue, however, is circumvented in many cases. In fact, let us consider the scenario where functional derivatives in an FDE are expressed in terms of the inner-product $(v, \frac{\delta F([\theta])}{\delta\theta})_H$, which is satisfied by our examples ($v = u$ in the FTE and $v = A\boldsymbol{a}$

in Eq. (97)) (see App. E). Importantly, its cylindrical approximation $(v, P_m \frac{\delta F([P_m\theta])}{\delta\theta})_H$ uniformly converge to $(v, \frac{\delta F([\theta])}{\delta\theta})_H$ even if $P_m \frac{\delta F([P_m\theta])}{\delta\theta}$ does not uniformly converge to $\frac{\delta F([\theta])}{\delta\theta}$.

**Lemma C.31** (Uniform convergence of inner products (ours)). *Let $K$ and $K'$ be compact subsets of a real separable Hilbert space $H$. Let $F$ be a continuously differentiable functional on $K$. Then,*

$$\forall\epsilon > 0,\ \exists M \in \mathbb{N}\ s.t.\ \forall m \geq M,\ \forall\theta \in K,\ \forall\theta' \in K',\ \left|\left(\theta', \frac{\delta F([\theta])}{\delta\theta}\right)_H - \left(\theta', P_m\frac{\delta F([P_m\theta])}{\delta\theta}\right)_H\right| < \epsilon \tag{46}$$

*holds; i.e.,* $\left(\theta', P_m\frac{\delta F([P_m\theta])}{\delta\theta}\right)_H$ *converges uniformly to* $\left(\theta', \frac{\delta F([\theta])}{\delta\theta}\right)_H$ *on $K$ and $K'$.*

The proof is given in App. D.1. In App. E, we employ this lemma to show the consistency of our FDEs.

In short, Lem. C.31 states that the cylindrical approximation of inner products $(v, P_m\frac{\delta F([P_m\theta])}{\delta\theta})_H$ uniformly converge to $(v, \frac{\delta F([\theta])}{\delta\theta})_H$ even if $P_m\frac{\delta F([P_m\theta])}{\delta\theta}$ does not uniformly converge to $\frac{\delta F([\theta])}{\delta\theta}$. Because of this mechanism, in App. E, we show that the uniform convergence of $R_m$ is ensured, which is one of the assumptions for stability. The full proof of the convergence of solutions is provided in App. E.

# D  Proofs I

## D.1  Theorem: Pointwise Convergence of Functional Derivatives under Cylindrical Approximation

**Theorem D.1** (Thm. 3.1. Pointwise convergence of cylindrical approximation). *Let $K$ be a compact subset of a real separable Hilbert space $H$. Let $F$ be a continuously differentiable functional on $K$. Then,*

$$\forall \text{ orthonormal basis } \{\phi_0, \phi_1, \ldots\}, \ \forall \epsilon > 0, \ \forall \theta \in K, \ \exists M \in \mathbb{N}$$

$$s.t. \ \forall m \geq M, \ \left\| \frac{\delta F([\theta])}{\delta \theta} - P_m \frac{\delta F([P_m \theta])}{\delta \theta} \right\|_H < \epsilon, \tag{47}$$

*where $P_m$ is the projection onto $\mathrm{span}\{\phi_0, \ldots, \phi_{m-1}\}$.*

*Proof.* The triangle inequality gives

$$\left\| \frac{\delta F([\theta])}{\delta \theta} - P_m \frac{\delta F([P_m \theta])}{\delta \theta} \right\|_H \leq \left\| P_m \left( \frac{\delta F([\theta])}{\delta \theta} - \frac{\delta F([P_m \theta])}{\delta \theta} \right) \right\|_H + \left\| (1 - P_m) \frac{\delta F([\theta])}{\delta \theta} \right\|_H$$

$$\leq \left\| \frac{\delta F([\theta])}{\delta \theta} - \frac{\delta F([P_m \theta])}{\delta \theta} \right\|_H + \sqrt{\sum_{k=m}^{\infty} \left| \left( \frac{\delta F([\theta])}{\delta \theta}, \phi_k \right)_H \right|^2}. \tag{48}$$

The first term on the right-hand side converges to zero uniformly according to Thm. C.22. As for the second term, we first note that $\frac{\delta F([\theta])}{\delta \theta(x)}$ is bounded on $K$ in the sense of a function in $H$, according to Lem. C.16. This, together with Parseval's identity, implies $\|\frac{\delta F([\theta])}{\delta \theta}\|_H^2 = \sum_{k=0}^{\infty} |(\frac{\delta F([\theta])}{\delta \theta}, \phi_k)_H|^2 < \infty$. Therefore, the sequence of the partial sums $S_m = \sum_{k=0}^{m-1} |(\frac{\delta F([\theta])}{\delta \theta}, \phi_k)_H|^2$ is a convergent sequence and thus is a Cauchy sequence; i.e.,

$$\forall \epsilon' > 0, \ \exists M \in \mathbb{N}, \ s.t. \ \forall m, n \geq M, \ |S_m - S_n| < \epsilon'. \tag{49}$$

By taking $n \to \infty$, we can claim that

$$\forall \text{ orthonormal basis } \{\phi_0, \phi_1, \ldots\}, \ \forall \epsilon' > 0, \ \forall \theta \in K, \ \exists M \in \mathbb{N}$$

$$s.t. \ \forall m \geq M, \ \sum_{k=m}^{\infty} \left| \left( \frac{\delta F([\theta])}{\delta \theta}, \phi_k \right)_H \right|^2 < \epsilon'. \tag{50}$$

Therefore, the second term on the right-hand side of Ineq. (48) converges pointwisely. The theorem was thus proved.  $\square$

**Convergence rate.**  The convergence rates of the approximated functional derivative can be derived from Ineq. (48). The first term on the r.h.s. converges at the same rate as $||\theta - P_m\theta||$ (Eq. (39)). The convergence rate of $||\theta - P_m\theta||$ depends on the basis functions and is provided in [41] (Chapters 4 & 6) and [85] (Sec. 3.5) for several bases. The convergence rate of the second term on the r.h.s. depends on the compact subset of functions $K \in H$ under consideration, and further assumptions on $K$ are required.

**Lemma D.2** (Lem. C.31. Uniform convergence of inner products). *Let $K$ and $K'$ be compact subsets of a real separable Hilbert space $H$. Let $F$ be a continuously differentiable functional on $K$. Then,*

$$\forall \epsilon > 0, \ \exists M \in \mathbb{N} \ s.t. \ \forall m \geq M, \ \forall \theta \in K, \ \forall \theta' \in K',$$

$$\left| \left( \theta', \frac{\delta F([\theta])}{\delta \theta} \right)_H - \left( \theta', P_m \frac{\delta F([P_m \theta])}{\delta \theta} \right)_H \right| < \epsilon \tag{51}$$

*holds; i.e., $\left( \theta', P_m \frac{\delta F([P_m \theta])}{\delta \theta} \right)_H$ converges uniformly to $\left( \theta', \frac{\delta F([\theta])}{\delta \theta} \right)_H$ on $K$ and $K'$.*

*Proof.* Using the triangle inequality and the Cauchy-Schwarz inequality, we have

$$
\left| \left( \theta', \frac{\delta F([\theta])}{\delta \theta} \right)_H - \left( \theta', P_m \frac{\delta F([P_m \theta])}{\delta \theta} \right)_H \right|
$$

$$
\leq \left| \left( \theta', P_m \left( \frac{\delta F([\theta])}{\delta \theta} - \frac{\delta F([P_m \theta])}{\delta \theta} \right) \right)_H \right| + \left| \left( \theta', (1 - P_m) \frac{\delta F([\theta])}{\delta \theta} \right)_H \right|
$$

$$
\leq \|\theta'\|_H \left\| P_m \left( \frac{\delta F([\theta])}{\delta \theta} - \frac{\delta F([P_m \theta])}{\delta \theta} \right) \right\|_H + \left| \left( (1 - P_m)\theta', \frac{\delta F([\theta])}{\delta \theta} \right)_H \right|
$$

$$
\leq \|\theta'\|_H \left\| \frac{\delta F([\theta])}{\delta \theta} - \frac{\delta F([P_m \theta])}{\delta \theta} \right\|_H + \left\| \frac{\delta F([\theta])}{\delta \theta} \right\|_H \sqrt{\sum_{k=m}^{\infty} |(\theta', \phi_k)_H|^2}. \tag{52}
$$

Note that $\frac{\delta F([\theta])}{\delta \theta}$ and $\theta'$ are bounded on $K$ and $K'$, respectively; thus, $C$ and $C'$ s.t. $\sup_{\theta \in K} \left\| \frac{\delta F([\theta])}{\delta \theta} \right\|_H \leq C < \infty$ and $\sup_{\theta' \in K'} \|\theta'\|_H \leq C' < \infty$. Therefore, Ineq. (52) gives

$$
\left| \left( \theta', \frac{\delta F([\theta])}{\delta \theta} \right)_H - \left( \theta', P_m \frac{\delta F([P_m \theta])}{\delta \theta} \right)_H \right| \leq C' \left\| \frac{\delta F([\theta])}{\delta \theta} - \frac{\delta F([P_m \theta])}{\delta \theta} \right\|_H + C \sum_{k=m}^{\infty} |(\theta', \phi_k)_H|^2.
\tag{53}
$$

Finally, according to Thm. C.9, the compactness of $K'$ means

$$
\forall \epsilon' > 0, \ \exists M \in \mathbb{N} \ s.t. \ \forall \theta' \in K', \ \forall m \geq M, \ \sum_{k=m}^{\infty} |(\theta', \phi_k)_H|^2 < \epsilon'. \tag{54}
$$

This, together with Ineq. (53) and Thm. C.22, proves the lemma. $\qquad \square$

**Tail term.** In the cylindrical approximation (2), $P_m$ projects the "tail term" $\sum_{k=m}^{\infty} (\frac{\delta F([\theta])}{\delta \theta}, \phi)_H \phi_k(x)$ to zero, unlike the cylindrical approximation (36) adopted in [91, 92]. The tail term vanishes if one considers an inner product of a functional derivative and $v \in D_m$. However, it is not always the case that functional derivatives appear in the form of $(\frac{\delta F([\theta])}{\delta \theta}, v)$ **with** $v \in D_m$ in FDEs. In fact, in the FTE, the functional derivative appears as an inner product with $u(x)$, which can be chosen arbitrarily. The point is that Lem. C.31, which plays a fundamental role in proving the convergence of approximated solutions, guarantees the convergence of the inner product $(\frac{\delta F([\theta])}{\delta \theta}, v)$ in a wider variety of situations including $v \notin D_m$. In other words, our theorems extend the class of FDEs whose uniform convergence of the approximated solution is theoretically guaranteed.

### D.2 Derivation of Exact Solution of Burgers-Hopf Equation

#### D.2.1 Derivation of Eq. (93)

We show the derivation of Eq. (93). It is based on the functional Taylor expansion

$$
W([\Theta], \tau) = \sum_{n=0}^{\infty} \frac{1}{n!} \int_{-\frac{1}{2}}^{\frac{1}{2}} d\xi_1 \cdots \int_{-\frac{1}{2}}^{\frac{1}{2}} d\xi_n W^{(n)}(\tau, \xi_1, \ldots, \xi_n) \Theta(\xi_1) \cdots \Theta(\xi_n). \tag{55}
$$

This turns Eq. (92) as follows:

$$
\frac{\partial}{\partial \tau} W^{(n)}(\tau, \xi_1, \ldots, \xi_n) = \left( \frac{\partial^2}{\partial \xi_1^2} + \cdots + \frac{\partial^2}{\partial \xi_n^2} \right) W^{(n)}(\tau, \xi_1, \ldots, \xi_n). \tag{56}
$$

In the momentum space, this is written in the following form

$$
\frac{\partial}{\partial \tau} \tilde{W}^{(n)}(\tau, k_1, \ldots, k_n) = - \left( k_1^2 + \cdots + k_n^2 \right) \tilde{W}^{(n)}(\tau, k_1, \ldots, k_n), \tag{57}
$$

$$
W^{(n)}(\tau, \xi_1, \ldots, \xi_n) = \sum_{k_1} \cdots \sum_{k_n} e^{-ik_1 \xi_1 + \cdots - ik_n \xi_n} \tilde{W}^{(n)}(\tau, k_1, \ldots, k_n), \tag{58}
$$

where $k_{1,\cdots,n} = 2\pi l_{1,\cdots,n}$ with $l_{1,\cdots,n} \in \mathbb{Z}$. The solution of this equation is

$$\tilde{W}^{(n)}(\tau, k_1, \ldots, k_n) = e^{-(k_1^2 + \cdots + k_n^2)\tau}\tilde{W}^{(n)}(0, k_1, \ldots, k_n). \tag{59}$$

Using this result, Eq. (55) gives

$$
\begin{aligned}
W([\Theta], t) &= \sum_{n=0}^{\infty} \frac{1}{n!} \sum_{k_1} \cdots \sum_{k_n} \tilde{W}^{(n)}(\tau, k_1, \ldots, k_n)\tilde{\Theta}(-k_1) \cdots \tilde{\Theta}(-k_n) \\
&= \sum_{n=0}^{\infty} \frac{1}{n!} \int_{-\frac{1}{2}}^{\frac{1}{2}} d\xi_1 \cdots \int_{-\frac{1}{2}}^{\frac{1}{2}} d\xi_n W^{(n)}(0, x_1, \ldots, x_n) \prod_{m=1}^{n} \left( \sum_{k_m} e^{-k_m^2\tau + ik_m\xi_m}\tilde{\Theta}(-k_m) \right) \\
&= \sum_{n=0}^{\infty} \frac{1}{n!} \int_{-\frac{1}{2}}^{\frac{1}{2}} d\xi_1 \cdots \int_{-\frac{1}{2}}^{\frac{1}{2}} d\xi_n W^{(n)}(0, \xi_1, \ldots, \xi_n) \prod_{m=1}^{n} \left( \int_{-\frac{1}{2}}^{\frac{1}{2}} d\xi_m' \Theta(\xi_m') \sum_{k_m} e^{-k_m^2\tau + ik_m(\xi_m - \xi_m')} \right) \\
&= \sum_{n=0}^{\infty} \frac{1}{n!} \int_{-\frac{1}{2}}^{\frac{1}{2}} d\xi_1 \cdots \int_{-\frac{1}{2}}^{\frac{1}{2}} d\xi_n W^{(n)}(0, \xi_1, \ldots, \xi_n) \prod_{m=1}^{n} \left( \int_{-\frac{1}{2}}^{\frac{1}{2}} d\xi_m' \Theta(\xi_m') \sum_{l_m=-\infty}^{\infty} e^{-4\pi^2\tau l_m^2 + 2\pi i l_m(\xi_m - \xi_m')} \right),
\end{aligned}
\tag{60}
$$

where

$$\tilde{\Theta}(k) = \int_{-\frac{1}{2}}^{\frac{1}{2}} d\xi e^{ik\xi}\Theta(\xi). \tag{61}$$

The Poisson's summation formula transforms the summation w.r.t. $l_m$:

$$
\begin{aligned}
\sum_{l_m=-\infty}^{\infty} e^{-4\pi^2\tau l_m^2 + 2\pi i l_m(\xi_m - \xi_m')} &= \sum_{q=-\infty}^{\infty} \int_{-\infty}^{\infty} dy e^{-2\pi i q y} \left( e^{-4\pi^2\tau y^2 + 2\pi i y(\xi_m - \xi_m')} \right) \\
&= \frac{1}{\sqrt{4\pi\tau}} \sum_{q=-\infty}^{\infty} e^{-\frac{1}{4\tau}(\xi_m - \xi_m' - q)^2}.
\end{aligned}
\tag{62}
$$

Plugging this into Eq. (60), we arrive at the following expression:

$$W([\Theta], \tau) = \sum_{n=0}^{\infty} \frac{1}{n!} \int_{-\frac{1}{2}}^{\frac{1}{2}} d\xi_1 \cdots \int_{-\frac{1}{2}}^{\frac{1}{2}} d\xi_n W^{(n)}(0, \xi_1, \ldots, \xi_n) \prod_{m=1}^{n} \left( \int_{-\frac{1}{2}}^{\frac{1}{2}} d\xi_m' \frac{\Theta(\xi_m')}{\sqrt{4\pi\tau}} \sum_{q=-\infty}^{\infty} e^{-\frac{1}{4\tau}(\xi_m - \xi_m' - q)^2} \right). \tag{63}$$

From this result, we conclude that the solution is given by

$$W([\Theta], \tau) = W_0([\Theta_\tau([\Theta])]), \quad \Theta_\tau([\Theta], \xi) = \frac{1}{\sqrt{4\pi\tau}} \sum_{q=-\infty}^{\infty} \int_{-\frac{1}{2}}^{\frac{1}{2}} d\xi' e^{-\frac{1}{4\tau}(\xi - \xi' - q)^2} \Theta(\xi') \quad \text{(Eq. (93))}.$$

### D.2.2 Derivation of Eq. (104)

We show the derivation of Eq. (104). The cylindrical approximation of Eq. (93) is represented by

$$W([P_{2M}\Theta], \tau) = W_0([\Theta_\tau([P_{2M}\Theta])]), \quad \Theta_\tau([P_{2M}\Theta], \xi) = \frac{1}{\sqrt{4\pi\tau}} \sum_{q=-\infty}^{\infty} \int_{-1/2}^{1/2} d\xi' e^{-\frac{1}{4\tau}(\xi - \xi' - q)^2} P_{2M}\Theta(\xi').$$

The basis (103) satisfies

$$
\begin{aligned}
\phi_k(\xi + \xi') &= \begin{cases} 1 & (k = 0) \\ \sqrt{2}\sin(\pi(k+1)(\xi + \xi')) & (k : \text{odd}) \\ \sqrt{2}\cos(\pi k(\xi + \xi')) & (k : \text{nonzero even}) \end{cases} \\
&= \begin{cases} 1 & (k = 0) \\ \sqrt{2}\sin(\pi(k+1)\xi)\cos(\pi(k+1)\xi') + \sqrt{2}\cos(\pi(k+1)\xi)\sin(\pi(k+1)\xi') & (k : \text{odd}) \\ \sqrt{2}\cos(\pi k\xi)\cos(\pi k\xi') - \sqrt{2}\sin(\pi k\xi)\sin(\pi k\xi') & (k : \text{nonzero even}) \end{cases} \\
&= \begin{cases} \cos(\pi(k+1)\xi')\phi_k(\xi) + \sin(\pi(k+1)\xi')\phi_{k+1}(\xi) & (k : \text{odd}) \\ \cos(\pi k\xi')\phi_k(\xi) - \sin(\pi k\xi')\phi_{k-1}(\xi) & (k : \text{even}) \end{cases}.
\end{aligned}
\tag{64}
$$

Using this relation, we evaluate $\Theta_\tau([P_{2M}\Theta],\xi)$ as

$$
\begin{aligned}
\Theta_\tau([P_{2M}\Theta],\xi) =& \frac{1}{\sqrt{4\pi\tau}} \sum_{q=-\infty}^{\infty} \int_{-1/2}^{1/2} d\xi' e^{-\frac{1}{4\tau}(\xi'-q)^2} \sum_{k=0}^{2M-1} a_k \phi_k(\xi'+\xi) \\
=& \frac{1}{\sqrt{4\pi\tau}} \int_{-\infty}^{\infty} d\xi' e^{-\frac{\xi'^2}{4\tau}} \sum_{k=0}^{M-1} [\cos(2\pi k\xi')\phi_{2k}(\xi) - \sin(2\pi k\xi')\phi_{2k-1}(\xi)] a_{2k} \\
&+ \frac{1}{\sqrt{4\pi\tau}} \int_{-\infty}^{\infty} d\xi' e^{-\frac{\xi'^2}{4\tau}} \sum_{k=0}^{M-1} [\cos(2\pi(k+1)\xi')\phi_{2k+1}(\xi) + \sin(2\pi(k+1)\xi')\phi_{2k+2}(\xi)] a_{2k+1} \\
=& \sum_{k=0}^{M-1} \left( e^{-4\pi^2 k^2 \tau}\phi_{2k}(\xi)a_{2k} + e^{-4\pi^2(k+1)^2\tau}\phi_{2k+1}(\xi)a_{2k+1} \right),
\end{aligned}
\tag{65}
$$

where $a_k = (\phi_k,\Theta)_{L_{\mathrm{p}}^2([-1/2,1/2])}$.

When the initial condition is given by Eq. (102),

$$
\begin{aligned}
W([P_{2M}\Theta],\tau) =& -\bar{\mu} \int_{-1/2}^{1/2} d\xi\,\Theta_\tau(\xi) + \frac{1}{2} \int_{-1/2}^{1/2} d\xi \int_{-1/2}^{1/2} d\xi' C(\xi,\xi')\Theta_\tau([P_{2M}\Theta],\xi)\Theta_\tau([P_{2M}\Theta],\xi') \\
=& -\bar{\mu}a_0 + \frac{1}{2} \sum_{k=0}^{M-1}\sum_{l=0}^{M-1} \int_{-1/2}^{1/2} d\xi \int_{-1/2}^{1/2} d\xi' C(\xi,\xi') \\
& \times \left( e^{-4\pi^2 k^2\tau}\phi_{2k}(\xi)a_{2k} + e^{-4\pi^2(k+1)^2\tau}\phi_{2k+1}(\xi)a_{2k+1} \right) \left( e^{-4\pi^2 l^2\tau}\phi_{2l}(\xi')a_{2l} + e^{-4\pi^2(l+1)^2\tau}\phi_{2l+1}(\xi')a_{2l+1} \right) \\
=& -\bar{\mu}a_0 + \frac{1}{2} \sum_{k=0}^{M-1}\sum_{l=0}^{M-1} \left( e^{-4\pi^2(k^2+l^2)\tau}\tilde{C}_{2k,2l}a_{2k}a_{2l} + e^{-4\pi^2(k^2+(l+1)^2)\tau}\tilde{C}_{2k,2l+1}a_{2k}a_{2l+1} \right. \\
& \left. + e^{-4\pi^2((k+1)^2+l^2)\tau}\tilde{C}_{2k+1,2l}a_{2k+1}a_{2l} + e^{-4\pi^2((k+1)^2+(l+1)^2)\tau}\tilde{C}_{2k+1,2l+1}a_{2k+1}a_{2l+1} \right) \quad \text{(Eq. (104))},
\end{aligned}
$$

where we have introduced

$$
\tilde{C}_{kl} = \int_{-1/2}^{1/2} d\xi \int_{-1/2}^{1/2} d\xi' C(\xi,\xi')\phi_k(\xi)\phi_l(\xi').
$$

# E    Proofs II: Details of Functional Transport Equation and Burgers-Hopf Equation

In this Appendix, we provide the detailed background of the FTE and BHE and prove Thm. 3.2, i.e., the convergence of solutions under the cylindrical approximation (2). The main materials are Lem. C.31 and Thm. C.30. Technical assumptions are summarized in Thm. C.30.

## E.1    Functional Transport Equation

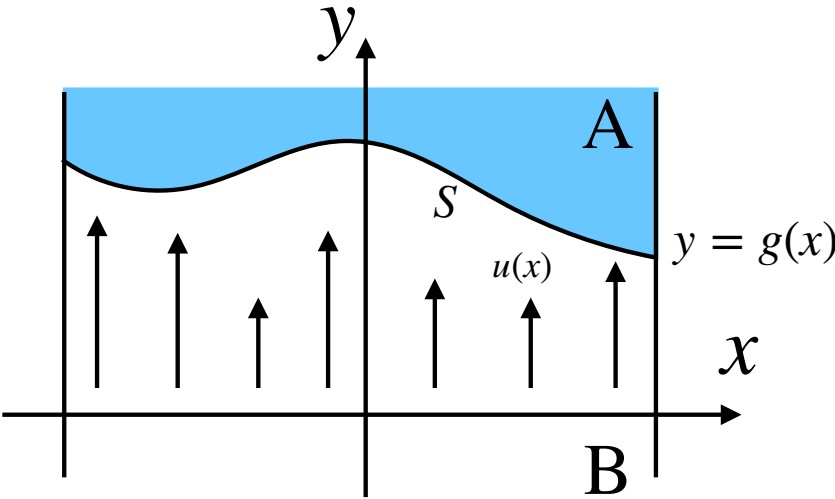

Figure 8: **Two-dimensional fluid.** Let $\boldsymbol{u}(\boldsymbol{x}) = (0, u(x))^\top$ denote velocity field. The surface $S$ is defined by $y = g(x)$, which separates the space into two areas, A and B.

We introduce the *functional transport equation* (FTE), a generalization of the transport equation (continuity equation) [53]. We consider a two-dimensional fluid system in $x$-$y$ coordinates (Fig. 8). We assume that the velocity field is given by $\boldsymbol{u}(\boldsymbol{x}) = (0, u(x))^\top$. The surface $S$, defined by $y = g(x)$, separates the space into two distinct areas, $y > g(x)$ (area A) and $y < g(x)$ (area B).

We calculate the amount of fluid entering the area A from B per unit time:

$$F([g], t) = \int dx\, u(x)\rho(x, g(x), t)\,, \tag{66}$$

where $\rho(x, y, t)$ denotes the fluid density at $(x, y)$ and at time $t$. Differentiating both sides w.r.t. $t$ and using the transport equation for the fluid density

$$\frac{\partial}{\partial t}\rho(x, y, t) = -u(x)\frac{\partial}{\partial y}\rho(x, y, t)\,, \tag{67}$$

we have

$$\frac{\partial}{\partial t}F([g], t) = -\int dx\, u(x)\frac{\delta F([g], t)}{\delta g(x)}\,. \tag{68}$$

We refer to this linear FDE as the FTE.

### E.1.1    Analytic Solution

Let $L^2([-1, 1])$ be a real Hilbert space of functions on $[-1, 1]$ with the inner product

$$(f, g)_{L^2([-1,1])} := \int_{-1}^{1} dx f(x)g(x), \tag{69}$$

where $f, g \in L^2([-1,1])$. Let $\mathcal{F}(L^2([-1,1]))$ be a Banach space of functionals that map $L^2([-1,1])$ to $\mathbb{R}$. The functional transport equation

$$\frac{\partial}{\partial t} F([g], t) = -\int_{-1}^{1} dx\, u(x) \frac{\delta F([g], t)}{\delta g(x)}, \quad F([g], 0) = F_0([g]), \tag{70}$$

is a linear FDE in $\mathcal{F}(L^2([-1,1]))$, where $F_0$ is a given functional. Here, $u \in L^2([-1,1])$ is assumed to have a finite norm $\|u\|_{L^2([-1,1])} < \infty$.

The solution of this equation is given by

$$F([g], t) = F_0[g_t], \quad g_t(x) := g(x) - u(x) t, \tag{71}$$

as can be seen immediately by substituting this into Eq. (70).

### E.1.2 Cylindrical Approximation and Convergence

We prove that the solution of the approximated equation converges to that of the non-approximated original equation. Let $\{\phi_0, \phi_1, \ldots\}$ be an orthonormal basis of $L^2([-1,1])$ and $P_m$ the projection onto $\mathrm{span}\{\phi_0, \ldots, \phi_{m-1}\}$. Following App. C.4.4, we apply $B_m$ on both sides and the initial condition, to obtain

$$\frac{\partial}{\partial t} f(\boldsymbol{a}, t) = -\sum_{k=0}^{m-1} u_k \frac{\partial}{\partial a_k} f(\boldsymbol{a}, t) + R_m([g], t), \quad f(\boldsymbol{a}, 0) = f_0(\boldsymbol{a}), \tag{72}$$

where we have introduced

$$\begin{aligned} f(\boldsymbol{a}, t) &:= F([P_m g], t), \quad f_0(\boldsymbol{a}, t) := F_0([P_m g]), \\ a_k &:= (\phi_k, g)_{L^2([-1,1])}, \quad u_k := (\phi_k, u)_{L^2([-1,1])}. \end{aligned} \tag{73}$$

The residual functional $R_m$ is defined as

$$R_m([g], t) := -\left( u, \frac{\delta F([g], t)}{\delta g} \right)_{L^2([-1,1])} + \left( u, P_m \frac{\delta F([P_m g], t)}{\delta g} \right)_{L^2([-1,1])}. \tag{74}$$

The cylindrical approximation of Eq. (70) is given by

$$\frac{\partial}{\partial t} f(\boldsymbol{a}, t) = -\sum_{k=0}^{m-1} u_k \frac{\partial}{\partial a_k} f(\boldsymbol{a}, t), \quad f(\boldsymbol{a}, 0) = f_0(\boldsymbol{a}). \tag{75}$$

We first prove the consistency of Eq. (75). According to Lem. C.31,[3] $R_m([g], t)$ uniformly converges to zero on an arbitrary compact subset $K$ in $L^2([-1,1])$ if $F([g], t)$ is a continuously differentiable functional on $K$. Thus, we can use Thm. C.27, which proves the consistency of the cylindrical approximation (75).

We next prove the stability (Def. C.29) of the cylindrical approximation (75) in the $L^\infty(\mathbb{R}^m)$ norm. Suppose that $f_0(\boldsymbol{a})$ is bounded by a constant $c$ independent of $m$.[4] From the cylindrical approximation of the solution Eq. (71), $f(\boldsymbol{a}, t) = F_0[P_m(g(x) - u(x)t)]$, we see

$$\sup_{\boldsymbol{a}} |f(\boldsymbol{a}, t)| = \sup_{\boldsymbol{a}} |F_0[P_m(g(x) - u(x)t)]| = \sup_{\boldsymbol{a}} |F_0[P_m g(x)]| = \sup_{\boldsymbol{a}} |f_0(\boldsymbol{a})|. \tag{76}$$

Therefore, we obtain

$$\|f(t)\|_{L^\infty(\mathbb{R}^m)} \left( = \|e^{t\mathcal{L}_m} f_0\|_{L^\infty(\mathbb{R}^m)} \right) = \|f_0\|_{L^\infty(\mathbb{R}^m)} \le c, \ \forall m \in \mathbb{N}, \tag{77}$$

---

[3] In this case, $K'$ in Lem. C.31 is set to a singleton $K' = \{u\}$. Obviously, this is compact because $\|u\|_{L^2([-1,1])} < \infty$.

[4] Strictly speaking, if $a_k$ are defined on an infinite interval, this assumption of boundedness is not valid for the FDEs used in our experiments because $\|F_0\|_{L^\infty(\mathbb{R}^m)} \to \infty$ as $a_k \to \infty$. However, the range of $a_k$ is usually set to a finite interval in numerical experiments, and thus $\|F_0\|_{L^\infty(\mathbb{R}^m)}$ is also finite; i.e., the assumption of boundedness holds.

for the operator $\mathcal{L}_m = -\sum_{k=0}^{m-1} u_k \frac{\partial}{\partial a_k}$. From this result, the operator norm of $e^{t\mathcal{L}_m}$ is given by

$$\|e^{t\mathcal{L}_m}\| = \sup_{f_0 \neq 0} \frac{\left\|e^{t\mathcal{L}_m} f_0\right\|_{L^\infty(\mathbb{R}^m)}}{\|f_0\|_{L^\infty(\mathbb{R}^m)}} = \sup_{f_0 \neq 0} \frac{\|f(t)\|_{L^\infty(\mathbb{R}^m)}}{\|f_0\|_{L^\infty(\mathbb{R}^m)}} = 1. \tag{78}$$

This shows that $\exists M, \omega$ s.t. $\forall m \in \mathbb{N}, \|e^{t\mathcal{L}_m}\| \leq M e^{\omega t}$ ($M = 1$ and $\omega = 0$), and thus the cylindrical approximation is stable in the sense of Def. C.29.

According to Thm. C.27, the fact that the cylindrical approximation (75) is consistent and stable guarantees the convergence $F([P_m g], t) \xrightarrow{m \to \infty} F([g], t)$ on a compact subset in a finite time interval $[0, T]$ if $F([P_m g], 0) \xrightarrow{m \to \infty} F([g], 0)$ (this is satisfied by the initial conditions used in our experiment) and if the approximated functional transport equation is well-posed in $[0, T]$.

### E.1.3 Initial Conditions

We define the initial condition $F([g], 0)$ in Eq. (70) as

$$F([g], 0) = \rho_0 \int_{-1}^{1} dx\, u(x) g(x). \tag{79}$$

The exact solution is given by

$$F([g], t) = \rho_0 \int_{-1}^{1} dx\, u(x) \left(g(x) - u(x)t\right). \tag{80}$$

Under the cylindrical approximation, the solution is

$$F([P_m g], t) = f(\boldsymbol{a}, t) = \rho_0 \sum_{k=0}^{m-1} u_k (a_k - u_k t). \tag{81}$$

With $\upsilon_0$ being a constant and $\{\phi_k\}_{k \geq 0}$ being the Legendre polynomials, we consider two types of $u(x)$:

$$u_k := \upsilon_0 \text{ for } k = 1, \text{ otherwise } 0, \tag{82}$$

$$u_k := \upsilon_0 \text{ for } k \leq 14, \text{ otherwise } 0, \tag{83}$$

corresponding to two types functional transport equations (70), initial conditions (79), and solutions (81). For convenience, we call them the linear initial condition and the nonlinear initial condition, respectively, though the form of Eq. (70) also changes in accordance with the form of $u(x)$.

### E.2 Burgers-Hopf Equation

The BHE, the one-dimensional analog of Eq. (10) in App. B.2, describes the statistical properties of one-dimensional fluids and is a basic tool for studying turbulence, as mentioned in App. B.2. We consider fluid in a one-dimensional box $[-L/2, L/2]$ that evolves in accordance with the Burgers equation

$$\frac{\partial u(x, t)}{\partial t} + u(x, t) \frac{\partial u(x, t)}{\partial x} = \nu \frac{\partial^2 u(x, t)}{\partial x^2}, \tag{84}$$

where $u(x, t)$ is a velocity field and $\nu$ is the kinematic viscosity. Specifically, we focus on the case where the initial value $u(x, 0)$ is given randomly, thus making $u(x, t)$ a random field. Let us introduce the characteristic functional

$$\Phi([\theta], t) = \mathop{\mathbb{E}}_{\{u(x,0)\} \sim \mathcal{P}_0} \left[ \exp\left( i \int_{-L/2}^{L/2} dx\, u(x, t) \theta(x) \right) \right], \tag{85}$$

where $\mathcal{P}_0$ is the probability distribution for the initial velocity field $u(x, 0)$, and $\theta(x)$ is called the test function. It provides statistical properties of the velocity field because the functional derivatives are equal to the moments of the velocity field:

$$\mathop{\mathbb{E}}_{\{u(x,0)\} \sim \mathcal{P}_0} [u(x_1, t) \cdots u(x_n, t)] = (-i)^n \left. \frac{\delta^n \Phi([\theta], t)}{\delta\theta(x_1) \cdots \delta\theta(x_n)} \right|_{\theta=0}, \tag{86}$$

as is already shown in Eq. (11) in App. B.2. The time evolution of $\Phi([\theta], t)$ is known to follow the BHE [91]:

$$\frac{\partial \Phi([\theta], t)}{\partial t} = \int_{-L/2}^{L/2} dx\, \theta(x) \left( \frac{i}{2} \frac{\partial}{\partial x} \frac{\delta^2 \Phi([\theta], t)}{\delta\theta(x)\delta\theta(x)} + \nu \frac{\partial^2}{\partial x^2} \frac{\delta \Phi([\theta], t)}{\delta\theta(x)} \right). \tag{87}$$

The initial condition $\Phi([\theta], 0)$ is determined by $\mathcal{P}_0$. For example, in the case of the Gaussian random field, we have

$$\Phi([\theta], 0) = \exp\left( i \int_{-L/2}^{L/2} dx\mu(x)\theta(x) - \frac{1}{2} \int_{-L/2}^{L/2} dx \int_{-L/2}^{L/2} dx' C(x, x')\theta(x)\theta(x') \right), \tag{88}$$

where $\mu(x)$ is the mean velocity and $C(x, x')$ is the infinite-dimensional covariance matrix.

We modify Eqs. (87) and (88) to facilitate numerical computation. First, we replace $-i\theta(x)$ with $\theta(x)$ and regard $\theta(x)$ as a real-valued function to avoid complex numbers. Second, we make Eqs. (87) and (88) dimensionless, a common convention in numerical computation, by introducing

$$T := \frac{L^2}{\nu}, \quad \tau := \frac{t}{T}, \quad \xi := \frac{x}{L}, \quad \Theta(\xi) := \frac{L^2}{T}\theta(L\xi). \tag{89}$$

Third, we introduce

$$W([\Theta], \tau) := \ln \Phi([\theta], t) \tag{90}$$

to remove the exponential function in $\Phi([\theta], t)$ and stabilize numerical computation. Note that the functional derivatives of $W([\Theta], \tau)$ give the cumulants, not moments, of the velocity field. Fourth, we neglect the advection term (the first term on the right-hand side of Eq. (87)). We derive the analytic solution for the equation in the following.

### E.2.1 Analytic Solution

Let us consider $L_{\mathrm{p}}^2([-1/2, 1/2])$, a real Hilbert space of periodic functions on $[-1/2, 1/2]$ with the inner product

$$(f, g)_{L_{\mathrm{p}}^2([-1/2, 1/2])} := \int_{-1/2}^{1/2} dx f(x)g(x), \tag{91}$$

where $f, g \in L_{\mathrm{p}}^2([-1/2, 1/2])$. Let $\mathcal{F}(L_{\mathrm{p}}^2([-1/2, 1/2]))$ be a Banach space of functionals that map $L_{\mathrm{p}}^2([-1/2, 1/2])$ to $\mathbb{R}$. In $\mathcal{F}(L_{\mathrm{p}}^2([-1/2, 1/2]))$, the BHE without the advection term is given by

$$\frac{\partial W([\Theta], \tau)}{\partial \tau} = \int_{-\frac{1}{2}}^{\frac{1}{2}} d\xi \Theta(\xi) \frac{\partial^2}{\partial \xi^2} \frac{\delta W([\Theta], \tau)}{\delta \Theta(\xi)}, \quad W([\Theta], 0) = W_0([\Theta]). \tag{92}$$

The exact solution of this equation is given by

$$W([\Theta], \tau) = W_0([\Theta_\tau([\Theta], \xi)]), \quad \Theta_\tau([\Theta], \xi) := \frac{1}{\sqrt{4\pi\tau}} \sum_{q=-\infty}^{\infty} \int_{-\frac{1}{2}}^{\frac{1}{2}} d\xi' e^{-\frac{1}{4\tau}(\xi-\xi'-q)^2} \Theta(\xi'), \tag{93}$$

where $W_0$ is a given functional. The proof is a bit technical and is given in App. D.2.1.

### E.2.2 Cylindrical Approximation and Convergence

In this section, we prove the convergence of the solution for the BHE under the cylindrical approximation. Let $\{\phi_0, \phi_1, \ldots\}$ be an orthonormal basis of $L_{\mathrm{p}}^2([-1/2, 1/2])$ and $P_m$ be the projection onto $\mathrm{span}\{\phi_0, \ldots, \phi_{m-1}\}$. Following App. C.4.4, we apply $B_m$ on both sides and the initial condition, to obtain

$$\frac{\partial}{\partial \tau} w(\boldsymbol{a}, \tau) = \sum_{k,l=0}^{m-1} A_{kl} a_k \frac{\partial}{\partial a_l} w(\boldsymbol{a}, \tau) + R_m([\Theta], \tau), \ w(\boldsymbol{\alpha}, 0) = w_0(\boldsymbol{a}), \tag{94}$$

where we have introduced

$$w(\boldsymbol{a}, \tau) := W([P_m \Theta], \tau), \ w_0(\boldsymbol{a}, \tau) := W_0([P_m \Theta]), \ a_k := (\phi_k, \Theta)_{L^2_{\mathrm{p}}([-1/2, 1/2])},$$

$$A_{kl} := \left(\varphi_k, \frac{\partial^2 \varphi_l}{\partial \xi^2}\right)_{L^2_{\mathrm{p}}([-1/2, 1/2])}. \tag{95}$$

The residual functional is defined as

$$R_m([\Theta], \tau) = \left(\frac{\partial^2 \Theta}{\partial \xi^2}, \frac{\delta W([\Theta], \tau)}{\delta \Theta}\right)_{L^2_{\mathrm{p}}([-1/2, 1/2])} - \left(\frac{\partial^2 \Theta}{\partial \xi^2}, P_m \frac{\delta W([P_m \Theta], \tau)}{\delta \Theta}\right)_{L^2_{\mathrm{p}}([-1/2, 1/2])}. \tag{96}$$

The cylindrical approximation of Eq. (92) is defined as

$$\frac{\partial}{\partial \tau} w(\boldsymbol{a}, \tau) = \sum_{k,l=0}^{m-1} A_{kl} a_k \frac{\partial}{\partial a_l} w(\boldsymbol{a}, \tau), \ w(\boldsymbol{a}, 0) = w_0(\boldsymbol{a}). \tag{97}$$

We first prove the consistency of Eq. (97). Let $K$ be a compact subset in $L^2_{\mathrm{p}}([-1/2, 1/2])$ such that $\overline{\{\frac{\partial^2 \Theta}{\partial \xi^2} : \Theta \in K\}}$ is compact. According to Lem. C.31, $R_m([W], \tau)$ uniformly converges to zero on $K$ if $W([\Theta], \tau)$ is a continuously differentiable functional on $K$. Thus, we can use Thm. C.27, which proves the consistency of Eq. (97).

We next prove the stability (Def. C.29) of Eq. (97) in the $L^\infty(\mathbb{R}^m)$ norm. Suppose that $w_0(\boldsymbol{a})$ is bounded by a constant $c$ independent of $m$.[5] From the cylindrical approximation of the solution Eq. (71), $w(\boldsymbol{a}, t) = W_0([P_m \Theta_\tau [P_m \Theta]])$, we see

$$\sup_{\boldsymbol{a} \in \mathbb{R}^m} |w(\boldsymbol{a}, \tau)| = \sup_{\boldsymbol{a} \in \mathbb{R}^m} |W_0([P_m \Theta_\tau [P_m \Theta]])| = \sup_{\boldsymbol{a} \in S_\tau} |W_0([P_m \Theta])| \le \sup_{\boldsymbol{a} \in \mathbb{R}^m} |W_0([P_m \Theta])| = \sup_{\boldsymbol{a} \in \mathbb{R}^m} |w_0(\boldsymbol{a})|, \tag{98}$$

where

$$S_\tau = \left\{ \boldsymbol{a}' : a'_l = \sum_{k=0}^{m-1} \left( \frac{1}{\sqrt{4\pi\tau}} \sum_{q=-\infty}^{\infty} \int_{-1/2}^{1/2} d\xi \int_{-1/2}^{1/2} d\xi' e^{-\frac{1}{4\tau}(\xi-\xi'-q)^2} \phi_l(\xi)\phi_k(\xi') \right) a_k, \ \boldsymbol{a} \in \mathbb{R}^m \right\}$$
$$\subseteq \mathbb{R}^m. \tag{99}$$

Therefore, we obtain

$$\|w(\tau)\|_{L^\infty(\mathbb{R}^m)} = \left\|e^{t\mathcal{L}_m} w_0\right\|_{L^\infty(\mathbb{R}^m)} \le \|w_0\|_{L^\infty(\mathbb{R}^m)} \le c, \ \forall m \in \mathbb{N}, \tag{100}$$

for the operator $\mathcal{L}_m = -\sum_{k,l=0}^{m-1} A_{kl} a_k \frac{\partial}{\partial a_l}$. From this result, the operator norm of $e^{t\mathcal{L}_m}$ is evaluated as

$$\|e^{t\mathcal{L}_m}\| = \sup_{w_0 \ne 0} \frac{\left\|e^{t\mathcal{L}_m} w_0\right\|_{L^\infty(\mathbb{R}^m)}}{\|w_0\|_{L^\infty(\mathbb{R}^m)}} \le 1. \tag{101}$$

This shows that $\exists M, \omega$ s.t. $\forall m \in \mathbb{N}, e^{t\mathcal{L}_m} \le M e^{\omega t}$ (e.g., $M = 1$ and $\omega = 0$), and thus the cylindrical approximation is stable in the sense of Def. C.29.

Therefore, the cylindrical approximation (97) is consistent and stable, and Thm. C.30 gives the convergence $W([P_m \Theta], \tau) \xrightarrow{m \to \infty} W([\Theta], \tau)$ on a compact subset in finite time interval $[0, T]$ if $W([P_m \Theta], 0) \xrightarrow{m \to \infty} W([\Theta], 0)$ and if Eq. (97) is well-posed in $[0, T]$.

---

[5]Strictly speaking, if $a_k$ are defined on an infinite interval, this assumption of boundedness is not valid for the FDEs used in our experiments because $\|w_0\|_{L^\infty(\mathbb{R}^m)} \to \infty$ as $a_k \to \infty$. However, the range of $a_k$ is usually set to a finite interval in numerical experiments, and thus $\|w_0\|_{L^\infty(\mathbb{R}^m)}$ is also finite; i.e., the assumption of boundedness holds.

### E.2.3 Initial Conditions

We use the Gaussian random field as the initial condition:

$$W_0([\Theta]) = -\overline{\mu} \int_{-1/2}^{1/2} d\xi \,\Theta(\xi) + \frac{1}{2} \int_{-1/2}^{1/2} d\xi \int_{-1/2}^{1/2} d\xi' C(\xi, \xi')\Theta(\xi)\Theta(\xi'), \quad (102)$$

where $\overline{\mu}$ is the mean velocity and $C(\xi, \xi')$ is the covariance matrix. This is equivalent to the initial distribution of the velocity field following the Gaussian distribution (102). We use the Fourier series as the orthonormal basis:

$$\phi_k(\xi) = \begin{cases} 1 & (k = 0) \\ \sqrt{2}\sin(\pi(k+1)\xi) & (k : \text{odd}) \\ \sqrt{2}\cos(\pi k\xi) & (k : \text{nonzero even}) \end{cases}. \quad (103)$$

Then, the analytic solution of Eq. (93) under the cylindrical approximation is given by

$$W([P_{2M}\Theta], \tau) =$$
$$-\overline{\mu}a_0 + \frac{1}{2}\sum_{k=0}^{M-1}\sum_{l=0}^{M-1}\left(e^{-4\pi^2(k^2+l^2)\tau}\tilde{C}_{2k,2l}a_{2k}a_{2l} + e^{-4\pi^2(k^2+(l+1)^2)\tau}\tilde{C}_{2k,2l+1}a_{2k}a_{2l+1}\right.$$
$$\left. + e^{-4\pi^2((k+1)^2+l^2)\tau}\tilde{C}_{2k+1,2l}a_{2k+1}a_{2l} + e^{-4\pi^2((k+1)^2+(l+1)^2)\tau}\tilde{C}_{2k+1,2l+1}a_{2k+1}a_{2l+1}\right),$$
$$(104)$$

where $m = 2M$ and

$$a_k := (\phi_k, \Theta)_{L_p^2([-1/2,1/2])}, \quad \tilde{C}_{ij} := \int_{-1/2}^{1/2} d\xi \int_{-1/2}^{1/2} d\xi' C(\xi, \xi')\phi_i(\xi)\phi_j(\xi'). \quad (105)$$

The derivation is lengthy and is given in App. D.2.2. Note that the higher degree terms decay exponentially in terms of $k$, $l$, and $\tau$, and the solution is dominated by $a_k$ with $k \lesssim 1$.

We use three types of covariance matrices:

$$C(\xi, \xi') = \sigma^2\delta(\xi - \xi'), \quad (106)$$

$$C(\xi, \xi') = \sigma^2\sum_{k=0}^{99} e^{-k/10}\phi_k(\xi)\sum_{l=0}^{99} e^{-l/10}\phi_l(\xi'), \quad (107)$$

$$C(\xi, \xi') = \sigma^2. \quad (108)$$

Substituting them into Eq. (102), we have three types of initial conditions: the delta, moderate, and constant initial condition, respectively. They are equivalent to

$$\tilde{C}_{ij} = \sigma^2 \text{ for all } i = j \geq 0 \text{ (0 otherwise)}, \quad (109)$$

$$\tilde{C}_{ij} = \sigma^2 e^{-i/10}e^{-j/10} \text{ for } i = j \leq 99 \text{ (0 otherwise)}, \quad (110)$$

$$\tilde{C}_{ij} = \sigma^2 \text{ for } i, j = 0 \text{ (0 otherwise)}. \quad (111)$$

Eq. (106) is nonsmooth and represents the extremely short-range correlation of the initial velocity field; the velocities at two points in an infinitesimally small neighborhood have no correlation. The spectrum of $C(\xi, \xi')$ (Eq. (109)) has an infinite tail. Eq. (108) represents the extremely long-range correlation of the initial velocity; the velocities at any two points have the same correlation $\sigma^2$. The spectrum of $C(\xi, \xi')$ (Eq. (111)) decays immediately. Eq. (107) represents a moderate-range correlation of the two above; the velocities at two points have a periodic correlation. The spectrum of $C(\xi, \xi')$ (Eq. (110)) decays exponentially. Theoretically, it is said that $C(\xi, \xi')$ that has a long tail of spectrum is hard to simulate numerically [91]. In our experiment with PINNs, however, the error of the solution is dominated by the optimization error of PINNs (Sec. 4) and strongly depends on the training setups.

# F Supplementary Discussion

## F.1 Limitations

**Higher spacetime dimensions.** In our experiments, the spacetime dimension is limited to $1 + 1$ ($t$ and $x$). Generalization to $1 + d$ dimensions is feasible, albeit with increased computational costs. For spaces with $d > 1$ dimensional spaces, multiple options exist for expanding $\theta$ [79]. Despite the inclusion of additional spacetime dimensions, the computational complexity of our model, up to computing functional derivatives, remains $\mathcal{O}(m^r)$, where $r$ is the order of the FDE. However, note that $m$ scales exponentially w.r.t. $d$, which is typically $1, 2, 3$, or $4$.

**Higher-order functional derivatives.** The order of functional derivatives in FDEs in our experiments is limited to $r = 1$; however, extending to $r \geq 2$ is straightforward. For example, the cylindrical approximation of the second-order functional derivative is expressed as $\delta^2 F([\theta], t)/\delta\theta(x)\delta\theta(y) \approx \Sigma_{k,l=0}^{m-1} \partial^2 f(\boldsymbol{a}, t)/\partial a_k \partial a_l \phi_k(x)\phi_l(y)$, which can be computed via backpropagation twice.

**To further reduce errors.** The relative errors obtained in our experiments are $\sim 10^{-3}$, which might be unsatisfactory in some applications This is a general problem inherent in PINNs, for which PINNs are sometimes criticized. To further reduce the errors, one can use recently developed techniques for solving high-dimensional PINNs [106, 44, 96, 45, 21, 103, 33]. These methods can be easily equipped with our model; one of the strengths of our model is that it can be integrated with arbitrary techniques developed for PINNs. See also App. F.7 for a better sampling of collocation points.

**Challenges toward even higher degrees.** Instability of the numerical integration for computing $a_k$ given a function $\theta(x)$, i.e., $a_k = \int \theta(x)\phi_k(x)dx$, is observed for large $k$s. This instability comes from the intense oscillation of $\phi_k$ for large $k \gtrsim 500$, as we confirmed in our preliminary experiments. This is a general problem in numerical computation, not exclusive to our model. Nevertheless, such higher degrees are unnecessary to approximate smooth functions. Extremely large degrees are required only when one wants to include nonsmooth or divergent functions in the domain of functionals $D(F)$. Training on such functions with the cylindrical approximation is an open problem Dealing with highly oscillatory functions is also a central research interest in numerical methods for non-smooth dynamical systems [1].

**Inclusion and diversity of functions** What class of functions can the cylindrical approximation represent? The equi-small tail condition (Thm. C.9) characterizes the domain of functions $D(F)$ to which the cylindrical approximation can be applied. In most of the convergence theorems in App. C, such as Thm. C.19, the compactness of $K \subset H$ is assumed, which can be ensured if Thm. C.9 holds. In other words, the equi-small tail condition is part of a sufficient condition for the convergence of the cylindrical approximation. Surprisingly or not, step functions and ReLU satisfy the equi-small tail condition. In contrast, functions with divergent norms ($|\theta|_H = \infty$), which are not typically of interest, cannot be handled in the cylindrical approximation.

**Inclusion and diversity of functionals** We assume functional differentiability in this paper. Central theorems of differentiability are given in App. C.3, where functional Lipschitzness is assumed. An example of non-Lipschitz functionals is $F([\theta]) = \int dx\sqrt{|\theta(x)|}$ with $\theta(0) = 0$.

**Inclusion and diversity of FDEs** In this paper, we consider the abstract evolution equation, a crucial class of linear FDEs. It does not include, for instance, Eq. (13) in quantum field theory or Eq. (14) in functional renormalization group theory. Nevertheless, the cylindrical approximation *is* applicable to these equations, although the theoretical (non-)convergence of solutions is currently unknown. Finally, note that the approximated functional derivatives converge to the non-approximated ones independently of the specific FDEs.

## F.2 Computational Complexity of CP-ALS

In Sec. 1, we mentioned the computational complexity of the state-of-the-art method (CP-ALS) ($\mathcal{O}(m^6 T)$) We derived it from Eqs. (567) and (568) in [91] in Sec. 7.2.2 in [91]. These equations

require $\mathcal{O}(m^5)$ including the summation over the indices $e\ (= 0, \dots, m^2)$ and $z\ (= 0, \dots, m^2)$, and the Hadamard matrix product over the index $k\ (= 1, \dots, m)$. They are computed for each $q$ $(= 1, \dots m)$ and timestep $t\ (= 1, \dots, T)$, leading to $\mathcal{O}(m^5 \times m \times T) = \mathcal{O}(m^6 T)$ in total. Note that Eqs. (567) and (568) in the published version [91] correspond to Eqs. (568) and (569) in the arXiv version (`arXiv:1604.05250v3`).

## F.3 Choice of Bases

We use the Fourier series for periodic systems and the Legendre polynomials for non-periodic systems. Both are common basis functions in numerical computation. Legendre polynomials are part of the Jacobi polynomials family, a typical class of orthogonal bases. Many Jacobi polynomials are numerically unstable and require careful treatment; for example, the Hermite polynomials change scales factorially with degree $(1/\sqrt{m!})$ and often result in `nan`. The Laguerre polynomials have similar issues. The Chebyshev polynomials of the first kind and the Legendre polynomials are suitable for numerical computation; they are bounded, defined on a finite interval, and their weight functions are regular (note that the weight function of the Chebyshev polynomials of the first kind, $1/\sqrt{1-x^2}$, can be easily regularized by changing variables). Moreover, the Chebyshev and Legendre polynomials are known to minimize the $L^\infty$ and $L^2$ approximation errors, respectively [41]. Nevertheless, we did not observe a significant performance difference between them in curve-fitting experiments, and we use the Legendre polynomials in our experiments.

**How to choose basis functions?** The choice depends on the FDE, boundary conditions, symmetry, function spaces, and numerical stability. The Fourier series is suitable for periodic functions, but for non-periodic functions, it exhibits the Gibbs phenomenon, causing large numerical errors. Therefore, Legendre polynomials are a good choice for non-periodic functions. The choice of bases is a common concern in numerical analysis, such as in the finite element and spectral methods.

**Generalization to other bases: Riesz basis** Our approach can be generalized to general non-orthonormal bases, such as Riesz bases [37, 76, 14, 30, 52, 51, 15], where the orthonormality condition $(\phi_k, \phi_l)_H = \delta_{kl}$ is replaced with $(\phi_k, \phi_l)_H = g_{kl}$. Here, $g_{kl}$ is a certain matrix, or a *metric*, whose choice significantly affects the computational costs of the approximated FDEs. Sparse metrics are preferred for efficient computation. See also App. F.4.

## F.4 Computational Complexity Revisited

We showed the computational complexity of our model *up to the computation of functional derivatives* is given by $\mathcal{O}(m^r)$. This does not include the computational complexity of the loss function of PINNs. It could be $> \mathcal{O}(m^r)$, strongly depending on the form of the FDE and/or the choice of the orthonormal basis. For concreteness, let us consider Eq. (97) (BHE). While the BHE is first-order, the computational complexity of the loss function can be $\mathcal{O}(m^2)$ if matrix $A_{kl}$ in Eq. (97) is not diagonal. This is the case if we use the Legendre polynomials instead of the Fourier series.

On the other hand, the choice of the basis function can improve computational complexity. For example, the computational complexity of the second-order FDE can reduce from $\mathcal{O}(m^2)$ to $\mathcal{O}(m^1)$ when the approximated FDE includes only the diagonal elements of $\{\partial^2 f(\boldsymbol{a}, t)/\partial a_k \partial a_l\}_{k,l=0}^{m-1}$. Again, whether it is the case or not depends on the form of the FDE and/or the choice of the orthonormal basis.

## F.5 Comparison with DFT

In first-principles computations of density functional theory (DFT), an NN-based approach that utilizes finite element methods for spacetime grid approximation is commonly employed. For example, an NN, $\hat{F}(\{y_j := f(\mathbf{r}_j)\}_j)$, approximating a target functional $F([f])$ by evaluating $f$ at specific grid points $\{\mathbf{r}_j\}_j$. Functional derivatives at each grid point can be computed using automatic differentiation: $\{\frac{\delta F([f])}{\delta f(\boldsymbol{r})}\}_{\boldsymbol{r}} \doteq \{\frac{\partial \hat{F}(\{y_j\}_j)}{\partial y_i}\}_i$. However, the central focus of this area does not include solving PDEs, let alone FDEs. Moreover, this method requires the discretization of spacetime, leading to numerical error of derivatives, while in our approach, such discretization is not necessary, and spacetime differentiation can be performed analytically if the basis functions are analytic.

The finite element methods for spacetime and the cylindrical approximation (basis function expansion) are similar in the sense that they replace spacetime degrees of freedom with other quantities: spacetime grid points and expansion coefficients, respectively. Even when their degrees of freedom for a given problem are comparable, the latter (our approach) is suitable for solving FDEs.

## F.6 FDE Has Multiple Meanings

The term "Functional Differential Equation" (FDE) varies in meaning across research areas. In mathematics, an FDE typically refers to a differential equation with a deviating argument, e.g., $f'(x) = xf(x + 3)$. In physics, it denotes a PDE involving functional derivatives, such as the abstract evolution equation described in the main text. Additionally, FDE can imply a functional equation involving functional derivatives, e.g., $\frac{\delta F([\theta])}{\delta \theta(x)} = A(x)F([\theta])$, where $A(x)$ is a given function. Research on the last interpretation is limited, and the uniqueness, existence, and practical applications remain ambiguous.

## F.7 Radial Gaussian Sampling

The region $\theta \sim 0$, or $a_k \sim 0_k$ is the central focus when solving the BHE. Thus, sampling as many collocation points from $\theta \sim 0$ as possible is of crucial importance. To make the training set free from the curse of dimensionality, we can use the following sampling method based on the polar coordinates:

1. Sample a unit vector $\mathbf{v} \in \mathbb{R}^d$, where $d$ is the dimension of the input space.
2. Sample a real number (radius) $r > 0$ from a Gaussian distribution $\mathcal{N}(0, \sigma^2)$ (or a truncated Gaussian distribution).
3. Obtain a training collocation point $r\mathbf{v} \in \mathbb{R}^d$.

The collocation points thus obtained obviously concentrate around $\theta \sim 0$ more than the Latin hypercube sampling, which we used in our experiment.

## F.8 Curvilinear Patterns in Fig. 4 and Noisy Patterns in Fig. 7 and

The white curves in Fig. 4 represent locations where the predictions and the exact solutions happen to coincide. The noisy patterns that occur for $a_k$, $k \geq 1$ in Fig. 7 are because the solution is almost independent of $a_k$ with $k \gtrsim 1$. Thus, optimizing the model in the directions of $a_k$ with $k \gtrsim 1$ have only a negligible effect on minimizing the loss function, keeping the random predictions of the model. If we delve deeper, they might be related to the loss landscapes of PINNs [8, 35], which is also an interesting research topic.

## F.9 Second-order Functional Derivatives

**Fig. 9** shows the estimated second-order derivative of the FTE's solution at $\theta = 0$ and $t = 0$ under the linear initial condition. The error is as small as $\sim 10^{-3}$, despite that $\theta \sim 0$ is not included in the training set due to the curse of dimensionality and that the second-order information is not included in the loss function.

However, such small errors only emerge when the analytic solution is so simple that, e.g., it includes only one coefficient: $f(\mathbf{a}, t) = \rho_0 v_0 (a_1 - v_0 t)$. The errors of the second-order derivatives obtained from other conditions and equations were significant ($\gtrsim 10^3$ in some cases). We could reduce the error by (i) using another sampling method for training sets that does not suffer from the curse of dimensionality (an example is given in App. F.7) and (ii) adding the second-order derivative to the loss function (e.g., differentiate both sides of the BHE w.r.t. $a_k$ and use it to the residual loss for PINNs).

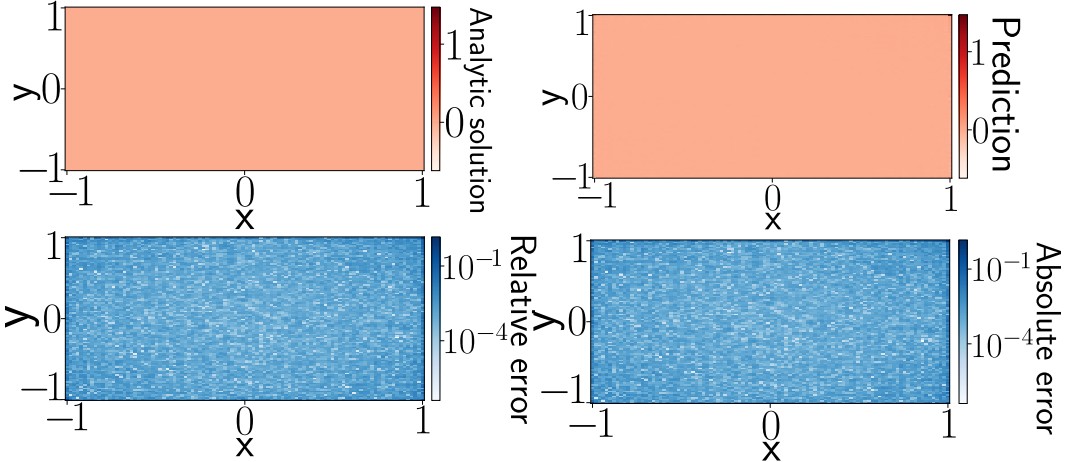

Figure 9: **Second-order derivative at** $\theta = 0$ **&** $t = 1$. The model is trained on the FTE with degree 100 under the linear initial condition. The errors are as small as $\sim 10^{-3}$.

# G Detailed Experimental Settings

We describe detailed setups of our experiment, including those on the advection-reaction functional equation (ARE) (App. H.4). The experimental settings for the ARE align with those provided in [91].

**Datasets.** The training, validation, and test sets are sampled with Latin hypercube sampling [64, 39] with range $[0, 1], [-1, 1], [0, 0.001], [-0.1, 0.1], [0, 0.5]$, and $[h - 0.01, h + 0.01]$ for $t$ (FTE), $a_k$ (FTE), $t$ (BHE), $a_k$ (BHE), $t$ (ARE), and $a_k$ (ARE), respectively, where $h := 0.698835274542439$. Training mini-batches are randomly sampled every time a mini-batch is created. For the BHE, we decay the sampling range quadratically in terms of $k \in \{0, 1, \ldots, m\}$. This sampling stabilizes the training because most of the loss comes from the region where $k$ and $a_k$ are large, which can be seen from the definition of the BHE. That is, all the terms on the right-hand side are proportional to one of $a_k$s, while $a_k$s for large $k$s are negligible in the analytic solution. Therefore, the collocation points with large $a_k$s with large $k$s can be seen as noise in training. Note that this decaying sampling does not limit the quality of the solution at all because the solution is the characteristic function and we are interested only in $\theta(x) \approx 0$, e.g., $\frac{\delta W([\theta], t)}{\delta \theta(x)}|_{\theta=0} = \langle u(x) \rangle$ (see also App. B.2).

**Miscellaneous settings.** We use a 4-layer PINN with $3\times$ (linear + sin activation + layer normalization) + last linear layer. Unless otherwise noted, the widths are 1024 for the ARE and FTE and 2048 for the BHE. The batch size is 1024. The activation function is the sin function. The loss function for the FTE with the nonlinear initial condition (main text) and the ARE is the sum of the $L^1$ and $L^\infty$ losses. The loss function for the others is the smooth $L^1$ loss. The optimizer is AdamW [59]. The learning rate scheduler is the linear warmup with cosine annealing with warmup [58], which is defined as the following `scheduler`:

```
scheduler1 = torch.optim.lr_scheduler.LinearLR(
    optimizer, start_factor=start_factor, total_iters=milestone)
scheduler2 = torch.optim.lr_scheduler.CosineAnnealingWarmRestarts(
    optimizer, T_0=T_0, T_mult=T_mult, eta_min=eta_min)
scheduler = torch.optim.lr_scheduler.SequentialLR(
    optimizer, schedulers=[scheduler1, scheduler2], milestones=[milestone])
```

See PyTorch official documentation [70] for the detailed definitions of the parameters. `eta_min` is set to 0. `start_factor` is set to $10^{-10}$. $\rho_0 = 1$ and $v_0 = 1$ are used for the FTE. $\bar{\mu} = 8$ and $\sigma^2 = 10$ are used for the BHE. The Legendre polynomials and real Fourier series are used for the FTE and BHE, respectively, as the orthonormal basis. For the ARE, the modified Chebyshev polynomials described in [91] are used.

**Hyperparameter tuning.** The full search space is given Tab. 3. Unless otherwise noted, the number of iterations is 300,000 and 500,000 for the BHE and FTE, respectively. The number of trials for hyperparameter tuning is 50. Optuna [2] is used for hyperparameter tuning with the TPE sampler and median pruner. The number of tuning trials is 50 for all conditions, and the best hyperparameters within the 50 trials are used. See the config file in our repository for more details.

Table 3: **Search space of hyperparameters.** Learning rates and weight decays are sampled log-uniformly. $N = 300{,}000$ and $500{,}000$ for the BHE and FTE, respectively. `T_0`, `T_mult`, and `milestone` are used for the aforementioned `scheduler`. For the ARE and the main text results of FTE with the nonlinear initial condition, weight decay is fixed to 0 and the search space of learning rates is from $10^{-7}$ to $10^{-4}$.

| HYPERPARAMETERS | SEARCH SPACE |
|---|---|
| LEARNING RATE | FROM $10^{-6}$ TO $10^{-3}$ |
| WEIGHT DECAY | FROM $10^{-7}$ TO $10^{-4}$ (OTHERS) |
| T_0 | $N/5$, $N/2$, OR $N$ |
| T_MULT | 1 OR 2 |
| MILESTONE | 0, $N/10$, OR $N/100$ |

**Remaining hyperparameters.** We show the hyperparameters used in our experiments in Tabs. 4–10

Table 4: **Hyperparameters: FTE with linear initial condition.**

| Hyperparameters | Degree | | |
|---|---|---|---|
| | 4 | 20 | 100 |
| Learning rate | $8.675 \times 10^{-6}$ | $1.878 \times 10^{-5}$ | $2.839 \times 10^{-5}$ |
| Weight decay | $4.534 \times 10^{-6}$ | $6.184 \times 10^{-7}$ | $2.008 \times 10^{-5}$ |
| T_0 | 250000 | 500000 | 500000 |
| T_MULT | 2 | 1 | 1 |
| MILESTONE | 0 | 5000 | 0 |

Table 5: **Hyperparameters: FTE with nonlinear initial condition (main text).**

| Hyperparameters | Degree | | |
|---|---|---|---|
| | 4 | 100 | 1000 |
| Learning rate | $8.456 \times 10^{-6}$ | $1.042 \times 10^{-5}$ | $1.130 \times 10^{-5}$ |
| Weight decay | 0 | 0 | 0 |
| T_0 | 500000 | 500000 | 250000 |
| T_MULT | 1 | 1 | 1 |
| MILESTONE | 5000 | 0 | 5000 |

Table 6: **Hyperparameters: FTE with nonlinear initial condition (Appendix).**

| Hyperparameters | Degree | | |
|---|---|---|---|
| | 4 | 10 | 20 |
| Learning rate | $4.837 \times 10^{-6}$ | $5.810 \times 10^{-5}$ | $6.076 \times 10^{-6}$ |
| Weight decay | $8.637 \times 10^{-5}$ | $2.877 \times 10^{-5}$ | $2.068 \times 10^{-7}$ |
| T_0 | 250000 | 500000 | 500000 |
| T_MULT | 2 | 2 | 2 |
| MILESTONE | 0 | 50000 | 50000 |

Table 7: **Hyperparameters: BHE with delta initial condition.**

| | DEGREE | | |
|---|---|---|---|
| HYPERPARAMETERS | 4 | 20 | 100 |
| LEARNING RATE | $6.680 \times 10^{-5}$ | $1.372 \times 10^{-4}$ | $1.204 \times 10^{-4}$ |
| WEIGHT DECAY | $4.429 \times 10^{-6}$ | $6.644 \times 10^{-7}$ | $4.893 \times 10^{-7}$ |
| T_0 | 300000 | 300000 | 150000 |
| T_MULT | 2 | 1 | 1 |
| MILESTONE | 0 | 0 | 3000 |

Table 8: **Hyperparameters: BHE with constant initial condition.**

| | DEGREE | | |
|---|---|---|---|
| HYPERPARAMETERS | 4 | 20 | 100 |
| LEARNING RATE | $5.131 \times 10^{-5}$ | $9.989 \times 10^{-5}$ | $8.637 \times 10^{-5}$ |
| WEIGHT DECAY | $1.083 \times 10^{-5}$ | $9.353 \times 10^{-7}$ | $8.378 \times 10^{-5}$ |
| T_0 | 300000 | 150000 | 300000 |
| T_MULT | 2 | 1 | 1 |
| MILESTONE | 3000 | 3000 | 30000 |

Table 9: **Hyperparameters: BHE with moderate initial condition.**

| | DEGREE | | |
|---|---|---|---|
| HYPERPARAMETERS | 4 | 20 | 100 |
| LEARNING RATE | $2.248 \times 10^{-5}$ | $3.254 \times 10^{-5}$ | $1.209 \times 10^{-5}$ |
| WEIGHT DECAY | $3.669 \times 10^{-5}$ | $1.771 \times 10^{-6}$ | $1.423 \times 10^{-6}$ |
| T_0 | 300000 | 300000 | 300000 |
| T_MULT | 1 | 2 | 1 |
| MILESTONE | 3000 | 0 | 0 |

Table 10: **Hyperparameters: ARE.**

| | DEGREE |
|---|---|
| HYPERPARAMETERS | 6 |
| LEARNING RATE | $6.3105 \times 10^{-5}$ |
| WEIGHT DECAY | 0 |
| T_0 | 250000 |
| T_MULT | 1 |
| MILESTONE | 5000 |

**How to choose weight parameters for each training.** We use validation relative error, i.e., the relative error between the prediction and the analytic solution, as the measure for choosing the best weight parameter during each training trial.

**Loss reweighting.** The softmax reweighting is used to adaptively balance the residual and boundary loss functions. See the code below and our official code at GitHub (`https://github.com/TaikiMiyagawa/FunctionalPINN`), where `list_tensors` is [residual loss, boundary loss] of a mini-batch.

```python
@torch.no_grad()
def softmax_coeffs(
    list_tensors: List[Tensor],
    temperature: float = 0.25) -> Tensor:
    """
    # Args
    - list_tensors: List of scalar loss Tensors.
    - temperature: Temperature parameter. Default is 0.25

    # Returns
    - coeffs: Tensor with shape [len(list_tensors),].
    """
    ts = torch.tensor(list_tensors)
    ts /= torch.max(ts) + EPSILON  # generates scale-invariant weight
    coeffs = torch.softmax(ts / temperature, dim=0)
    return coeffs
```

This function generates the loss reweighting factors `coeffs` = $[\lambda_1, \lambda_2]$, where total loss is defined as $\lambda_1 \times$ residual loss $+ \lambda_2 \times$ boundary loss.

**Float vs. double.** No significant differences in performance measures were observed throughout our training. Accordingly, we used float32 to reduce computational time.

**Libraries and GPUs.** All the experiments are performed on Python 3.11.8 [89], PyTorch 2.2.0 [70], Numpy 1.26.0 [38], and Optuna 3.5.0 [2]. An NVIDIA A100 GPU is used.

**Runtime.** The training process takes 2–3 hours, but this duration can be significantly improved because it largely depends on the implementation of components such as data loaders. See App. H.2 for more details on runtime. GPU memory consumption is about 400 MBs for the ARE and 800–1500 MBs for the FTE and BHE. We note that previous approaches to numerically solving FDEs rely on CPU processing, where performance varies significantly with the level of parallelization. Consequently, directly comparing the computational speed of our GPU-based approach with these methods is inherently challenging.

**Figure 2.** $F([\theta])$ is approximated by $F([P_{1000}\theta])$ for convenience.

# H    Additional Experimental Results

## H.1    $P_m\theta$ Converges to $\theta$

The convergence rate of functional derivatives in Thm. 3.1 is given by $\|\theta - P_m\theta\|$. The convergence of $\|\theta - P_m\theta\|$ is visualized in Fig. 10 for four functions including non-smooth functions.

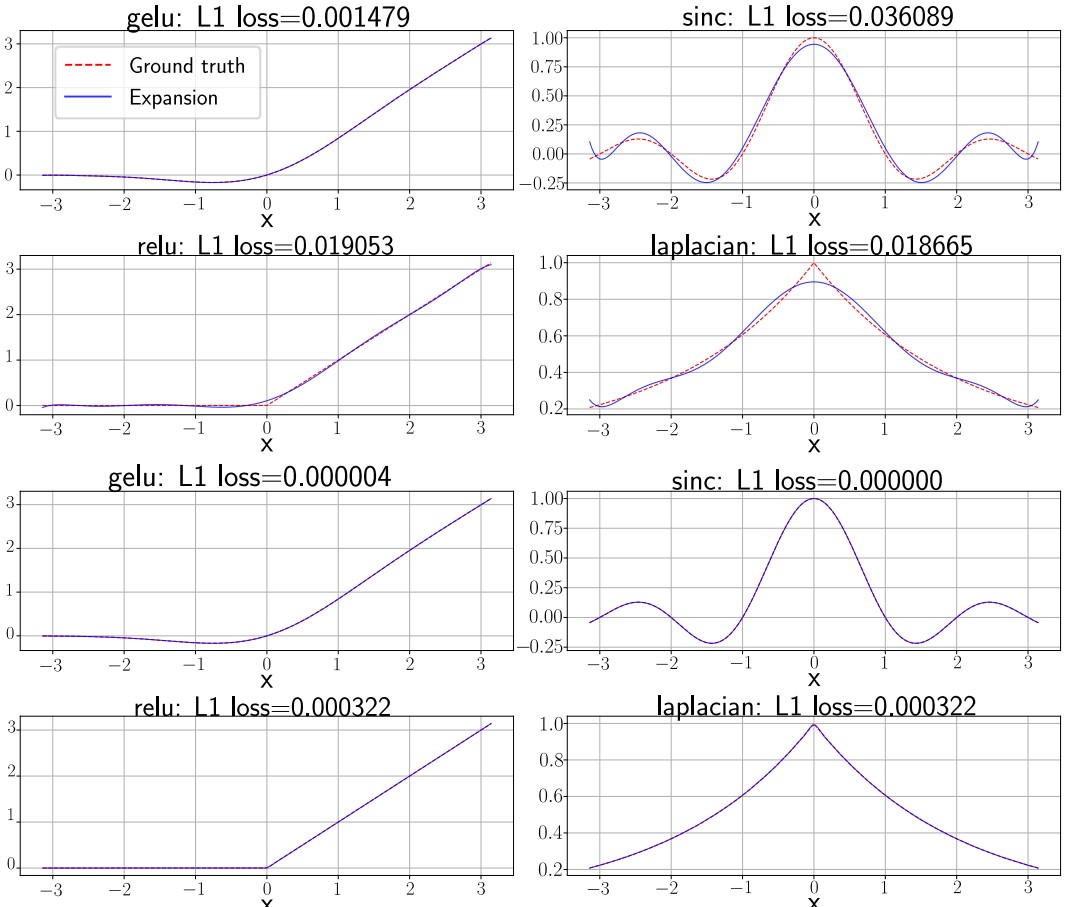

Figure 10: **Cylindrical approximation of** $\theta(x)$ **(red) by** $P_m\theta(x)$ **(blue) with degree 10 (top four panels) and 100 (bottom four panels).** $L^1$ absolute error $\frac{1}{N}\sum_{i=1}^{N}|\theta(x_i) - P_m\theta(x_i)|$ decreases as $m$ increases. $N = 10^5$. $x_i \in [-\pi, \pi]$ are linearly spaced. GeLU [40], sinc, ReLU, and Laplacian functions [78] are used. The basis is the Legendre polynomials.

## H.2    Runtime

Tab. 11 shows training runtime. The runtime remains nearly constant and does not significantly vary from $m \sim 1$ to $m \sim 100$. This aspect is of practical significance as it indicates that the computational time of our model remains stable across a broad range of $m$.

This consistency in runtime may be attributed to the efficiency of the PyTorch library, where operations such as matrix multiplication and automatic differentiation are highly optimized. On the other hand, GPU memory consumption does show variation, ranging from 800 to 1500 MBs as stated App. G, as is naturally expected.

Finally, the previous approaches to numerically solving FDEs rely on CPU processing, where performance significantly varies with the level of parallelization; thus, a direct comparison of computational speed with our GPU-based approach is inherently challenging.

Table 11: **Degree vs. runtime.** "x:yz" denotes "x hours yz minutes." A "RUN" refers to a training process with 500k, 300k, and 500k iterations for the FTE, BHE, and ARE, respectively.

| FTE WITH 500K ITERATIONS | | | |
|---|---|---|---|
| DEGREE | 4 | 20 | 100 |
| #RUNS | 11 | 8 | 3 |
| RUNTIME | $2:18 \pm 0:04$ | $2:09 \pm 0:10$ | $2:33 \pm 0:02$ |

| BHE WITH 300K ITERATIONS | | | |
|---|---|---|---|
| DEGREE | 4 | 20 | 100 |
| #RUNS | 10 | 9 | 8 |
| RUNTIME | $2:36 \pm 0:02$ | $2:29 \pm 0:06$ | $2:28 \pm 0:00$ |

| ARE WITH 500K ITERATIONS | | | |
|---|---|---|---|
| DEGREE | 6 | | |
| #RUNS | 10 | | |
| RUNTIME | $3:05 \pm 0:04$ | | |

### H.3 Latin Hypercube Sampling and Curse of Dimensionality

We show the histograms of the $L^2$ norm of collocation points in the test sets. The sampling method is the Latin hypercube sampling, as mentioned in Sec. 4. Apparently, the collocation points s.t. norms $\sim 0$ are not sampled, which would cause the curse of dimensionality in training. In other words, the Latin hypercube sampling cannot completely address the curse of dimensionality in sampling.

**Why does extrapolation mean (Sec. 4)?** Firstly, Figs. 11–16 show that no collocation points such that $\|\boldsymbol{a}\| \approx 0$ were included in the training sets. Secondly, Figs. 4 & 7 plot the errors of PINN predictions at the collocation points where $\boldsymbol{a} = (0, 0, \ldots, 0, a_k, 0, \ldots, 0)$ with $k = 0, 1, 2, 19$, or 99. Therefore, the collocation points such that $a_k \approx 0$ in Figs. 4 & 7 were not included in the training sets, but the errors were as small as the region where $a_k \not\approx 0$. We refer to this as "extrapolation."

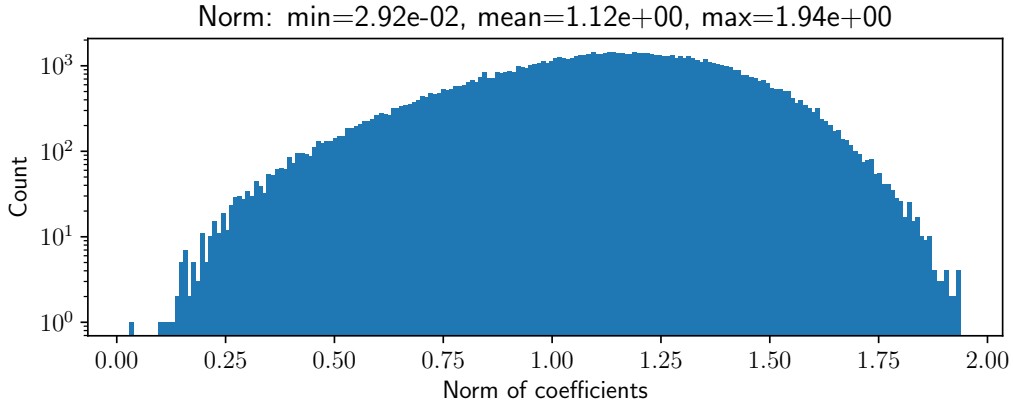

Figure 11: **Histogram of $L^2$ norms of 4-dimensional coefficients in test set for the FTE.**

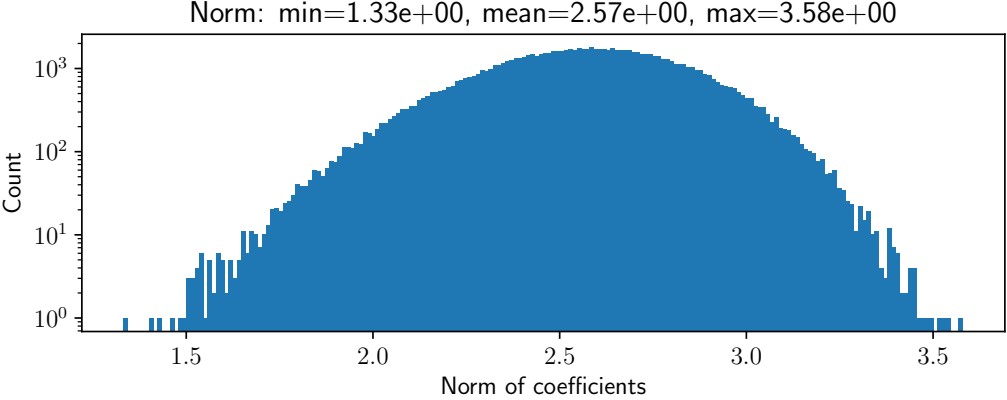

Figure 12: **Histogram of $L^2$ norms of 20-dimensional coefficients in test set for the FTE.**

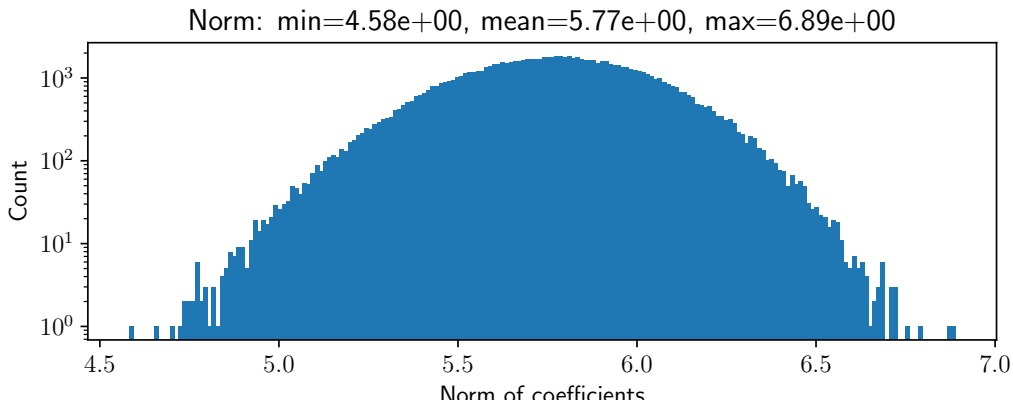

Figure 13: **Histogram of $L^2$ norms of 100-dimensional coefficients in test set for the FTE.**

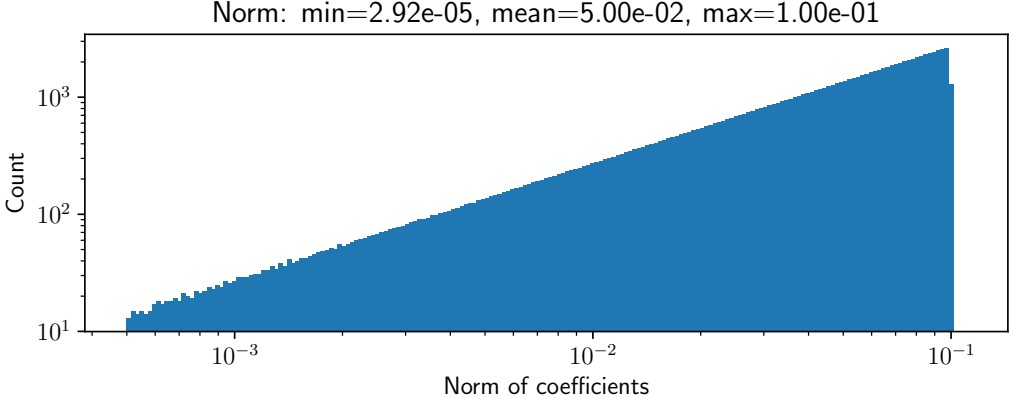

Figure 14: **Histogram of $L^2$ norms of 4-dimensional coefficients in test set for the BHE.**

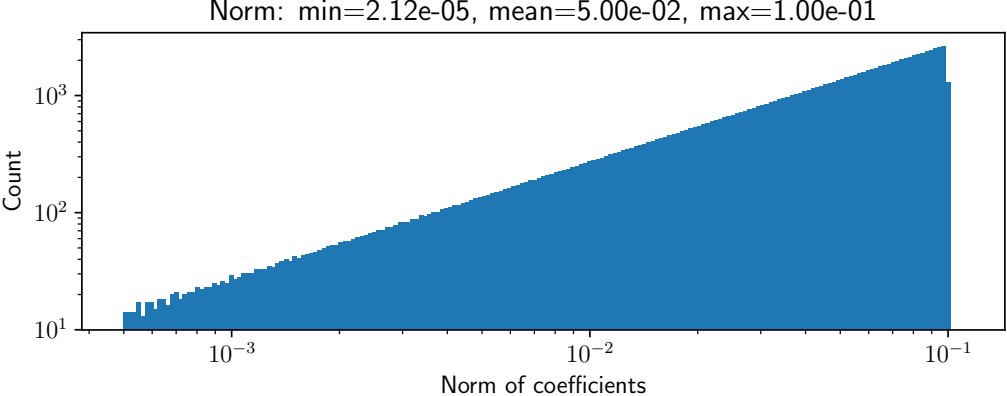

Figure 15: **Histogram of $L^2$ norms of 20-dimensional coefficients in test set for the BHE.**

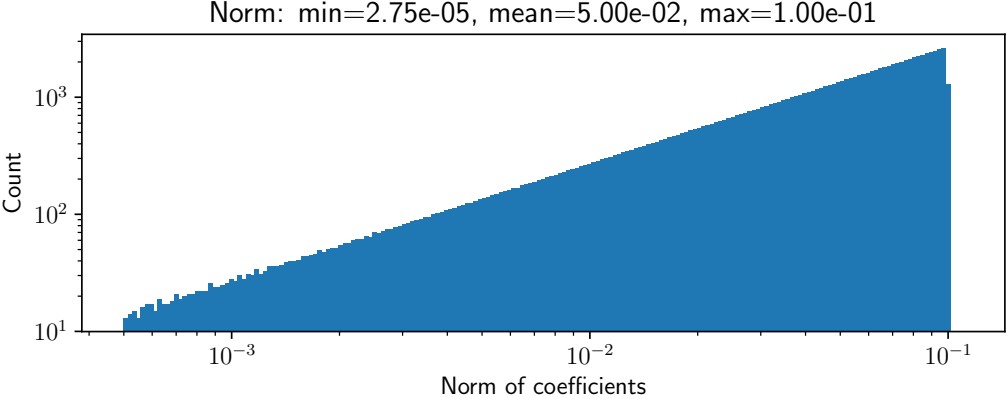

Figure 16: **Histogram of $L^2$ norms of 100-dimensional coefficients in test set for the BHE.**

## H.4 Comparison with Baseline: Advection-reaction Functional Equation (ARE)

We consider the advection-reaction functional equation (ARE).

$$\frac{\partial F([\theta], t)}{\partial t} = -\int_0^{2\pi} dx \theta(x) \frac{\partial}{\partial x} \left( \frac{\delta F([\theta], t)}{\delta \theta(x)} \right) \tag{112}$$

We follow the experimental setting described in [91].

In [91], the reported time-dependent relative error of the CP-ALS and HT fluctuates from $\sim 10^{-8}$ to $\sim 10^{-1}$, and the temporally-averaged relative error is estimated to be $\lesssim 10^{-2}$ (the bottom two panels in Fig. 38 in [91]), where we roughly approximated the time-dependent error curves as an exponential function of the form $y = 10^{-6+10t}$.

Our preliminary results are shown in Tab. 12. We observe that the learned functional is almost constant in time, a failure mode of PINNs, causing large errors. We anticipate the result can improve with further hyperparameter optimization [50].

Table 12: **Relative and absolute errors.** The models are trained on the ARE. The error bars are the standard deviation over 10 training runs with different random seeds. Relative and absolute error represent the averaged relative and absolute error over all collocation points and all runs. Best relative and absolute error represent the relative and absolute error of the best collocation point averaged over all runs. The worst relative and absolute errors represent the averaged relative and absolute errors of the worst collocation point averaged over all runs.

| DEGREE | RELATIVE ERROR | ABSOLUTE ERROR |
|---|---|---|
| 6 | $(0.954940 \pm 2.86337) \times 10^{-1}$ | $(6.17040 \pm 1.83985) \times 10^{-2}$ |

| DEGREE | BEST RELATIVE ERROR | BEST ABSOLUTE ERROR |
|---|---|---|
| 6 | $(0.12016 \pm 2.49220) \times 10^{-1}$ | $(0.58046 \pm 1.73938) \times 10^{-2}$ |

| DEGREE | WORST RELATIVE ERROR | WORST ABSOLUTE ERROR |
|---|---|---|
| 6 | $(9.29019 \pm 2.85457) \times 10^{-1}$ | $(8.49163 \pm 2.0718) \times 10^{-2}$ |

## H.5 Relative and Absolute Errors When $W([0], t) = 0$ Is Included In Loss

We show the histograms of the relative and absolute errors for the models trained on the BHE with the loss term corresponding to the identity $W([0], t) = 0$. They are almost the same as those for the models trained without $W([0], t) = 0$, but the errors of the first-order derivative are $10^{-1}$ times smaller, as shown in Sec. 4.

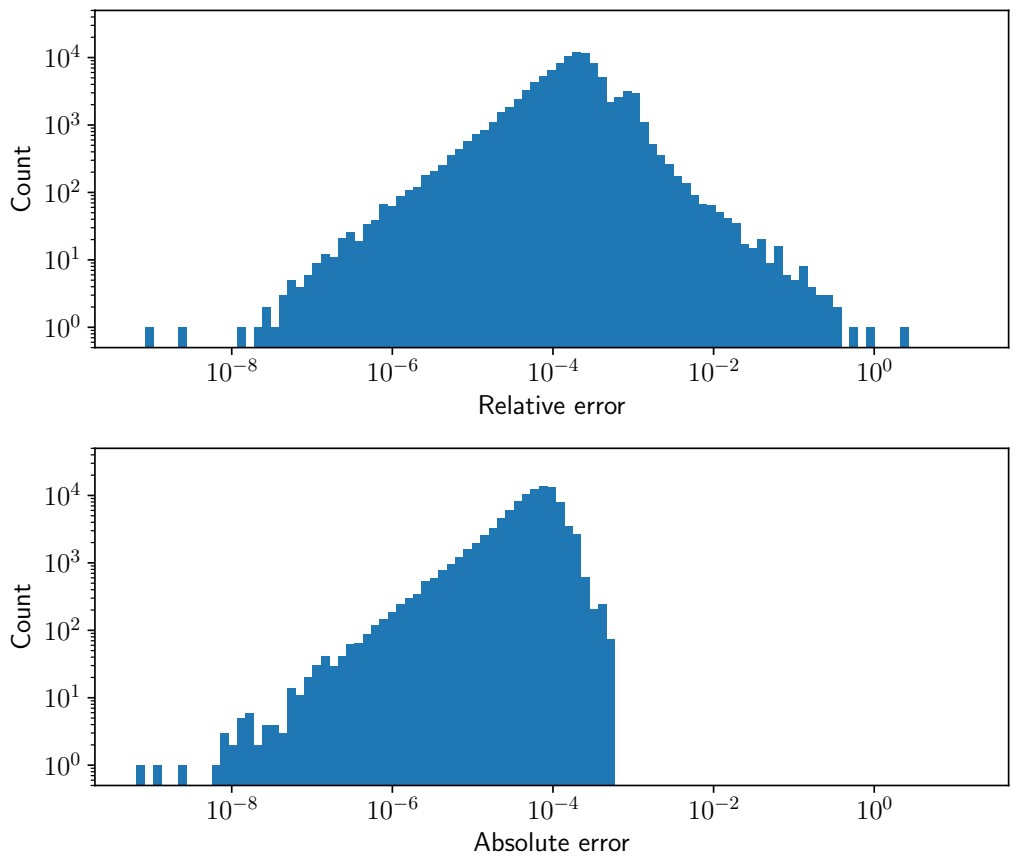

Figure 17: **Histogram of relative and absolute error.** The model is trained on the BHE of degree 4 with the delta initial condition. A single random seed is used for the training.

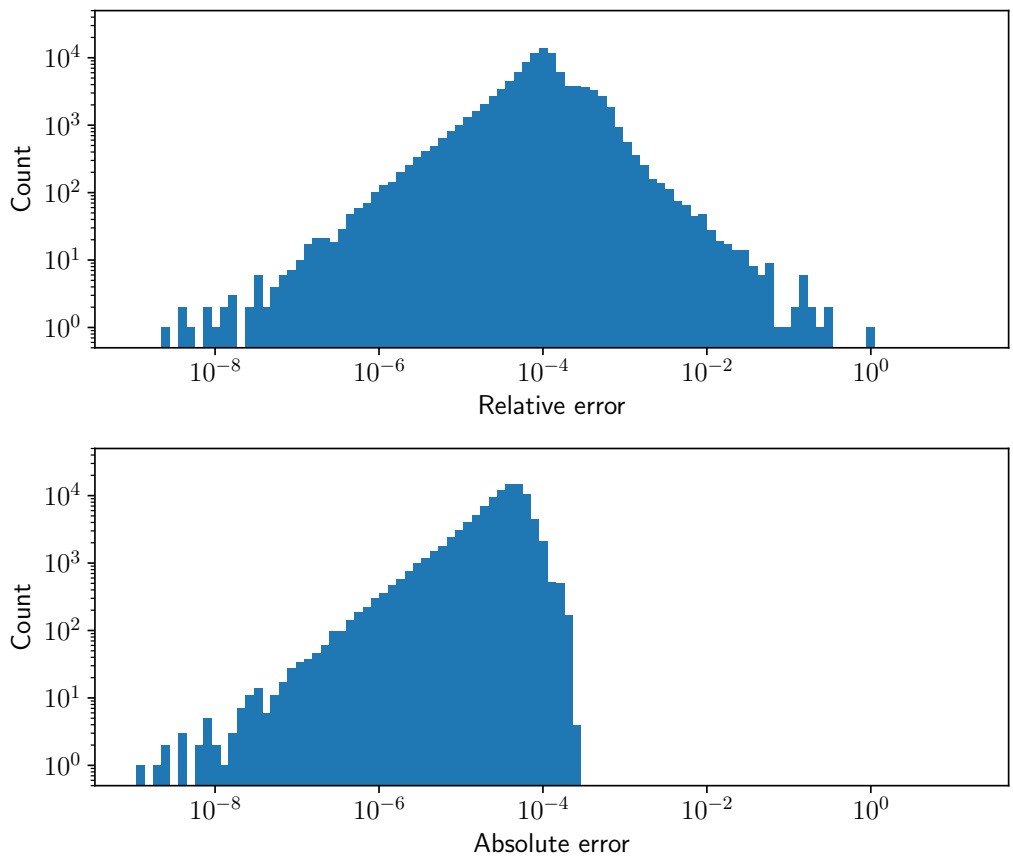

Figure 18: **Histogram of relative and absolute error.** The model is trained on the BHE of degree 100 with the delta initial condition. A single random seed is used for the training.

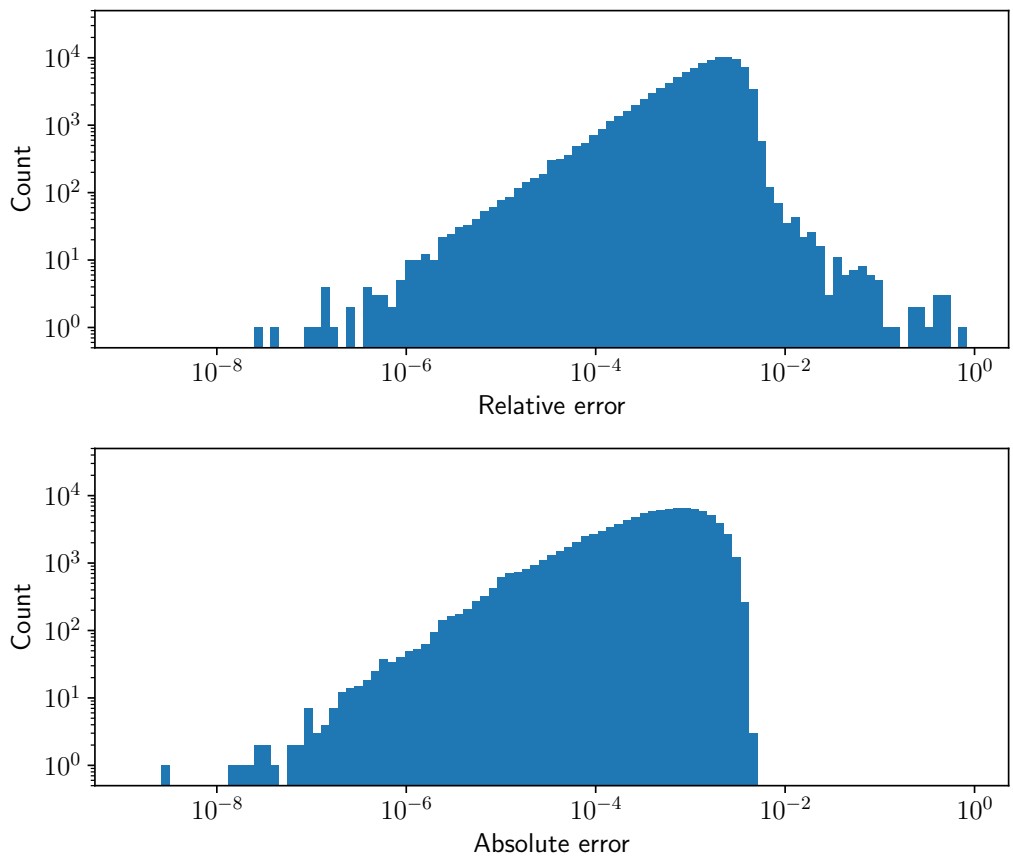

Figure 19: **Histogram of relative and absolute error.** The model is trained on the BHE of degree 4 with the constant initial condition. A single random seed is used for the training.

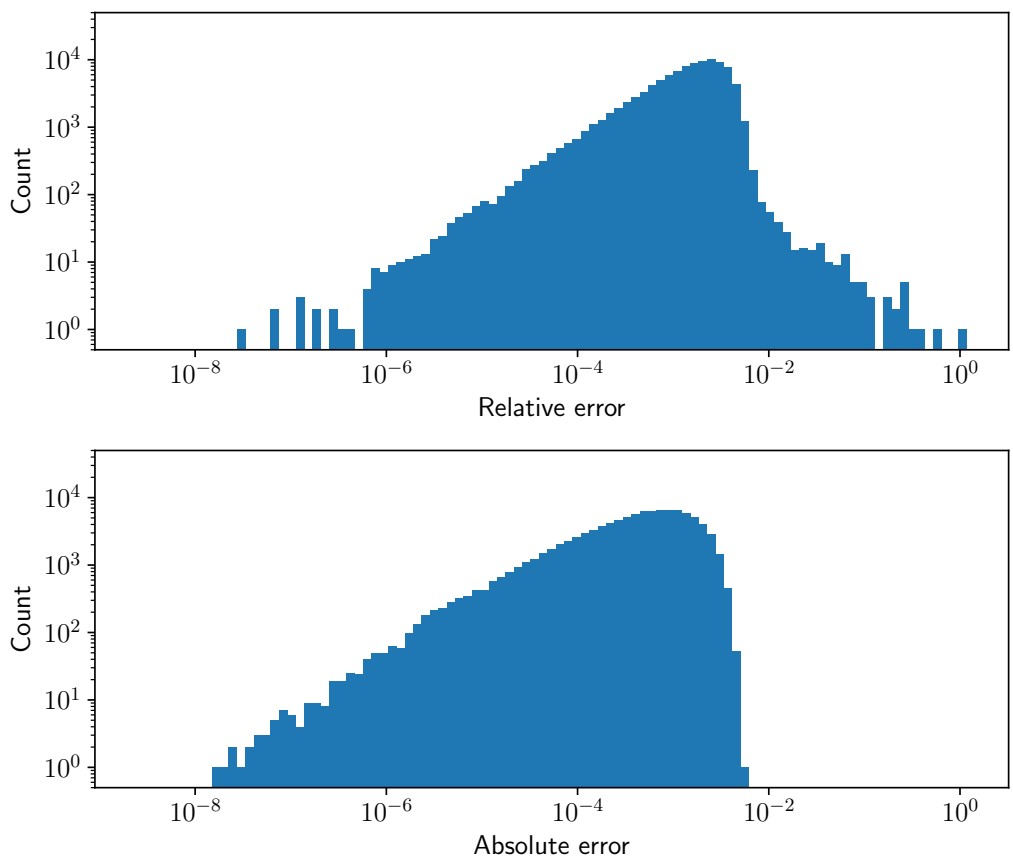

Figure 20: **Histogram of relative and absolute error.** The model is trained on the BHE of degree 100 with the constant initial condition. A single random seed is used for the training.

## H.6 Cross-degree Evaluation

In this section, we attempt to empirically investigate the theoretical convergence of the cylindrical approximation. The convergence is illustrated in Fig. 2, where PINN's training is not included, while our focus here is the convergence of the solutions estimated via PINN's training. According to the convergence theorem of the cylindrical approximation and the universal approximability of PINNs, the models trained on $D_m = \mathrm{span}\{\phi_0, \ldots, \phi_{m-1}\}$ with a large $m$ have a high capability of expression, and they can cover $D_n$ with $n \leq m$. On the other hand, the models trained on $D_m$ with a small $m$ have a high capability of expression only for small $m$s, and they *cannot* cover $D_n$ for $n \geq m$. In theory, this phenomenon can be observed by performing a *cross-degree evaluation*, in which we use training data sampled from $D_m$ and then evaluate the trained model on $D_{m'}$, where $m$ may not equal to $m'$.

The results are shown in Tabs. 13–22 below. Contrary to the aforementioned intuition, most of the cross-degree evaluations seem to contradict the convergence theorem of FDE solutions under the cylindrical approximation. Possible reasons are:

1. This is because of the curse of dimensionality, i.e., the optimization error of PINNs in high dimensions. Higher-dimensional PDEs require a much larger number of iterations to train PINNs; however, our experiments are performed with a fixed number of iterations (500,000 and 300,000 for the FTE and the BHE, respectively). Changing the number of iterations depending on the degree and applying other techniques for training PINNs [106, 44, 96, 45, 21, 103, 33] would decrease the optimization error, and we can focus on the approximation error and thus will observe the decay of error w.r.t. increasing $m$.

2. In addition, note that FDEs in our experiments, except for the FTE with the nonlinear initialization, are dominated by $a_0$ and/or $a_1$ only. Therefore, they show a tiny effect on relative and absolute errors by definition. Nevertheless, the FTE with the nonlinear initialization (Tabs. 15 and 16) does not show the theoretical convergence either, which means the optimization error of PINNs, rather than the cylindrical approximation error, dominates the errors anyways.

3. One can see a strong dependence of the errors on the experimental setups for PINNs. The absolute errors of the FTE under the nonlinear initial condition are reduced by a factor of 2 by simply changing the width of the PINN and the number of training iterations (1024 to 2048 and $5 \times 10^5$ to $8 \times 10^5$, respectively): from

   - $(2.46499 \pm 0.26492) \times 10^{-3}$ for $m = 4$,
   - $(20.3384 \pm 1.0449) \times 10^{-3}$ for $m = 10$, and
   - $(98.8854 \pm 2.7068) \times 10^{-3}$ for $m = 20$,

   to

   - $(1.43591 \pm 0.1832) \times 10^{-3}$ for $m = 4$,
   - $(11.0609 \pm 0.3671) \times 10^{-3}$ for $m = 10$, and
   - $(55.2715 \pm 2.3179) \times 10^{-3}$ for $m = 20$,

   where the error bars are the standard deviation over 10 runs. Note that the hyperparameters are different from those used in the main text. Moreover, as shown in Fig. 6, simply adding a regularization term $\|W([0], t)\|$ to the loss function reduces relative errors by an order of magnitude.

Therefore, to observe the convergence of the cylindrical approximation after PINN's training, careful hyperparameter tuning and optimization are needed. Note that optimizing PINNs for high-dimensional PDEs is an active field of research that has been developing (see Apps. A and F) and is of independent interest.

### H.6.1 Relative and Absolute Errors of Functional Transport Equation

Table 13: **Mean relative error** $\times 10^3$ **(functional transport equation with linear initialization) on test set.** The model is trained on the functional transport equation with the linear initialization condition. The error bars represent the standard error of mean with sample size 10 corresponding to different training seeds.

| | TRAINING SET DEGREE | | |
|---|---|---|---|
| TEST SET DEGREE | 4 | 20 | 100 |
| 4 | $1.26821 \pm 0.09936$ | $2.56697 \pm 0.20431$ | $14.6402 \pm 2.1739$ |
| 20 | $0.559035 \pm 0.026630$ | $2.01716 \pm 0.06875$ | $5.82534 \pm 0.49499$ |
| 100 | $0.483585 \pm 0.010872$ | $1.88511 \pm 0.05292$ | $6.24740 \pm 0.10591$ |

Table 14: **Mean absolute error** $\times 10^4$ **(functional transport equation with linear initialization) on test set.** The model is trained on the functional transport equation with the linear initialization condition. The error bars represent the standard error of mean with sample size 10 corresponding to different training seeds.

| | TRAINING SET DEGREE | | |
|---|---|---|---|
| TEST SET DEGREE | 4 | 20 | 100 |
| 4 | $1.32203 \pm 0.13933$ | $3.06281 \pm 0.28009$ | $15.2530 \pm 0.7296$ |
| 20 | $1.34579 \pm 0.14329$ | $2.29632 \pm 0.05205$ | $13.4665 \pm 0.7117$ |
| 100 | $1.32709 \pm 0.14107$ | $2.27854 \pm 0.05028$ | $12.3312 \pm 0.5986$ |

Table 15: **Mean relative error** $\times 10^3$ **(functional transport equation with nonlinear initialization) on test set.** The model is trained on the functional transport equation with the nonlinear initialization condition. The error bars represent the standard error of mean with sample size 10 corresponding to different training seeds.

| | TRAINING SET DEGREE | | |
|---|---|---|---|
| TEST SET DEGREE | 4 | 10 | 20 |
| 4 | $1.09131 \pm 0.02945$ | $2.06847 \pm 0.06531$ | $7.51237 \pm 0.25359$ |
| 10 | $1.14029 \pm 0.03166$ | $3.59902 \pm 0.05139$ | $8.61269 \pm 0.10351$ |
| 20 | $1.11229 \pm 0.02789$ | $3.62669 \pm 0.06044$ | $11.7414 \pm 0.1112$ |

Table 16: **Mean absolute error** $\times 10^3$ **(functional transport equation with nonlinear initialization) on test set.** The model is trained on the functional transport equation with the nonlinear initialization condition. Note that the hyperparameters are different from those used in the main text. The error bars represent the standard error of mean with sample size 10 corresponding to different training seeds.

| | TRAINING SET DEGREE | | |
|---|---|---|---|
| TEST SET DEGREE | 4 | 10 | 20 |
| 4 | $2.46499 \pm 0.08377$ | $9.00259 \pm 0.21308$ | $47.5449 \pm 0.6410$ |
| 10 | $2.46885 \pm 0.08409$ | $20.3384 \pm 0.3304$ | $69.6191 \pm 0.7526$ |
| 20 | $2.48688 \pm 0.08436$ | $20.5866 \pm 0.3346$ | $98.8854 \pm 0.8560$ |

## H.6.2 Relative and Absolute Errors of Burgers-Hopf Equation

Table 17: **Mean relative error** $\times 10^4$ **(BHE with delta initialization) on test set.** The model is trained on the BHE with the delta initialization condition. The error bars represent the standard error of mean with sample size 10 corresponding to different training seeds.

| | TRAINING SET DEGREE | | |
|---|---|---|---|
| TEST SET DEGREE | 4 | 20 | 100 |
| 4 | $2.93905 \pm 0.05503$ | $2.97886 \pm 0.03187$ | $2.99835 \pm 0.03851$ |
| 20 | $2.24827 \pm 0.07876$ | $2.20842 \pm 0.09022$ | $2.23898 \pm 0.05530$ |
| 100 | $4.11689 \pm 0.34740$ | $2.98177 \pm 0.20264$ | $2.41667 \pm 0.07989$ |

Table 18: **Mean absolute error** $\times 10^5$ **(BHE with delta initialization) on test set.** The model is trained on the BHE with the delta initialization condition. The error bars represent the standard error of mean with sample size 10 corresponding to different training seeds.

| | TRAINING SET DEGREE | | |
|---|---|---|---|
| TEST SET DEGREE | 4 | 20 | 100 |
| 4 | $1.66451 \pm 0.15036$ | $1.35700 \pm 0.03771$ | $1.62045 \pm 0.02767$ |
| 20 | $1.66216 \pm 0.15005$ | $1.34640 \pm 0.03828$ | $1.60869 \pm 0.02892$ |
| 100 | $1.66510 \pm 0.15030$ | $1.34980 \pm 0.03779$ | $1.60980 \pm 0.02831$ |

Table 19: **Mean relative error** $\times 10^5$ **(BHE with constant initialization) on test set.** The model is trained on the BHE with the constant initialization condition. The error bars represent the standard error of mean with sample size 10 corresponding to different training seeds.

| | TRAINING SET DEGREE | | |
|---|---|---|---|
| TEST SET DEGREE | 4 | 20 | 100 |
| 4 | $12.1782 \pm 2.70298$ | $8.32134 \pm 0.72992$ | $8.44530 \pm 1.4148$ |
| 20 | $9.07441 \pm 1.50640$ | $6.14352 \pm 0.40680$ | $5.99939 \pm 0.68357$ |
| 100 | $8.26214 \pm 1.17503$ | $5.75086 \pm 0.35933$ | $5.50375 \pm 0.55500$ |

Table 20: **Mean absolute error** $\times 10^5$ **(BHE with constant initialization) on test set.** The model is trained on the BHE with the constant initialization condition. The error bars represent the standard error of mean with sample size 10 corresponding to different training seeds.

| | TRAINING SET DEGREE | | |
|---|---|---|---|
| TEST SET DEGREE | 4 | 20 | 100 |
| 4 | $1.62849 \pm 0.21154$ | $1.12380 \pm 0.10234$ | $1.05562 \pm 0.05272$ |
| 20 | $1.62892 \pm 0.21162$ | $1.12371 \pm 0.10235$ | $1.05556 \pm 0.05272$ |
| 100 | $1.62931 \pm 0.21148$ | $1.12369 \pm 0.10231$ | $1.05558 \pm 0.05273$ |

Table 21: **Mean absolute error** $\times 10^3$ **(BHE with moderate initialization) on test set.** The model is trained on the BHE with the moderate initialization condition. The error bars represent the standard error of mean with sample size 10 corresponding to different training seeds.

| | TRAINING SET DEGREE | | |
|---|---|---|---|
| TEST SET DEGREE | 4 | 20 | 100 |
| 4 | $1.32699 \pm 0.06872$ | $1.60725 \pm 0.03521$ | $1.84851 \pm 0.01689$ |
| 20 | $1.24102 \pm 0.04793$ | $1.63232 \pm 0.02236$ | $1.92692 \pm 0.00862$ |
| 100 | $1.18515 \pm 0.04836$ | $1.55581 \pm 0.02152$ | $1.82844 \pm 0.00684$ |

Table 22: **Mean absolute error** $\times 10^4$ **(BHE with moderate initialization) on test set.** The model is trained on the BHE with the moderate initialization condition. The error bars represent the standard error of mean with sample size 10 corresponding to different training seeds.

| | TRAINING SET DEGREE | | |
|---|---|---|---|
| TEST SET DEGREE | 4 | 20 | 100 |
| 4 | $3.91187 \pm 0.10867$ | $5.49339 \pm 0.03494$ | $6.73537 \pm 0.01891$ |
| 20 | $3.93392 \pm 0.11011$ | $5.63933 \pm 0.03427$ | $6.87334 \pm 0.01884$ |
| 100 | $3.90431 \pm 0.10875$ | $5.60266 \pm 0.03403$ | $6.82414 \pm 0.01860$ |

## H.7    Loss Curves and Learning Rate Scheduling

Loss curves and learning rates are provided in Figs. 21–29.

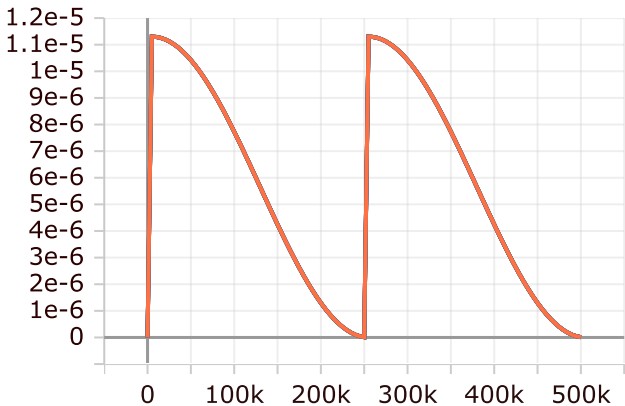

Figure 21: **Learning rate scheduling.** FTE with nonlinear initial condition with degree 1000.

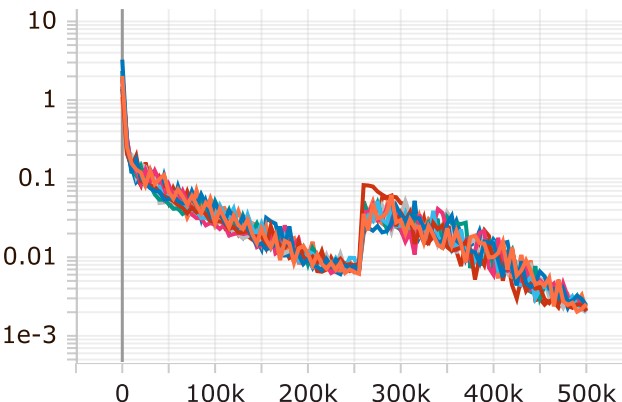

Figure 22: **Training loss.** FTE with nonlinear initial condition with degree 1000.

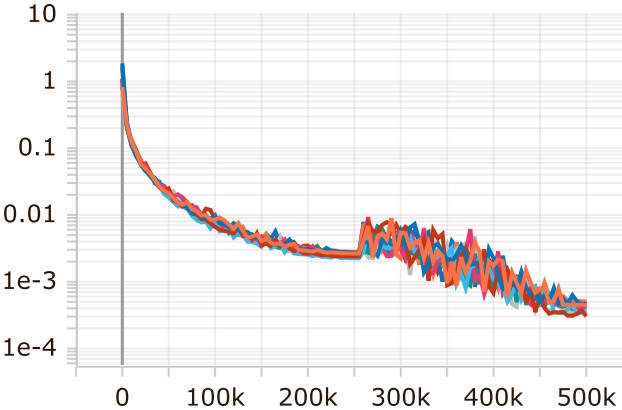

Figure 23: **Validation loss.** FTE with nonlinear initial condition with degree 1000.

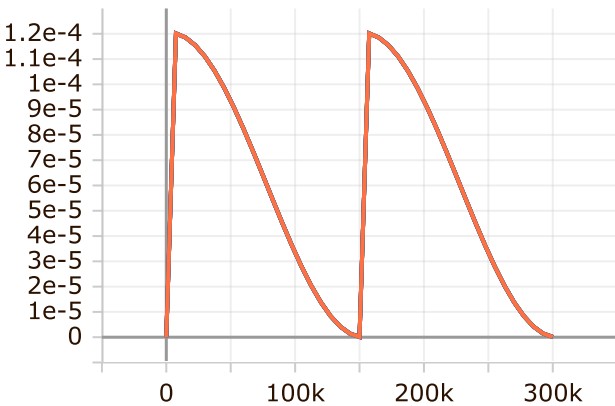

Figure 24: **Learning rate scheduling.** BHE with delta initial condition with degree 100.

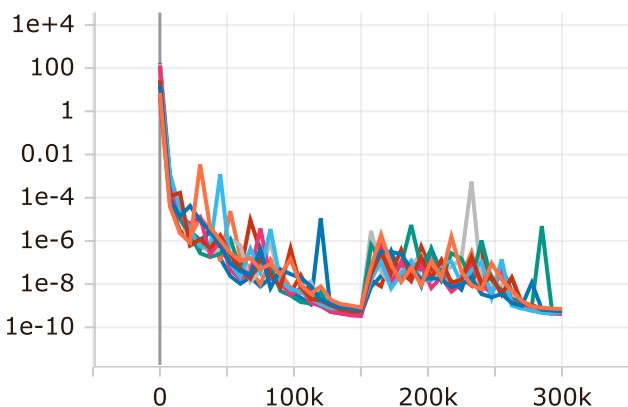

Figure 25: **Training loss.** BHE with delta initial condition with degree 100.

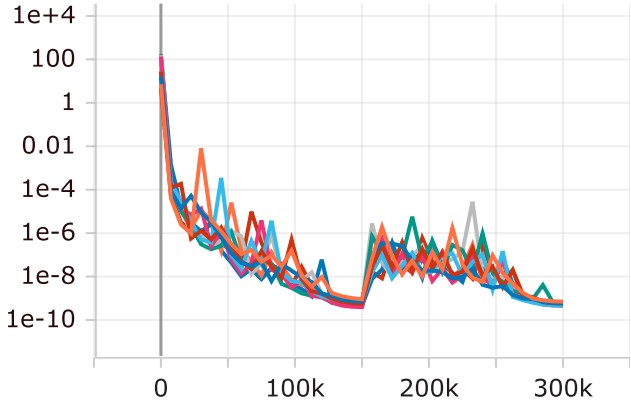

Figure 26: **Validation loss.** BHE with delta initial condition with degree 100.

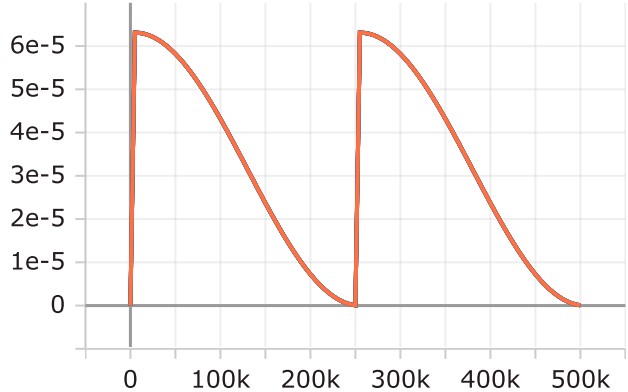

Figure 27: **Learning rate scheduling.** ARE with degree 6.

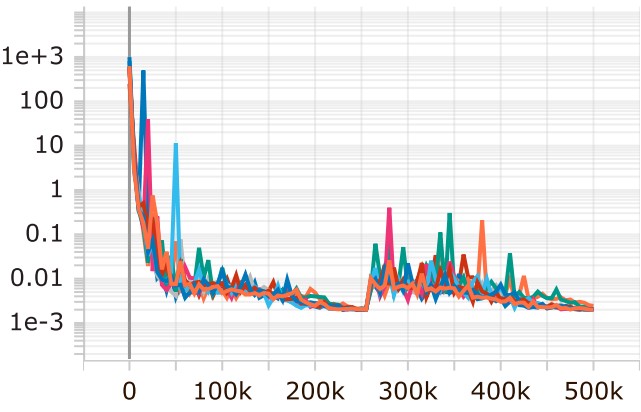

Figure 28: **Training loss.** ARE with degree 6.

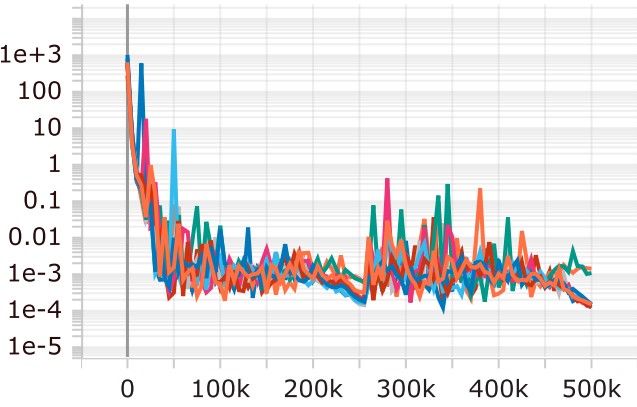

Figure 29: **Validation loss.** ARE with degree 6.

## H.8 Additional Figures: Functional Transport Equation

### H.8.1 Worst Coefficients and Best Coefficients

We provide the worst-/best-case relative/absolute error. The mean and minimum of errors in the test set do not strongly depend on training random seeds, while the maximum of errors intensely fluctuates. This observation tells us that the worst-case relative and absolute errors are not stable measures of performance.

Table 23: Relative error at the collocation point with the smallest relative error. The models are trained on the functional transport equation.

| | INITIAL CONDITION | | | INITIAL CONDITION |
|---|---|---|---|---|
| DEGREE | LINEAR | | DEGREE | NONLINEAR |
| 4 | $0.00000 \pm 0.00000$ | | 4 | $0.00000 \pm 0.00000$ |
| 20 | $0.00000 \pm 0.00000$ | | 10 | $0.00000 \pm 0.00000$ |
| 100 | $0.00000 \pm 0.00000$ | | 20 | $0.00000 \pm 0.00000$ |

Table 24: Absolute error at the collocation point whose absolute error is *smallest*. The models are trained on the functional transport equation.

| | INITIAL CONDITION | | | INITIAL CONDITION |
|---|---|---|---|---|
| DEGREE | LINEAR | | DEGREE | NONLINEAR |
| 4 | $0.00000 \pm 0.00000$ | | 4 | $0.00000 \pm 0.00000$ |
| 20 | $0.00000 \pm 0.00000$ | | 10 | $0.00000 \pm 0.00000$ |
| 100 | $0.00000 \pm 0.00000$ | | 20 | $0.00000 \pm 0.00000$ |

Table 25: Relative error at the collocation point whose relative error is *largest*. The models are trained on the functional transport equation.

| | INITIAL CONDITION | | | INITIAL CONDITION |
|---|---|---|---|---|
| DEGREE | LINEAR | | DEGREE | NONLINEAR |
| 4 | $51.4467 \pm 22.3805$ | | 4 | $2.60833 \pm 1.03956$ |
| 20 | $23.0638 \pm 11.7964$ | | 10 | $8.17635 \pm 3.50015$ |
| 100 | $52.0792 \pm 19.0932$ | | 20 | $13.9782 \pm 7.87448$ |

Table 26: Absolute error at the collocation point whose absolute error is *largest*. The models are trained on the functional transport equation.

| | INITIAL CONDITION |
| --- | --- |
| DEGREE | LINEAR |
| 4 | $(7.91653 \pm 2.14557) \times 10^{-3}$ |
| 20 | $(8.39847 \pm 1.54967) \times 10^{-3}$ |
| 100 | $(34.2531 \pm 6.2608) \times 10^{-3}$ |

| | INITIAL CONDITION |
| --- | --- |
| DEGREE | NONLINEAR |
| 4 | $0.19718 \pm 0.01640$ |
| 10 | $1.07628 \pm 0.06364$ |
| 20 | $2.53652 \pm 0.08511$ |

### H.8.2 First-order Functional Derivative: Linear Initial Condition & Relative Error

The relative errors of the first-order functional derivative are provided. The models are trained under the linear initial condition.

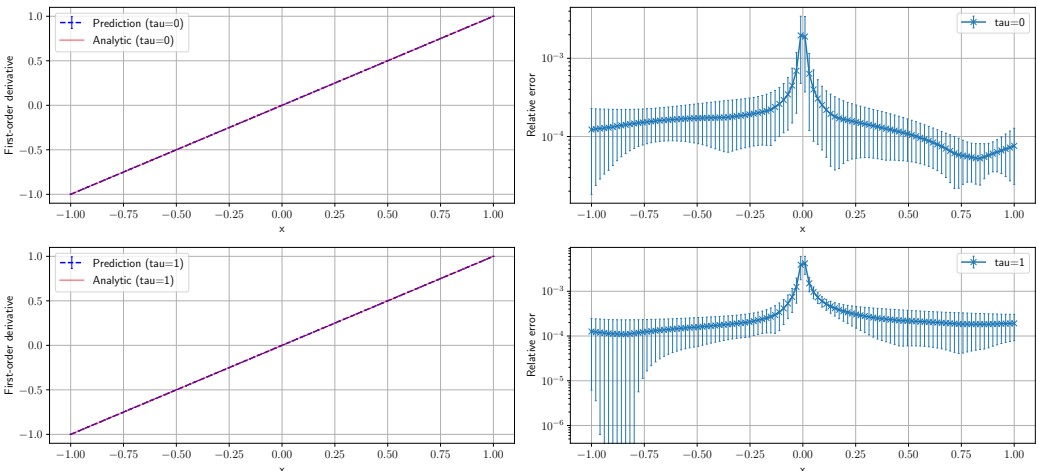

Figure 30: Degree 4.

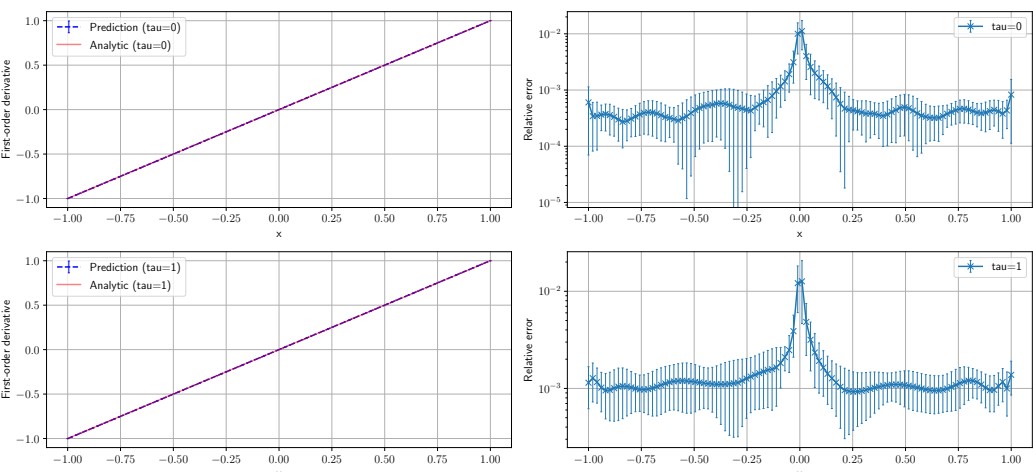

Figure 31: Degree 20.

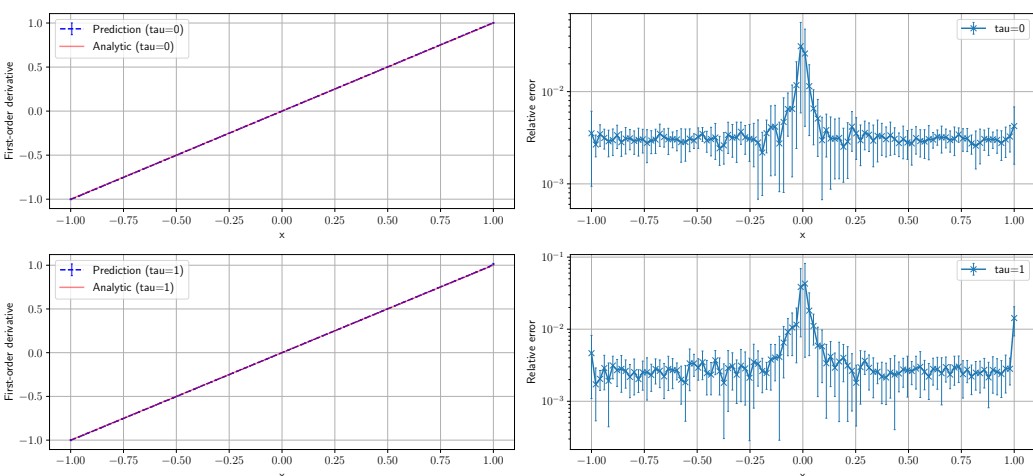

Figure 32: Degree 100.

### H.8.3 First-order Functional Derivative: Nonlinear Initial Condition & Relative Error

The relative errors of the first-order functional derivative are provided. The models are trained under the nonlinear initial condition.

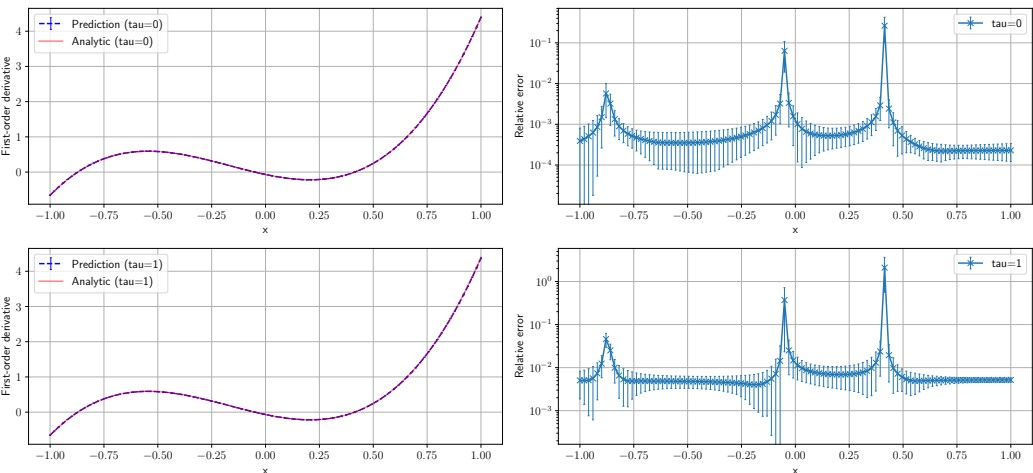

Figure 33: Degree 4.

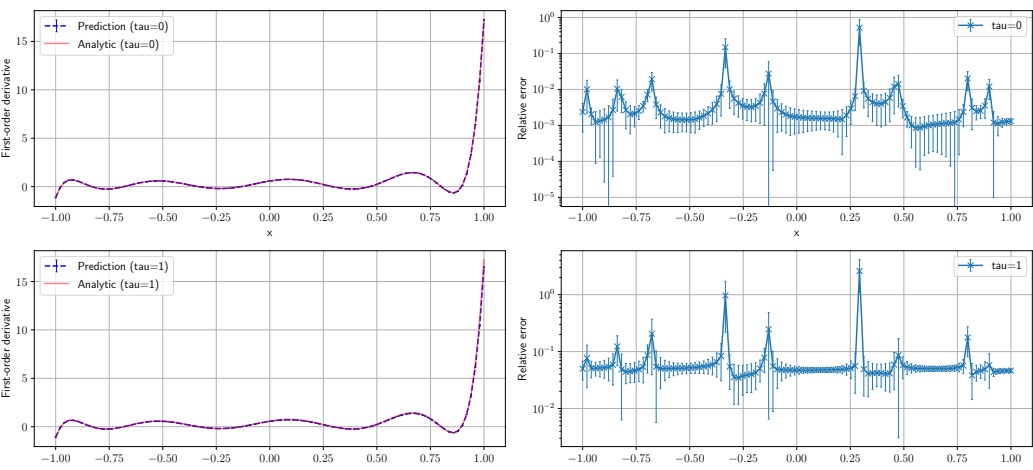

Figure 34: Degree 10.

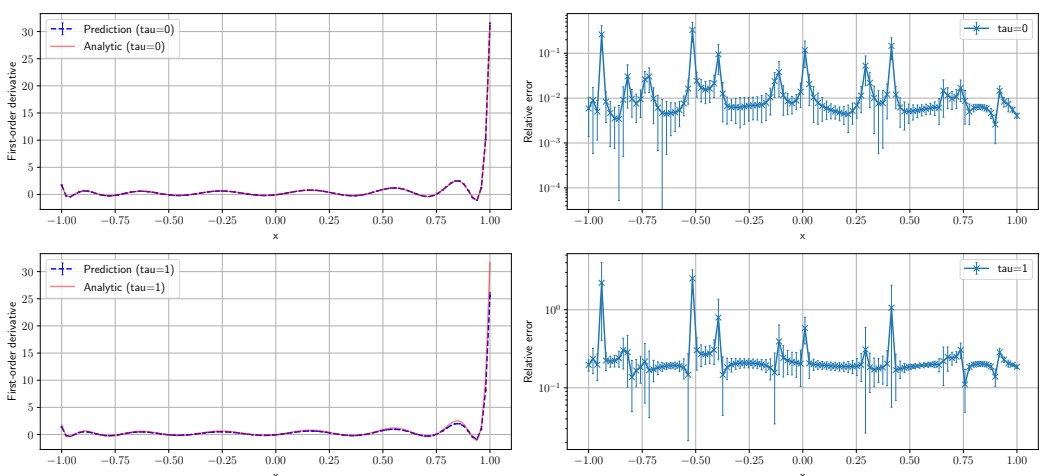

Figure 35: Degree 20.

### H.8.4 First-order Functional Derivative: Linear Initial Condition & Absolute Error

The absolute errors of the first-order functional derivative are provided. The models are trained under the linear initial condition.

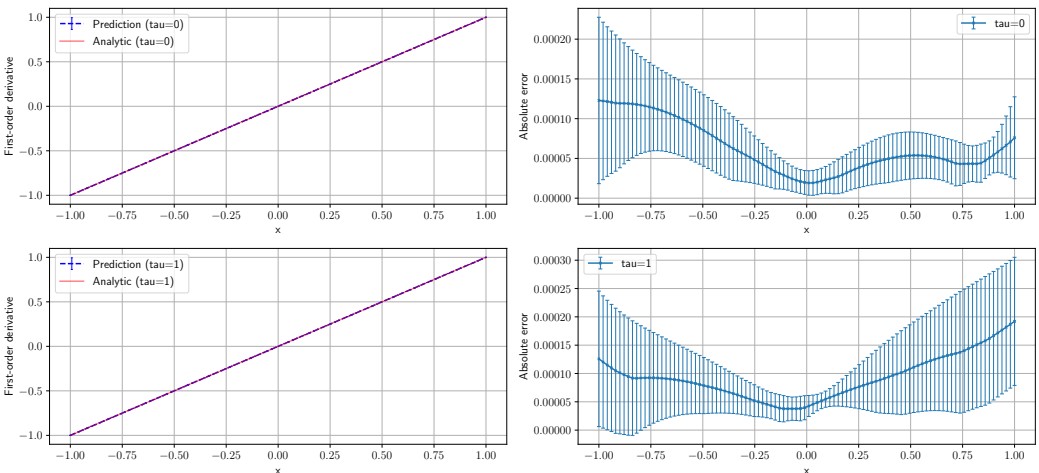

Figure 36: Degree 4.

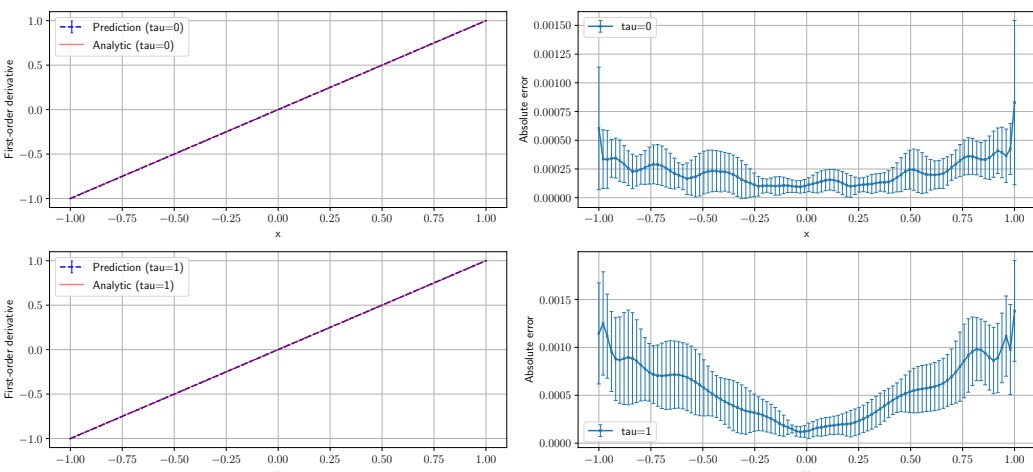

Figure 37: Degree 20.

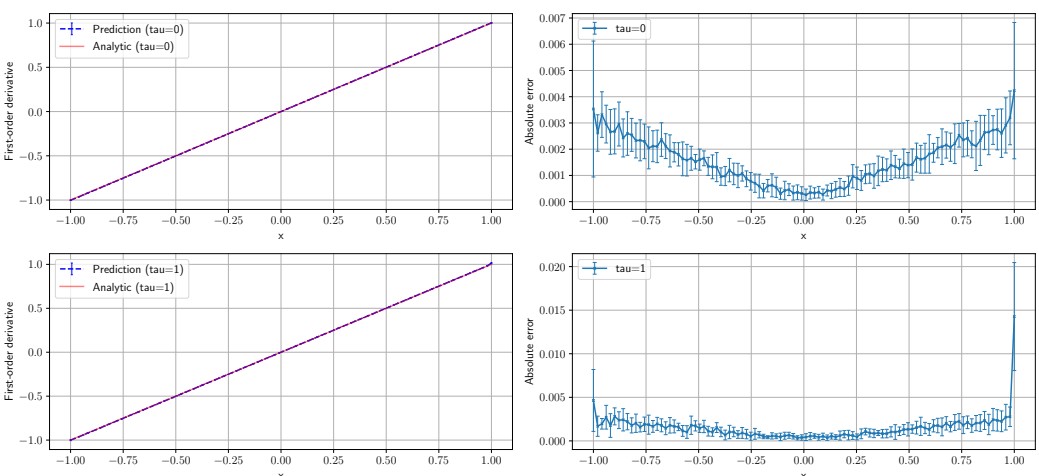

Figure 38: Degree 100.

### H.8.5 First-order Functional Derivative: Nonlinear Initial Condition & Absolute Error

The absolute errors of the first-order functional derivative are provided. The models are trained under the nonlinear initial condition.

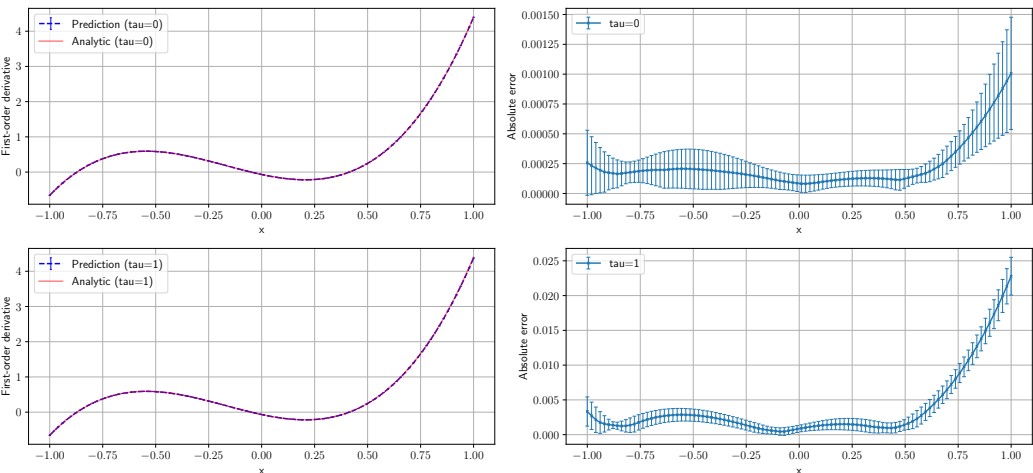

Figure 39: Degree 4.

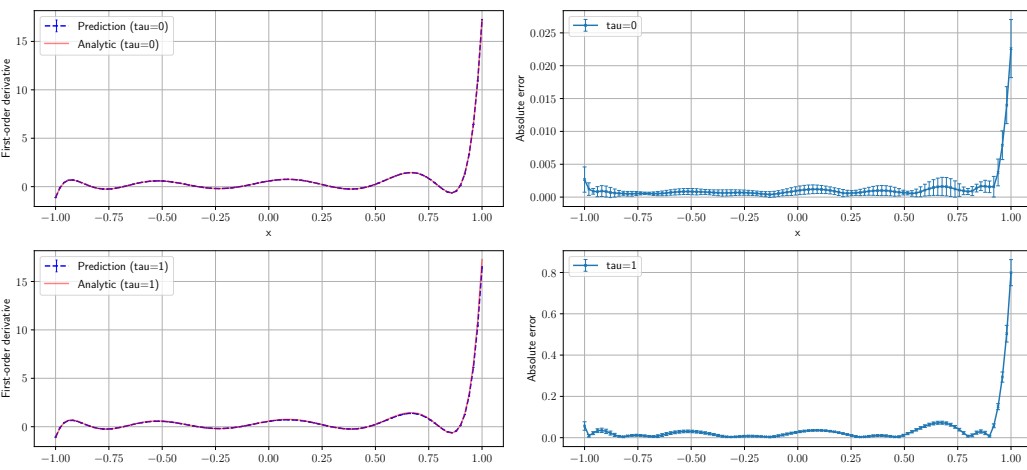

Figure 40: Degree 10.

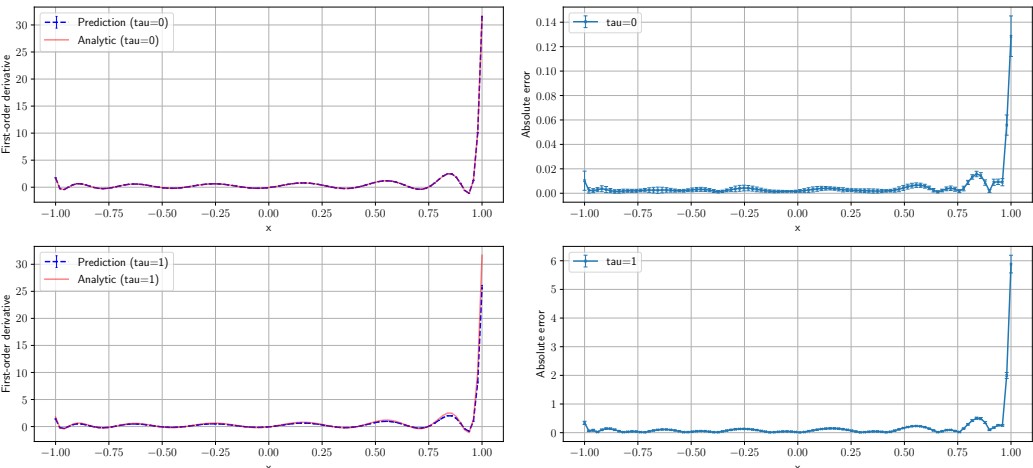

Figure 41: Degree 20.

### H.8.6 Time-dependent Relative and Absolute Errors at $\theta = 0$: Linear Initial Condition

The time-dependent relative and absolute errors at $\theta = 0$ are provided. The models are trained under the linear initial condition.

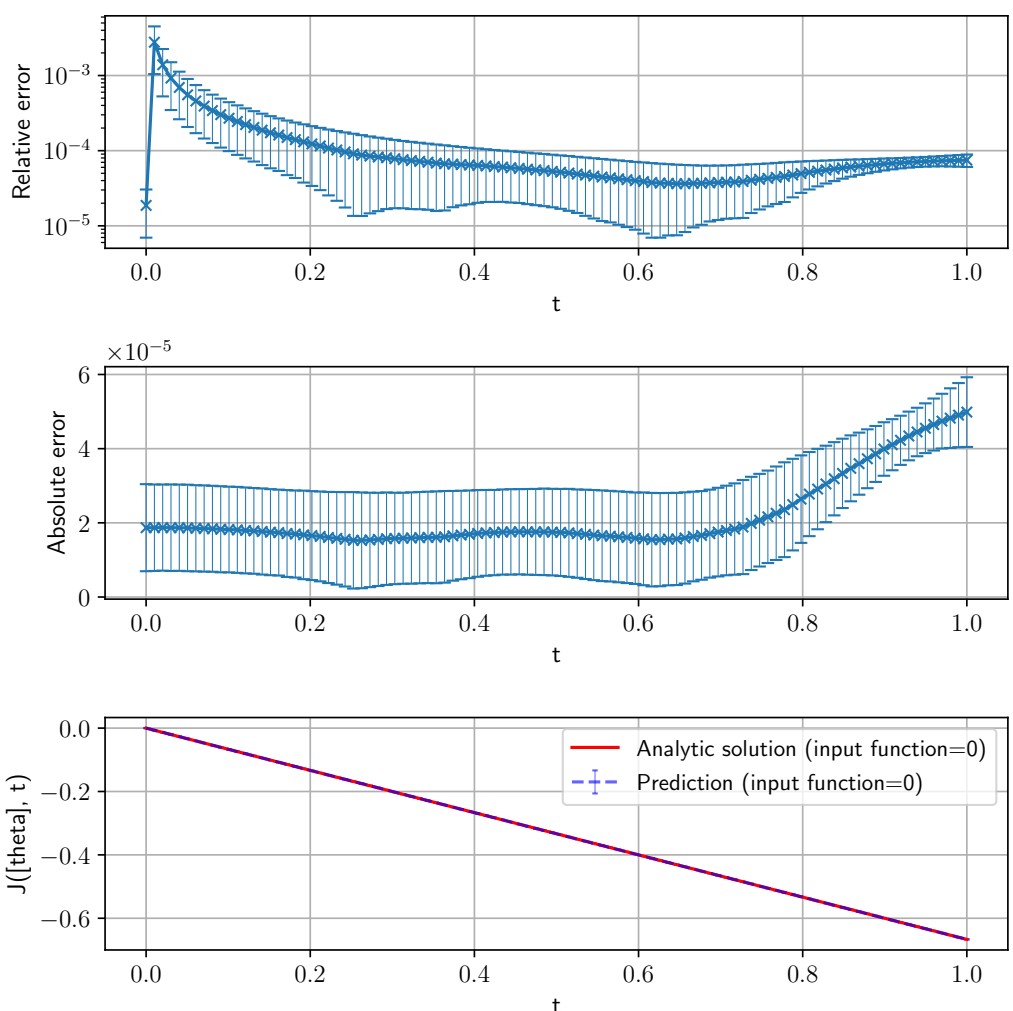

Figure 42: Degree 4.

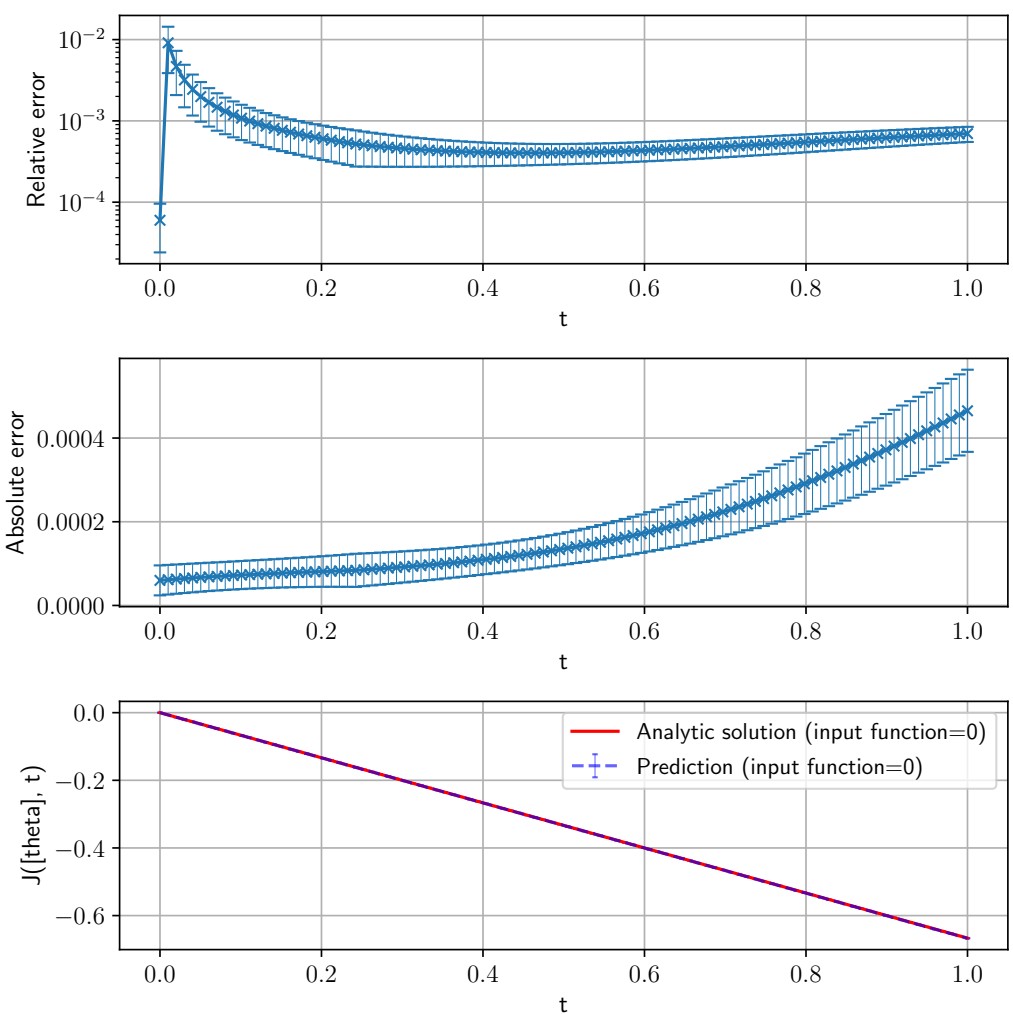

Figure 43: Degree 20.

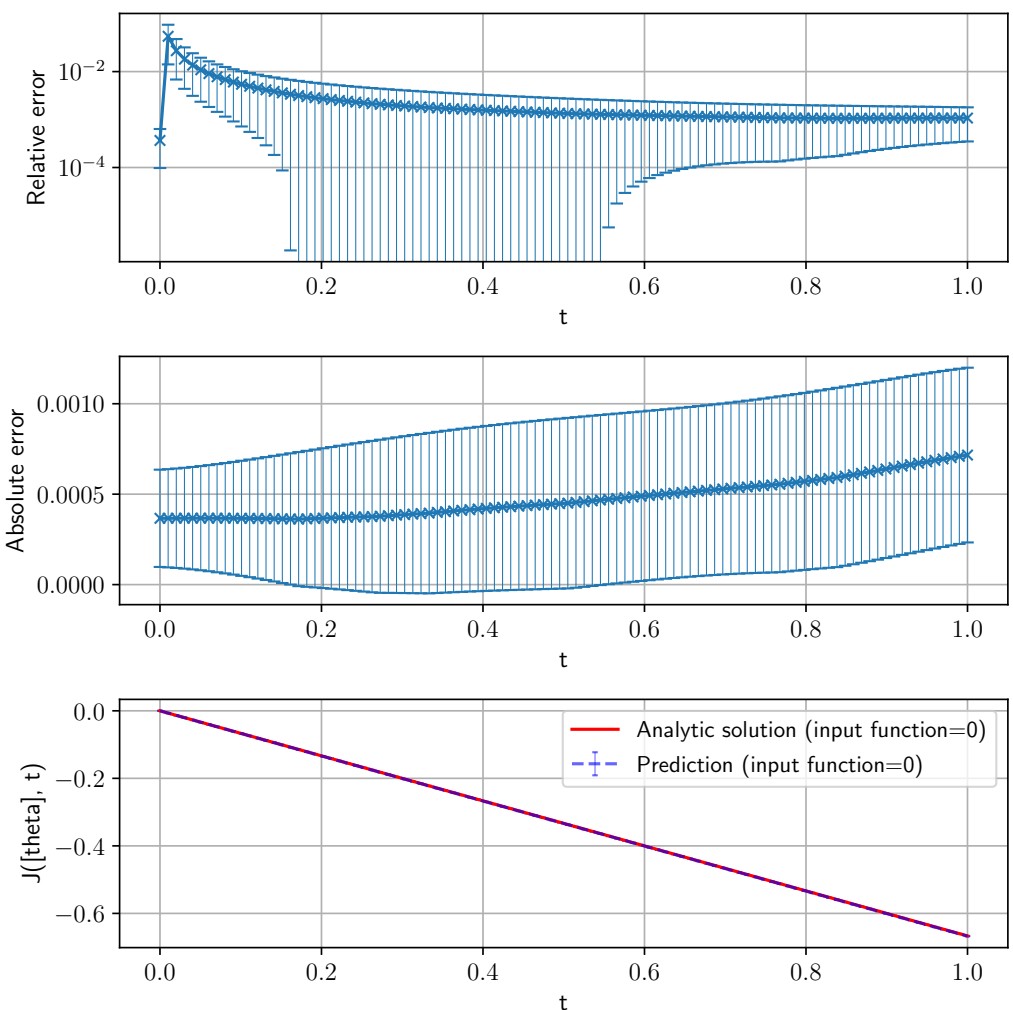

Figure 44: Degree 100.

### H.8.7  Time-dependent Relative and Absolute Errors at $\theta = 0$: Nonlinear Initial Condition

The time-dependent relative and absolute errors at $\theta = 0$ are provided. The models are trained under the nonlinear initial condition.

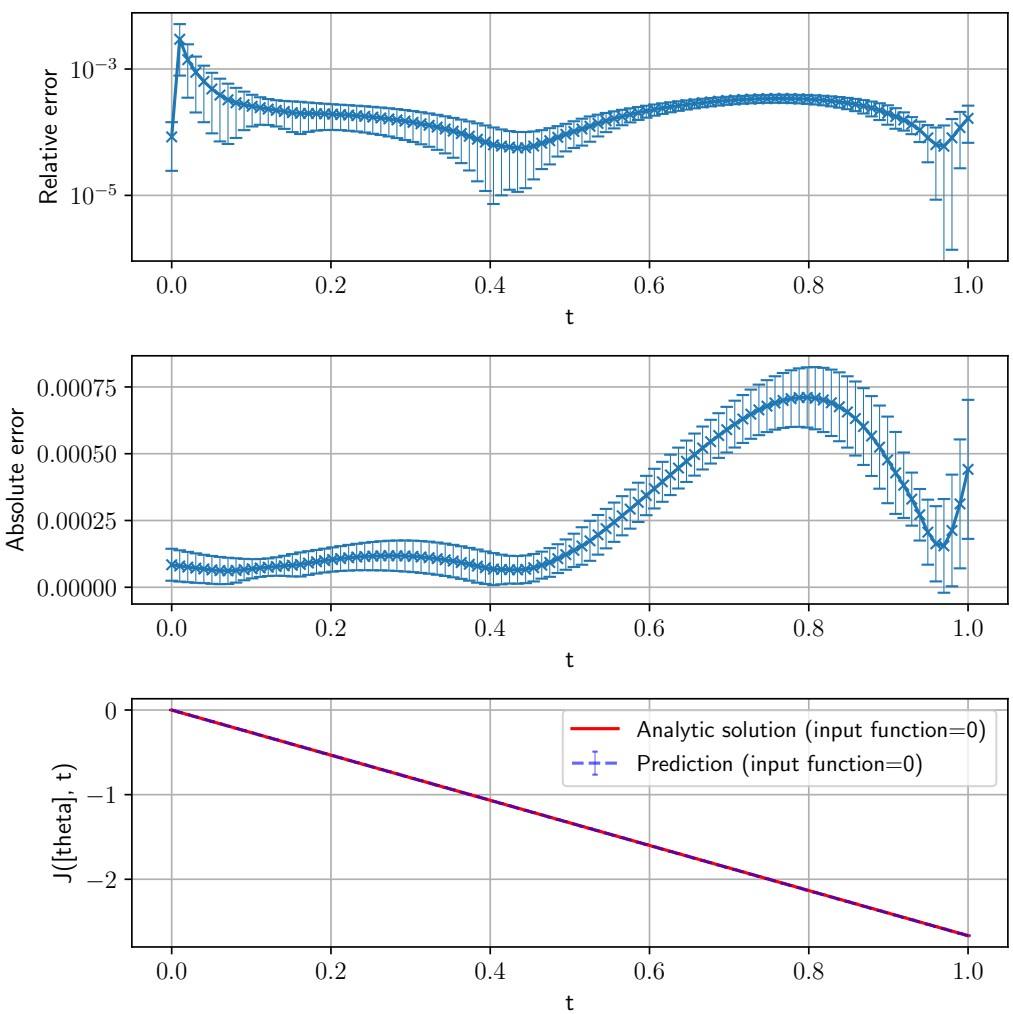

Figure 45: Degree 4.

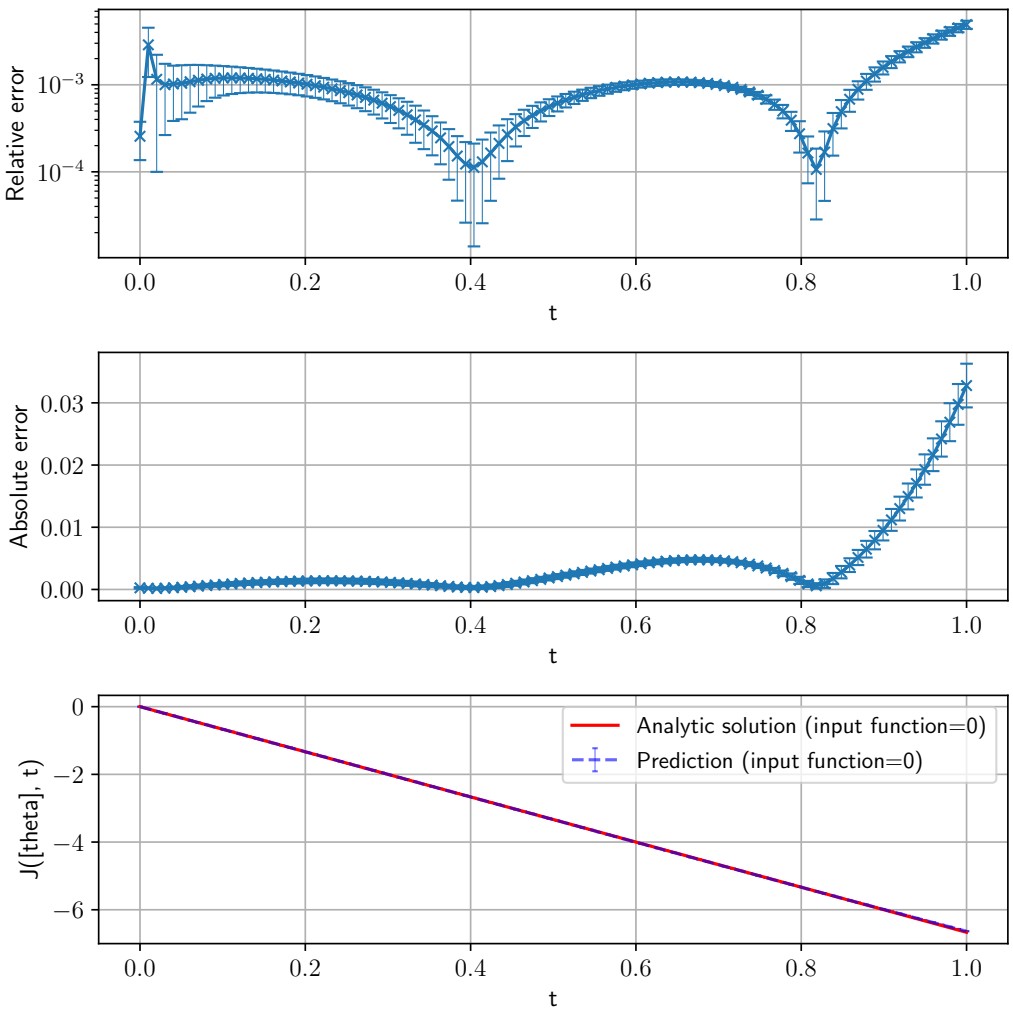

Figure 46: Degree 10.

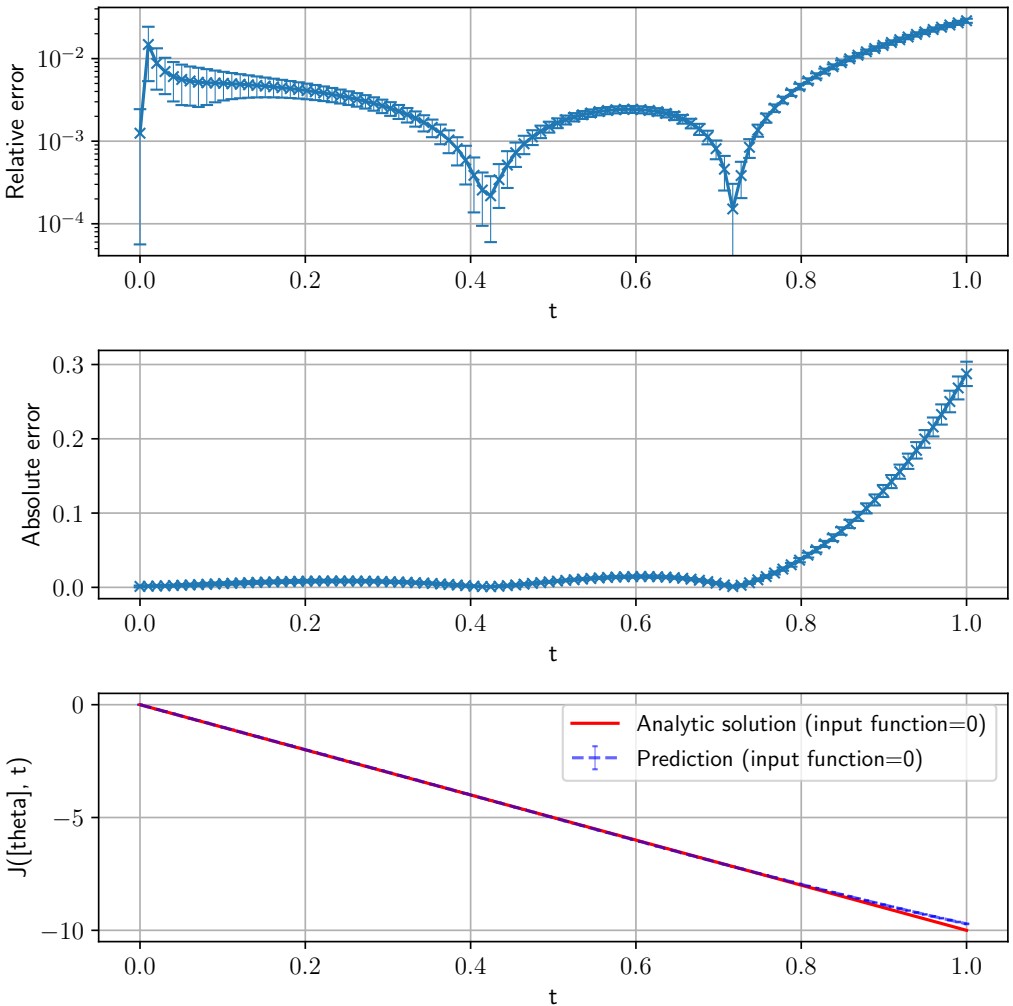

Figure 47: Degree 20.

### H.8.8 Time-averaged Pointwise Relative Error: Linear Initial Condition

The time-averaged pointwise relative errors are provided. The models are trained under the linear initial condition.

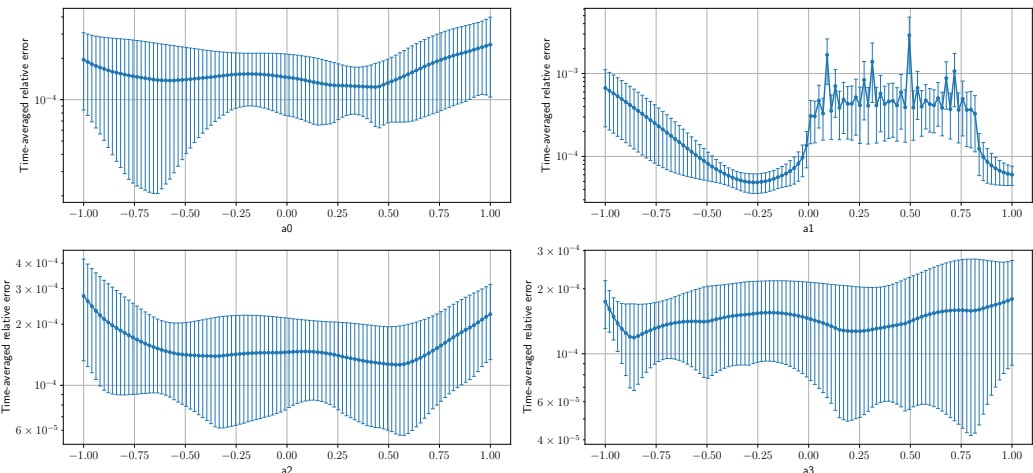

Figure 48: Degree 4.

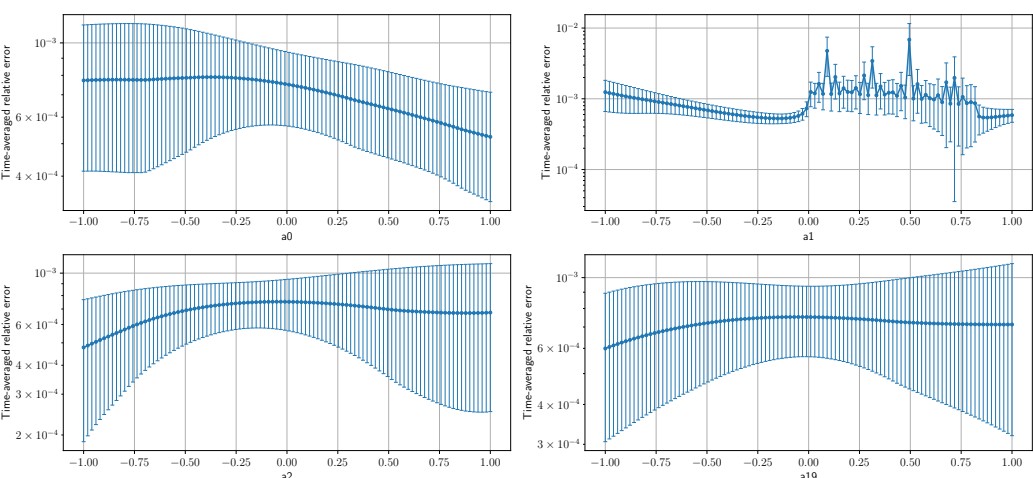

Figure 49: Degree 20.

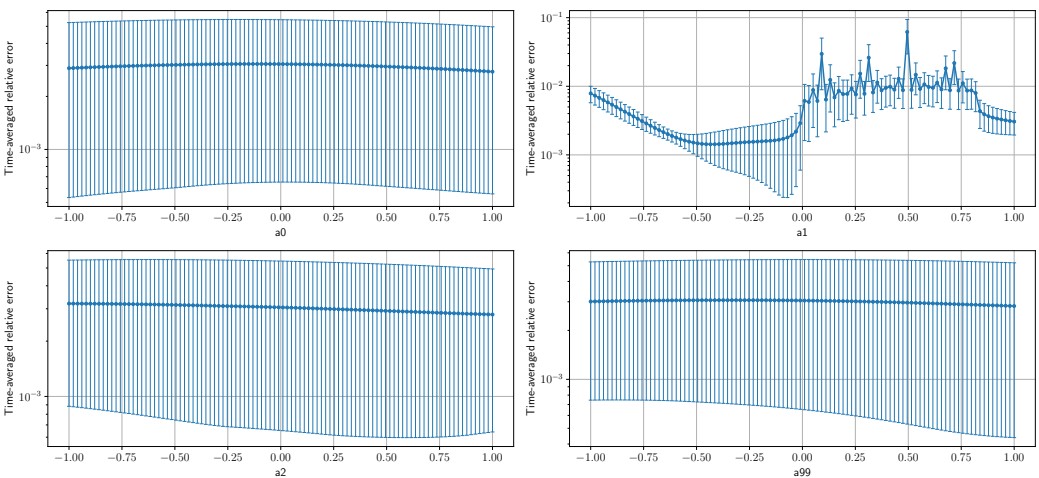

Figure 50: Degree 100.

### H.8.9  Time-averaged Pointwise Relative Error: Nonlinear Initial Condition

The time-averaged pointwise relative errors are provided. The models are trained under the nonlinear initial condition.

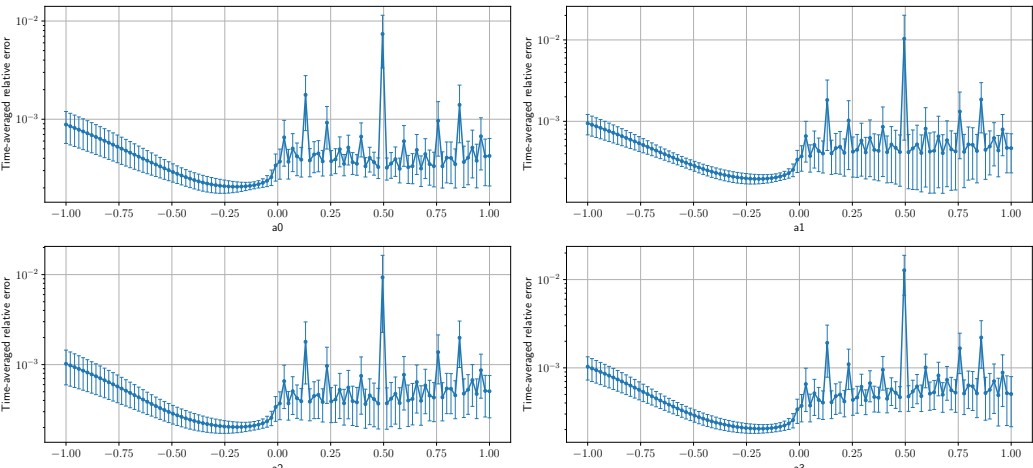

Figure 51: Degree 4.

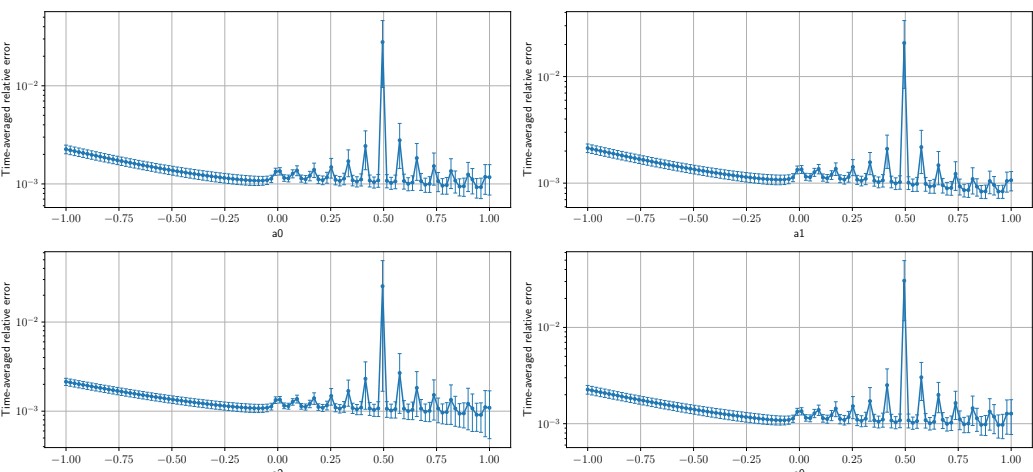

Figure 52: Degree 10.

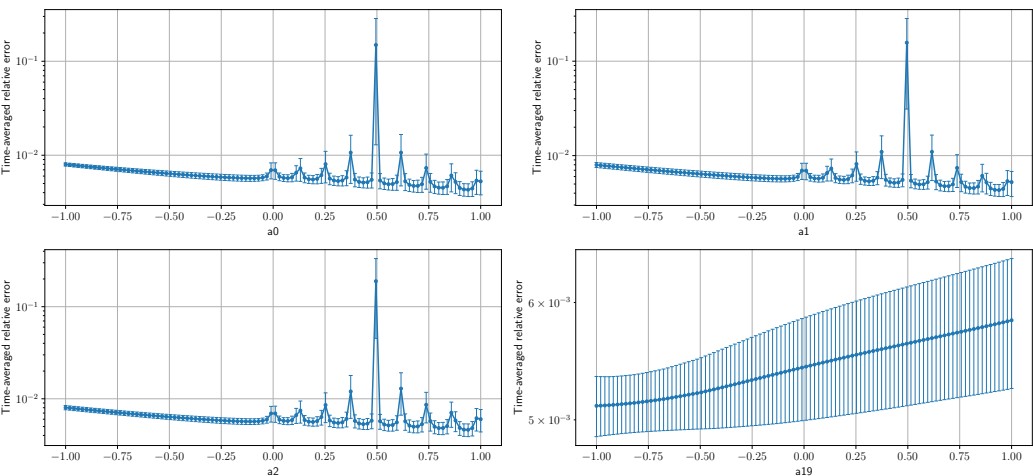

Figure 53: Degree 20.

### H.8.10 Time-averaged Pointwise Absolute Error: Linear Initial Condition

The time-averaged pointwise absolute errors are provided. The models are trained under the linear initial condition.

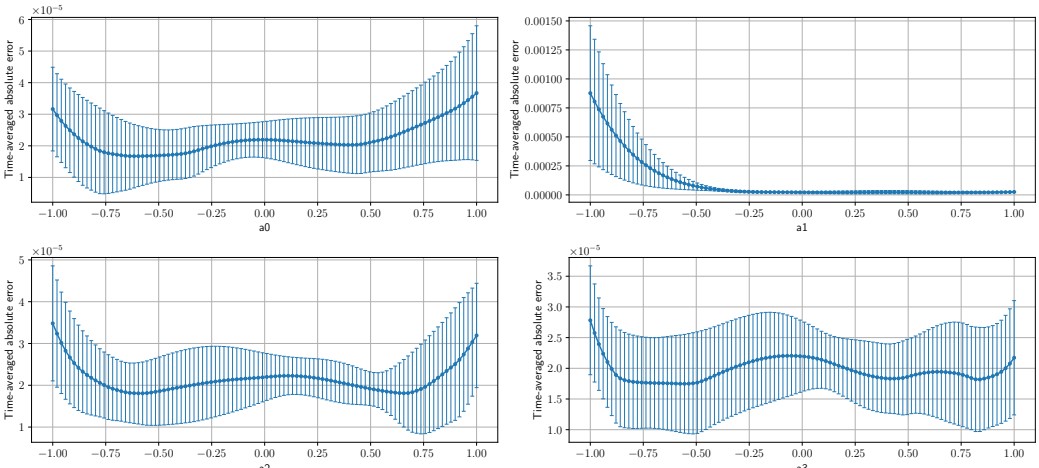

Figure 54: Degree 4.

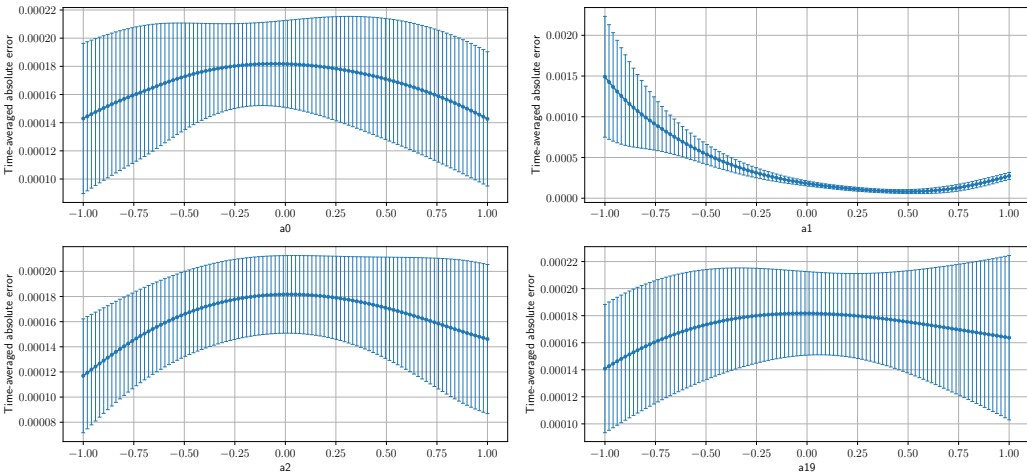

Figure 55: Degree 20.

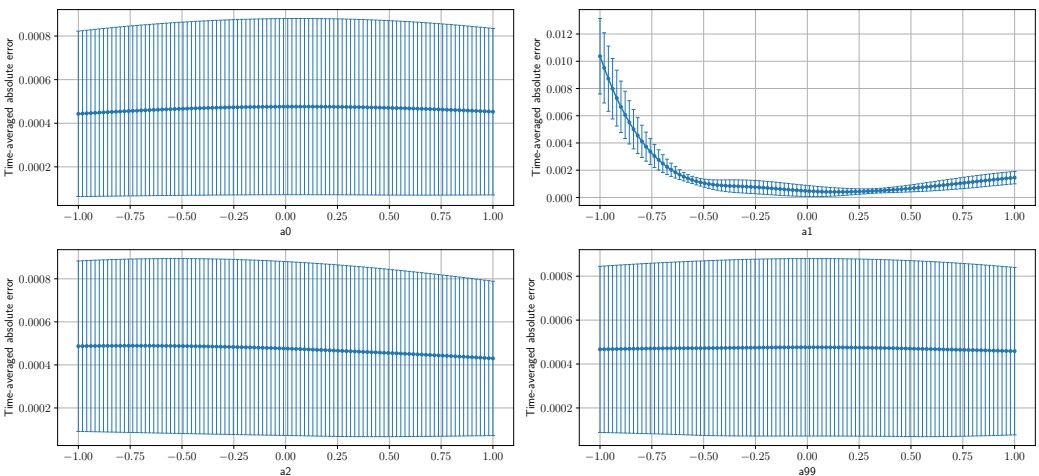

Figure 56: Degree 100.

### H.8.11 Time-averaged Pointwise Absolute Error: Nonlinear Initial Condition

The time-averaged pointwise absolute errors are provided. The models are trained under the nonlinear initial condition.

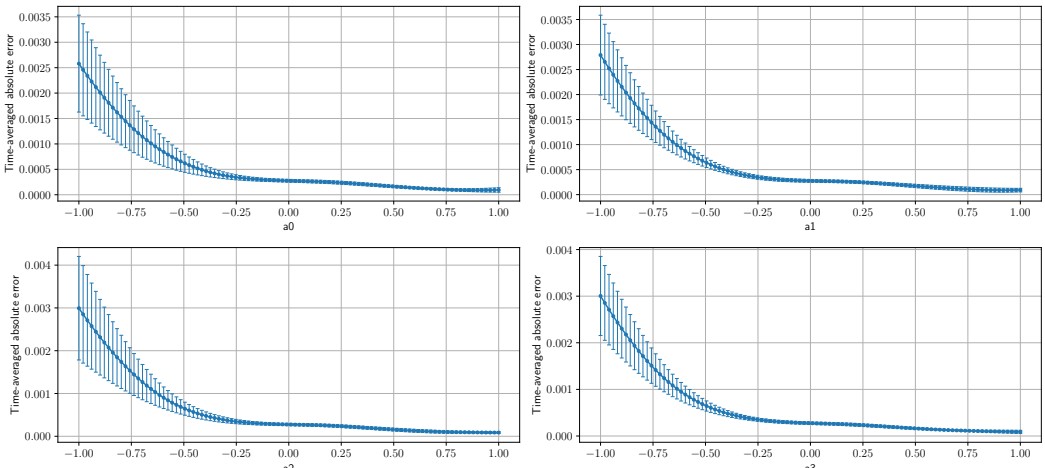

Figure 57: Degree 4.

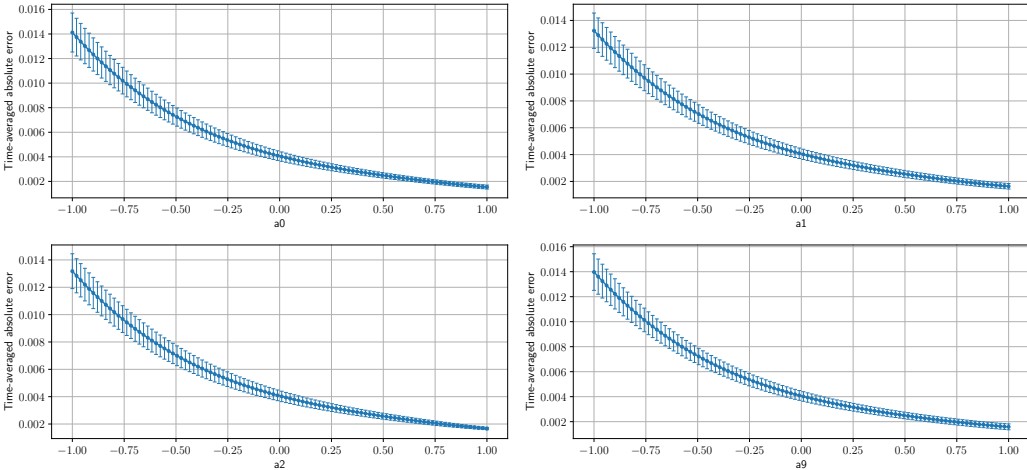

Figure 58: Degree 10.

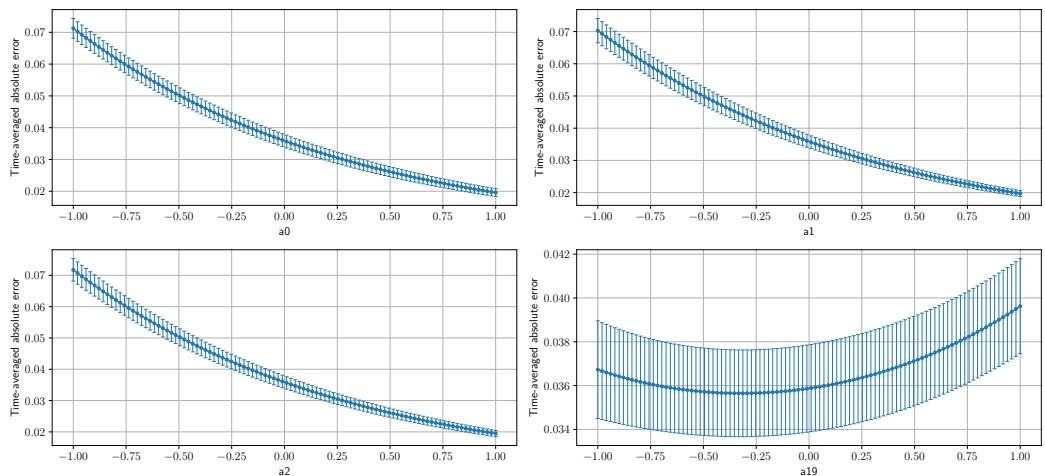

Figure 59: Degree 20.

## H.9 Additional Figures: Burgers-Hopf Equation

### H.9.1 Worst Coefficients and Best Coefficients

We provide the worst-/best-case relative/absolute error. The mean and minimum of errors in the test set do not strongly depend on training random seeds, while the maximum of errors intensely fluctuates. This observation tells us that the worst-case relative and absolute errors are not stable measures of performance.

Table 27: **Relative error at the collocation point whose relative error is *smallest*.** The models are trained on the BHE. The error bars are the standard deviation over 10 training runs with different random seeds.

| | INITIAL CONDITION | | |
| --- | --- | --- | --- |
| DEGREE | DELTA | MODERATE | CONSTANT |
| 4 | $(7.52793 \pm 7.45055) \times 10^{-10}$ | $(1.03507 \pm 1.15271) \times 10^{-8}$ | $(6.89188 \pm 9.74890) \times 10^{-10}$ |
| 20 | $(7.74760 \pm 3.91492) \times 10^{-10}$ | $(1.04685 \pm 0.68460) \times 10^{-8}$ | $(5.03755 \pm 4.77551) \times 10^{-10}$ |
| 100 | $(10.6824 \pm 4.8721) \times 10^{-10}$ | $(2.21203 \pm 1.65951) \times 10^{-8}$ | $(2.03155 \pm 2.45872) \times 10^{-10}$ |

Table 28: **Absolute error at the collocation point whose absolute error is *smallest*.** The models are trained on the BHE. The error bars are the standard deviation over 10 training runs with different random seeds.

| | INITIAL CONDITION | | |
| --- | --- | --- | --- |
| DEGREE | DELTA | MODERATE | CONSTANT |
| 4 | $(2.24063 \pm 1.79890) \times 10^{-10}$ | $(2.87494 \pm 2.50505) \times 10^{-9}$ | $(1.84662 \pm 2.89692) \times 10^{-10}$ |
| 20 | $(3.64107 \pm 1.88671) \times 10^{-10}$ | $(1.64646 \pm 1.14713) \times 10^{-9}$ | $(1.73323 \pm 1.31036) \times 10^{-10}$ |
| 100 | $(3.87652 \pm 2.24305) \times 10^{-10}$ | $(4.42345 \pm 3.11587) \times 10^{-9}$ | $(0.98167 \pm 1.1928) \times 10^{-10}$ |

Table 29: **Relative error at the collocation point whose relative error is *largest*.** The models are trained on the BHE. The error bars are the standard deviation over 10 training runs with different random seeds.

| | INITIAL CONDITION | | |
| --- | --- | --- | --- |
| DEGREE | DELTA | MODERATE | CONSTANT |
| 4 | $9.34454 \pm 1.45301$ | $13.4754 \pm 11.0877$ | $4.32330 \pm 5.31288$ |
| 20 | $3.60415 \pm 0.86643$ | $7.64168 \pm 2.60517$ | $0.404178 \pm 0.207159$ |
| 100 | $7.63891 \pm 2.85191$ | $1.03030 \pm 0.19394$ | $0.168201 \pm 0.176840$ |

Table 30: **Absolute error at the collocation point whose absolute error is *largest*.** The models are trained on the BHE. The error bars are the standard deviation over 10 training runs with different random seeds.

| | INITIAL CONDITION | | |
|---|---|---|---|
| DEGREE | DELTA | MODERATE | CONSTANT |
| 4 | $(2.16137 \pm 0.40053) \times 10^{-4}$ | $(2.89219 \pm 0.18509) \times 10^{-3}$ | $(2.97723 \pm 1.06194) \times 10^{-4}$ |
| 20 | $(1.16468 \pm 0.16788) \times 10^{-4}$ | $(4.13474 \pm 0.05627) \times 10^{-3}$ | $(2.51673 \pm 1.15930) \times 10^{-4}$ |
| 100 | $(2.41825 \pm 0.66709) \times 10^{-4}$ | $(4.64874 \pm 0.04121) \times 10^{-3}$ | $(2.55274 \pm 0.48728) \times 10^{-4}$ |

### H.9.2 First-order Functional Derivative: Delta Initial Condition & Relative Error

The relative errors of the first-order functional derivative are provided. The models are trained under the delta initial condition.

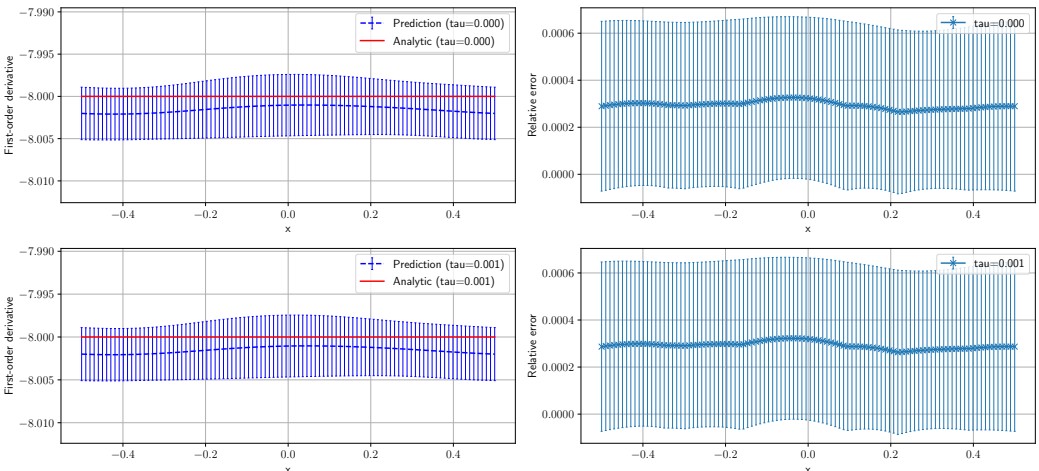

Figure 60: Degree 4.

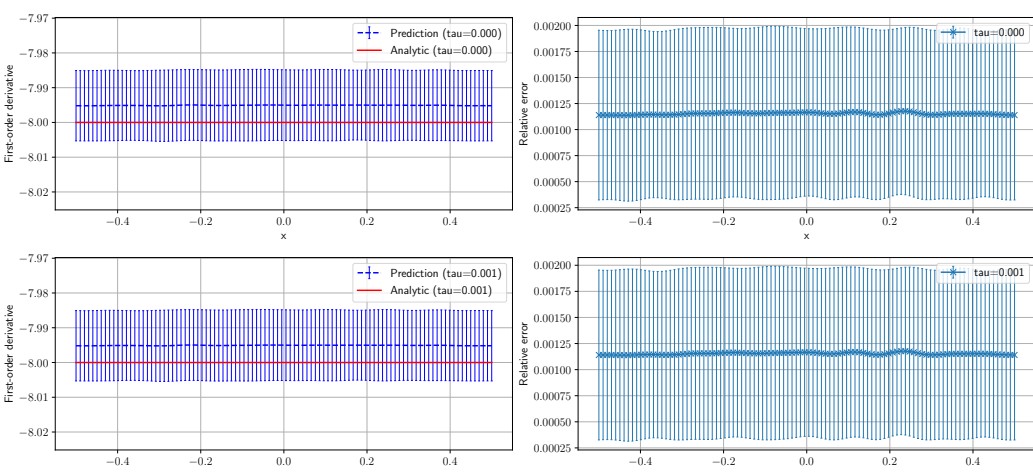

Figure 61: Degree 20.

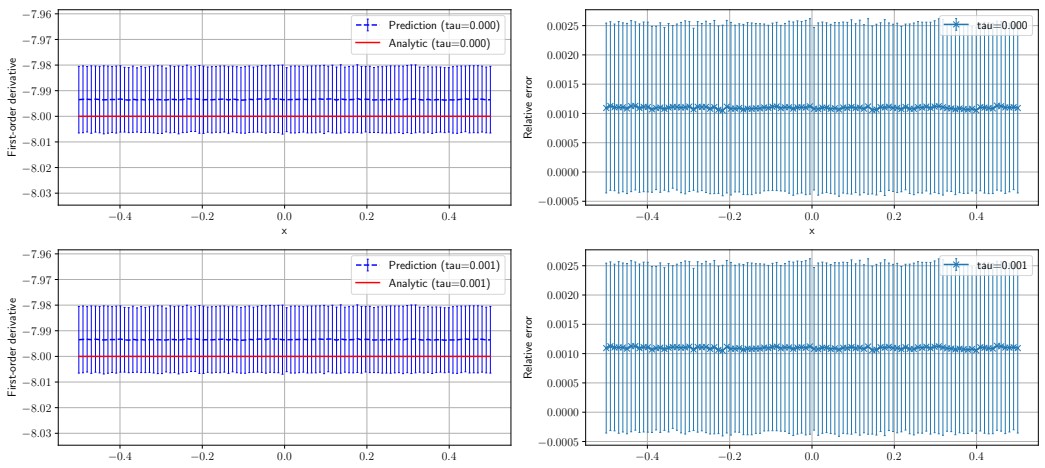

Figure 62: Degree 100.

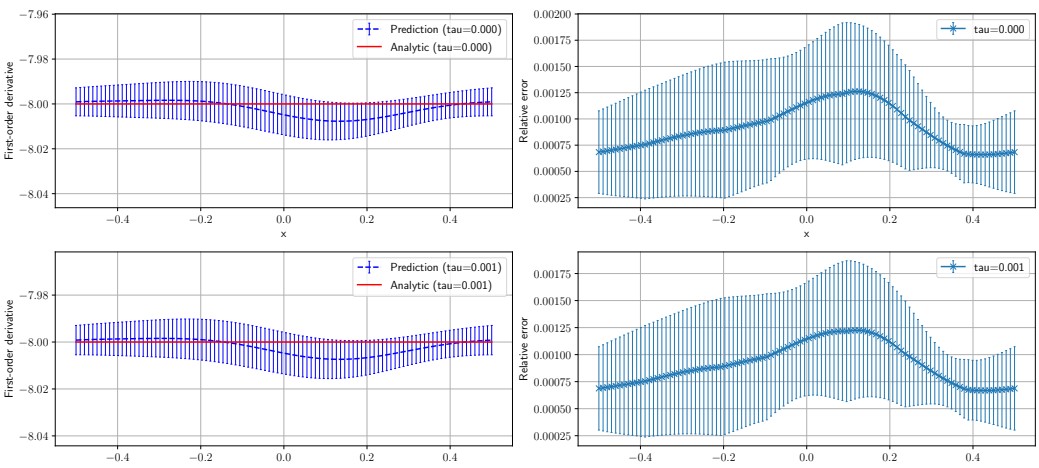

Figure 63: Degree 4. $W([0], \tau) = 0$ is included in the loss function.

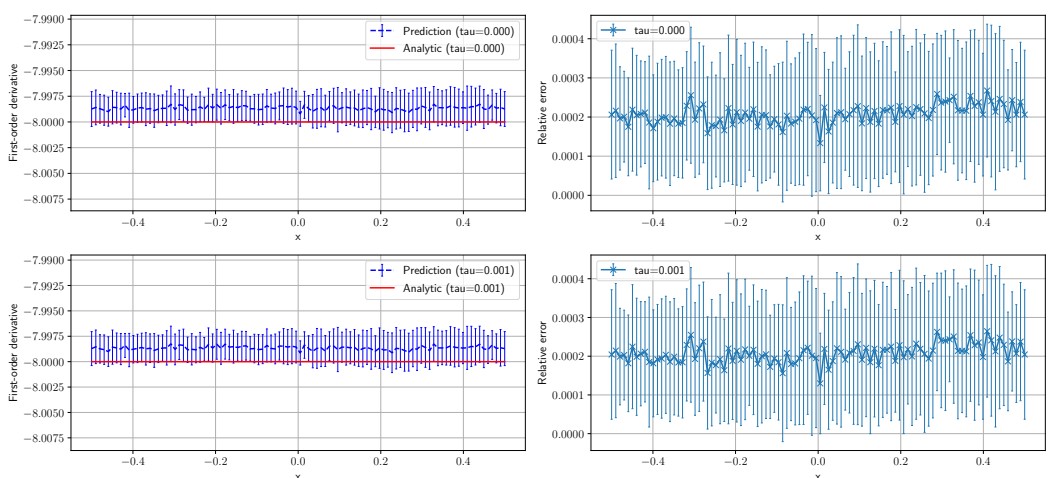

Figure 64: Degree 100. $W([0], \tau) = 0$ is included in the loss function.

### H.9.3 First-order Functional Derivative: Constant Initial Condition & Relative Error

The relative errors of the first-order functional derivative are provided. The models are trained under the constant initial condition.

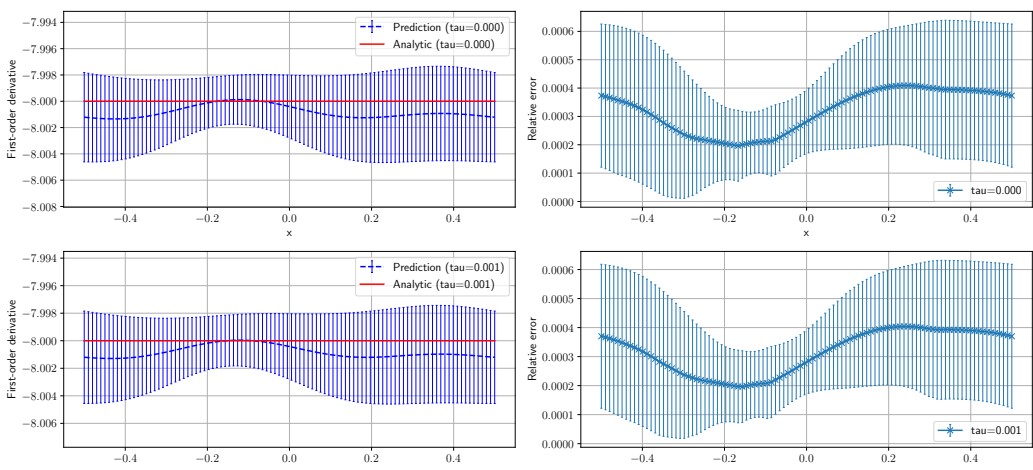

Figure 65: Degree 4.

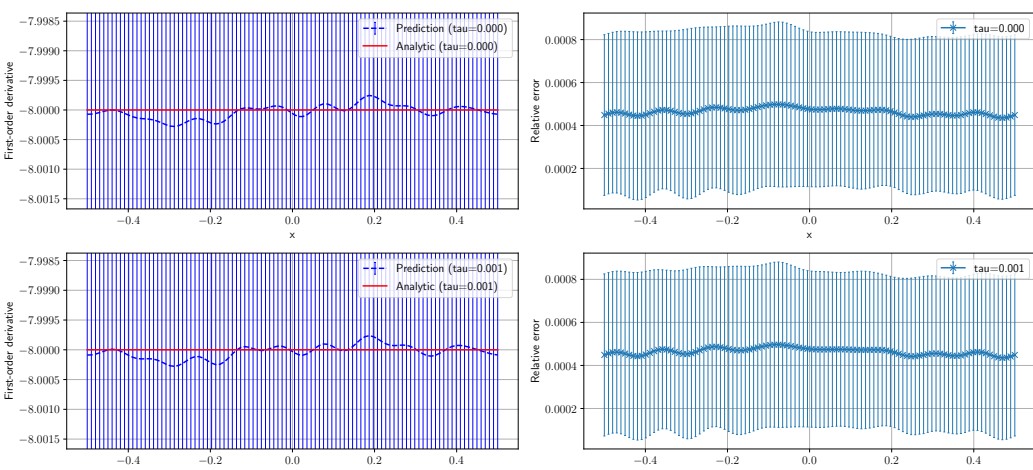

Figure 66: Degree 20.

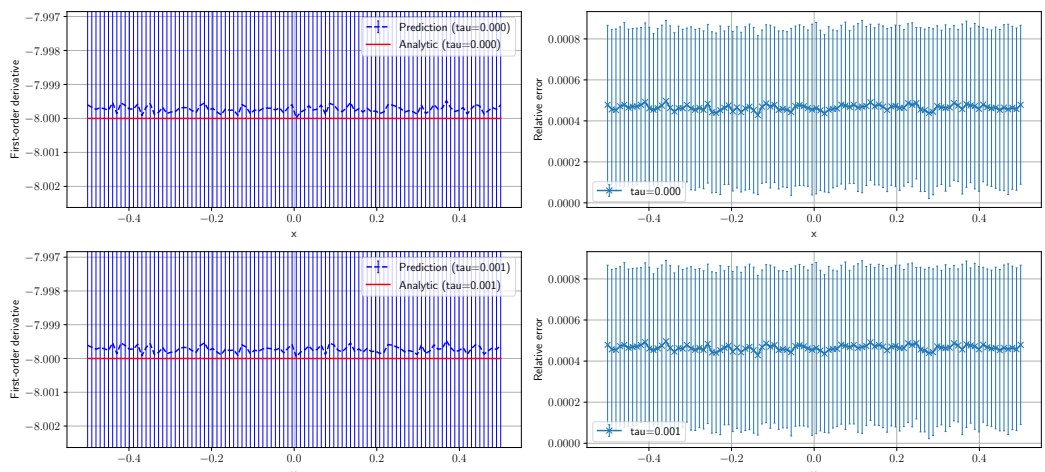

Figure 67: Degree 100.

### H.9.4 First-order Functional Derivative: Moderate Initial Condition & Relative Error

The relative errors of the first-order functional derivative are provided. The models are trained under the moderate initial condition.

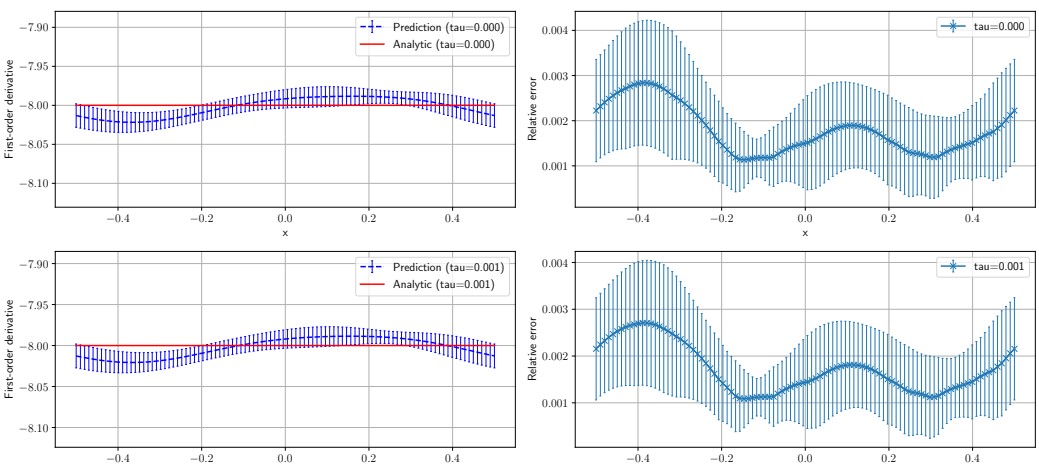

Figure 68: Degree 4.

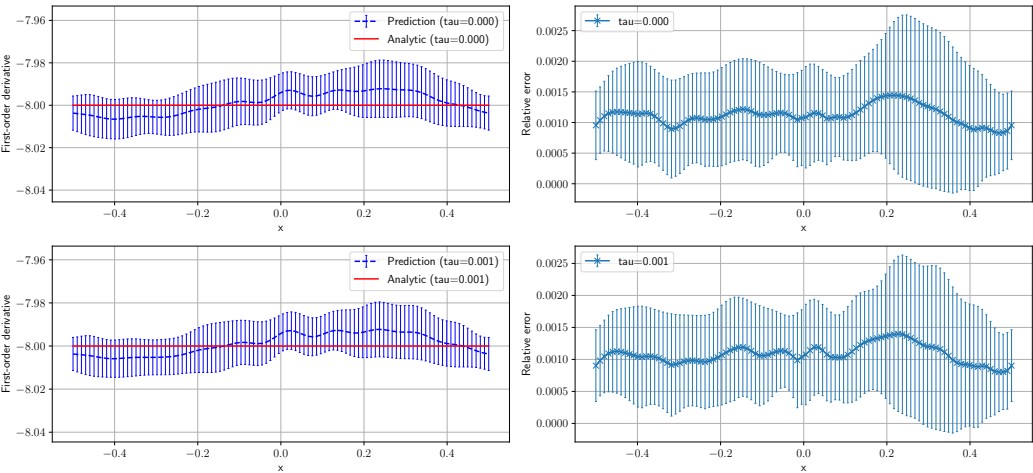

Figure 69: Degree 20.

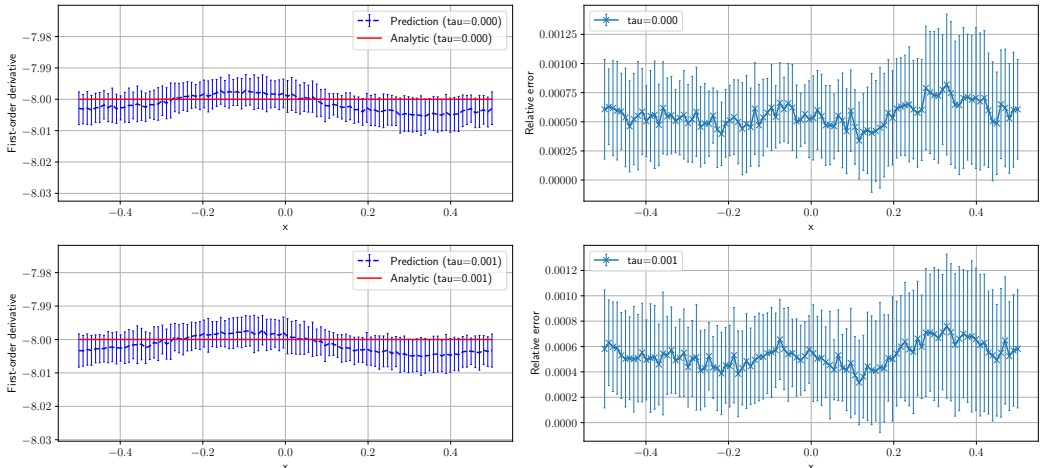

Figure 70: Degree 100.

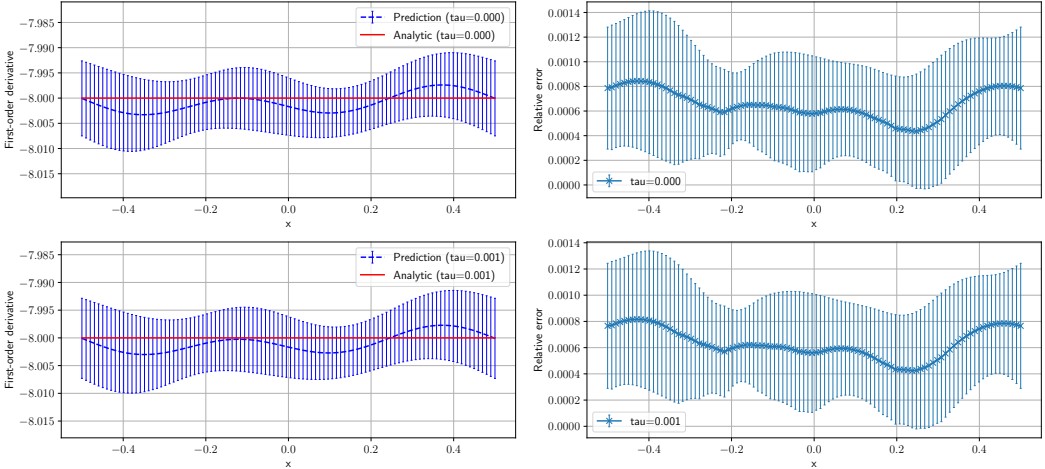

Figure 71: Degree 4. $W([0], \tau) = 0$ is included in the loss function.

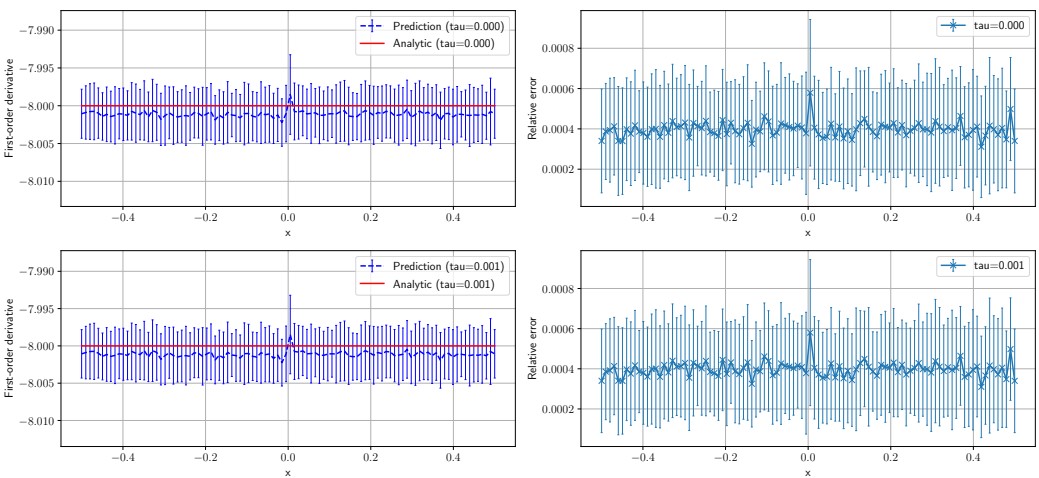

Figure 72: Degree 100. $W([0], \tau) = 0$ is included in the loss function.

### H.9.5 First-order Functional Derivative: Delta Initial Condition & Absolute Error

The absolute errors of the first-order functional derivative are provided. The models are trained under the delta initial condition.

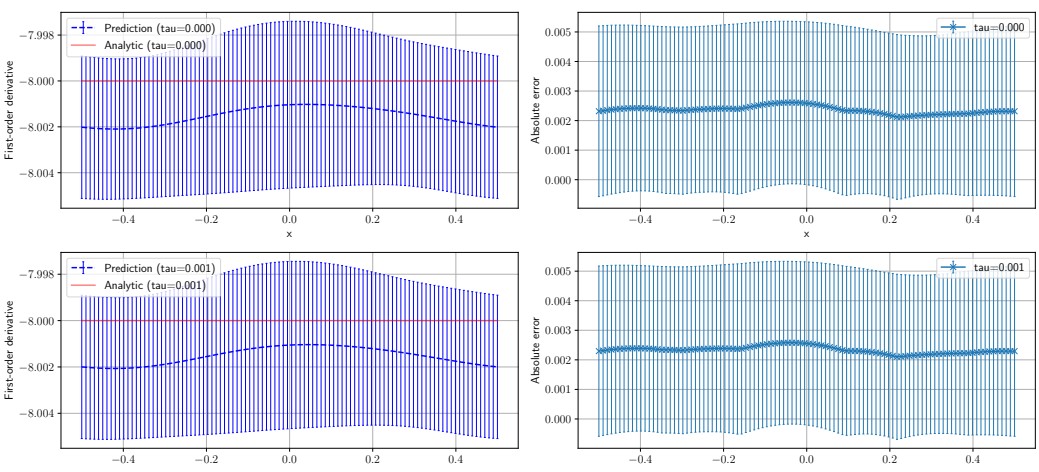

Figure 73: Degree 4.

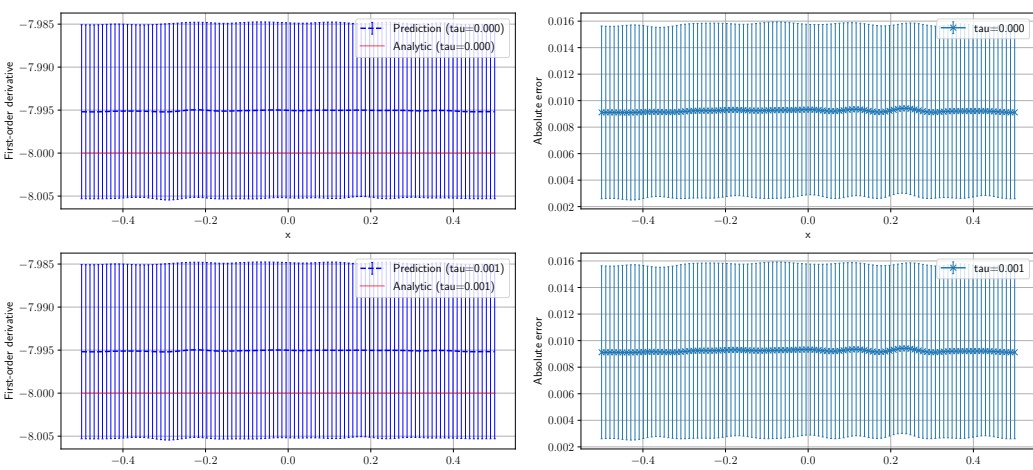

Figure 74: Degree 20.

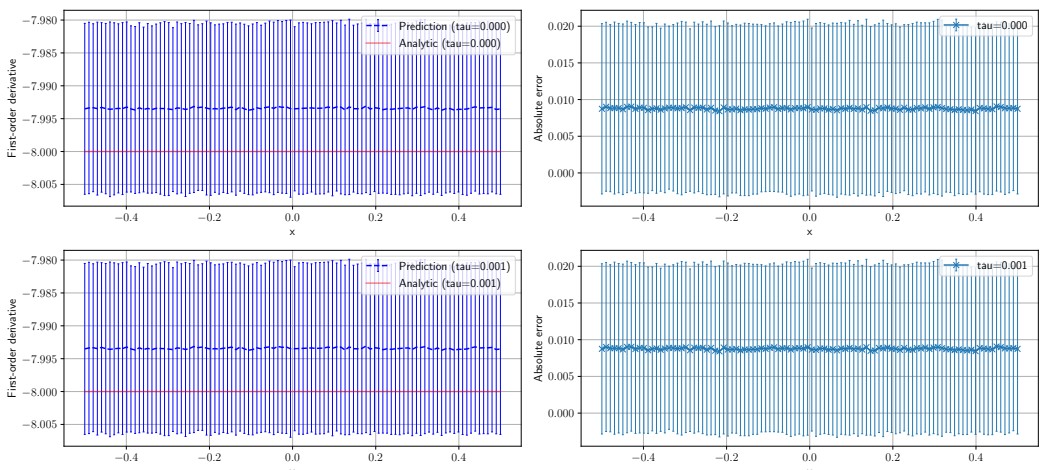

Figure 75: Degree 100.

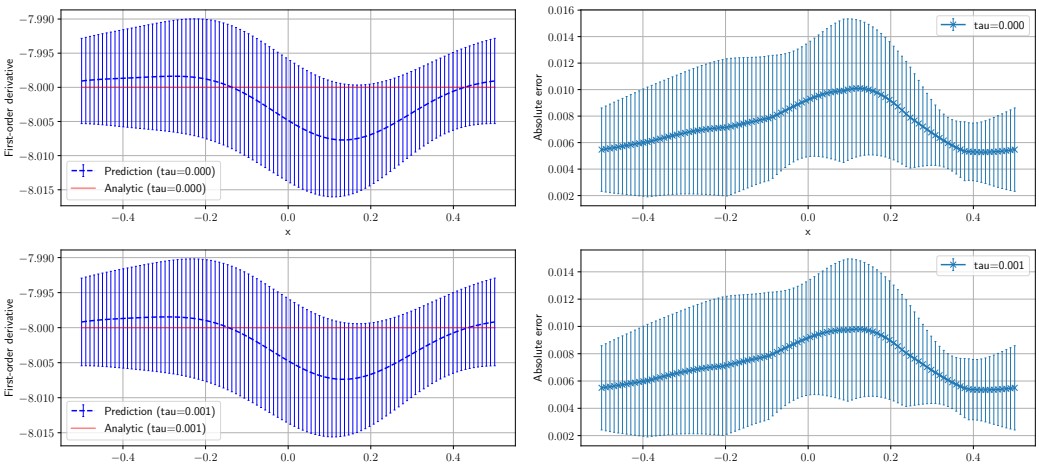

Figure 76: Degree 4. $W([0], \tau) = 0$ is included in the loss function.

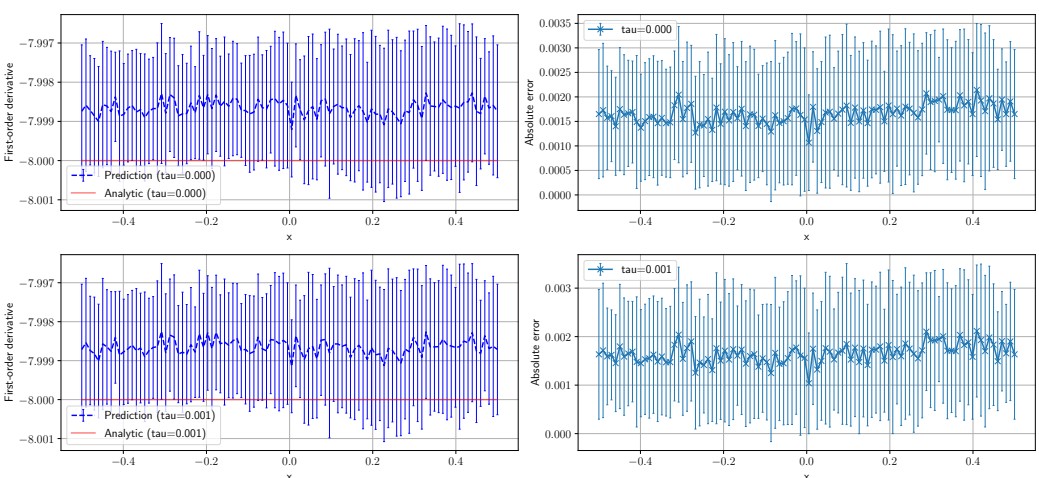

Figure 77: Degree 100. $W([0], \tau) = 0$ is included in the loss function.

### H.9.6 First-order Functional Derivative: Constant Initial Condition & Absolute Error

The absolute errors of the first-order functional derivative are provided. The models are trained under the constant initial condition.

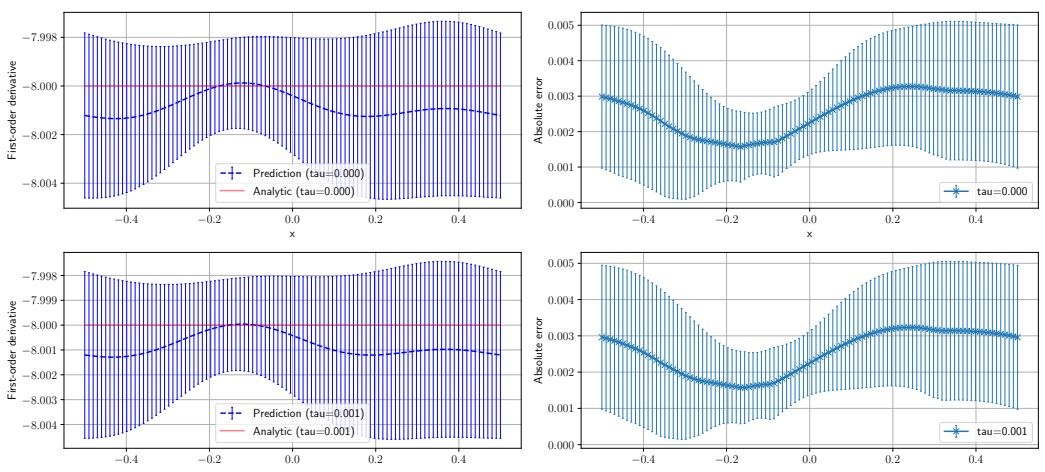

Figure 78: Degree 4.

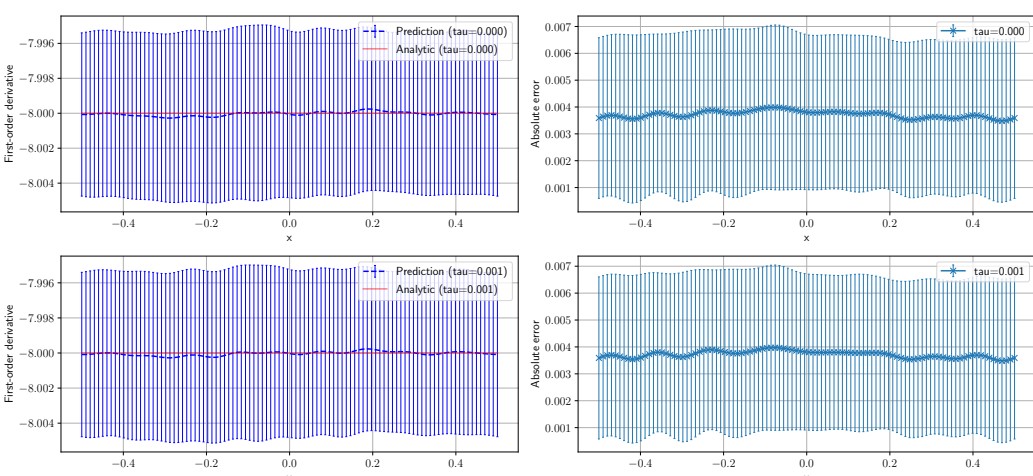

Figure 79: Degree 20.

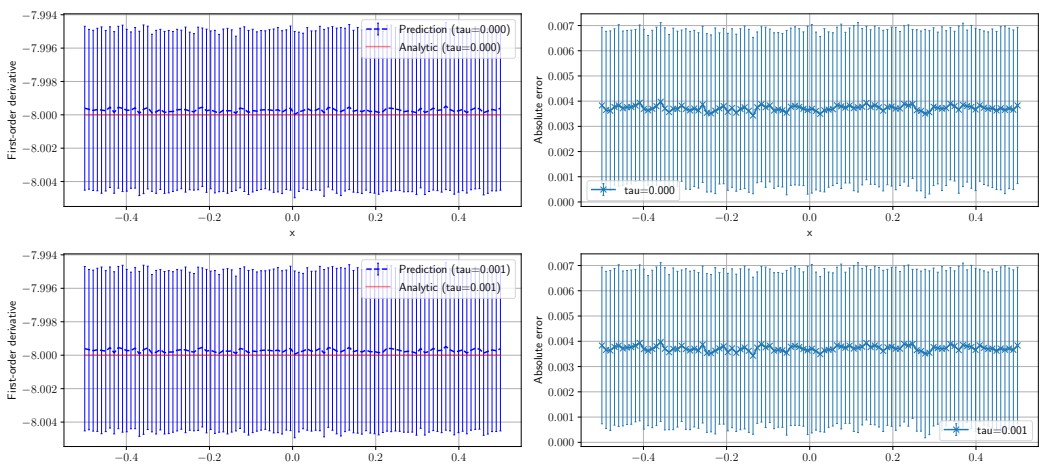

Figure 80: Degree 100.

### H.9.7 First-order Functional Derivative: Moderate Initial Condition & Absolute Error

The absolute errors of the first-order functional derivative are provided. The models are trained under the moderate initial condition.

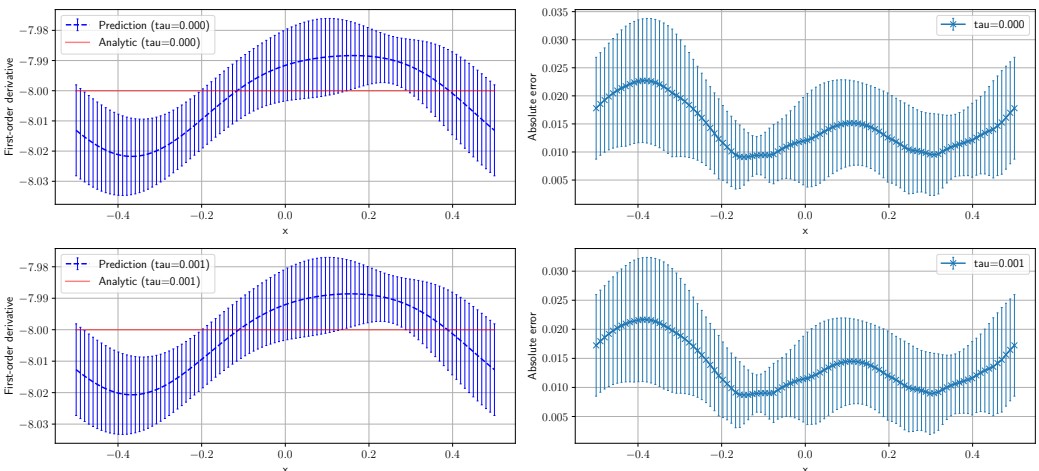

Figure 81: Degree 4.

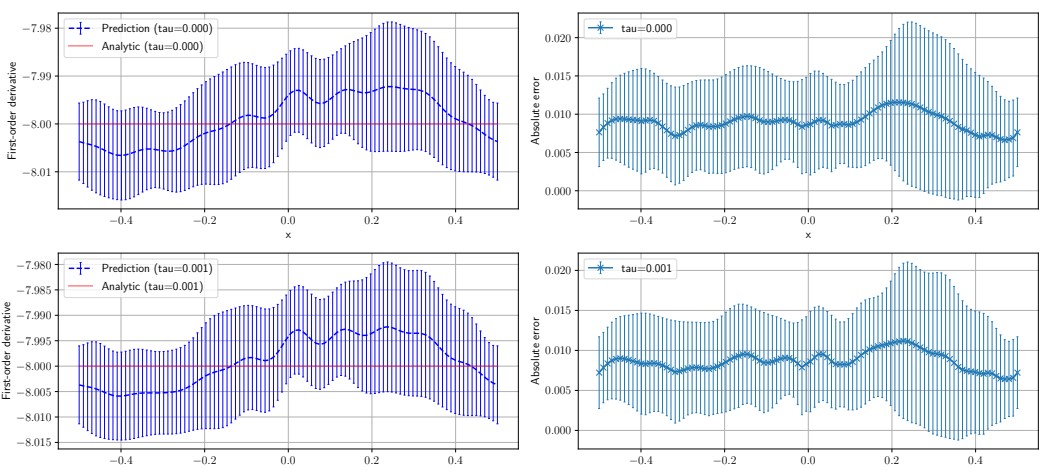

Figure 82: Degree 20.

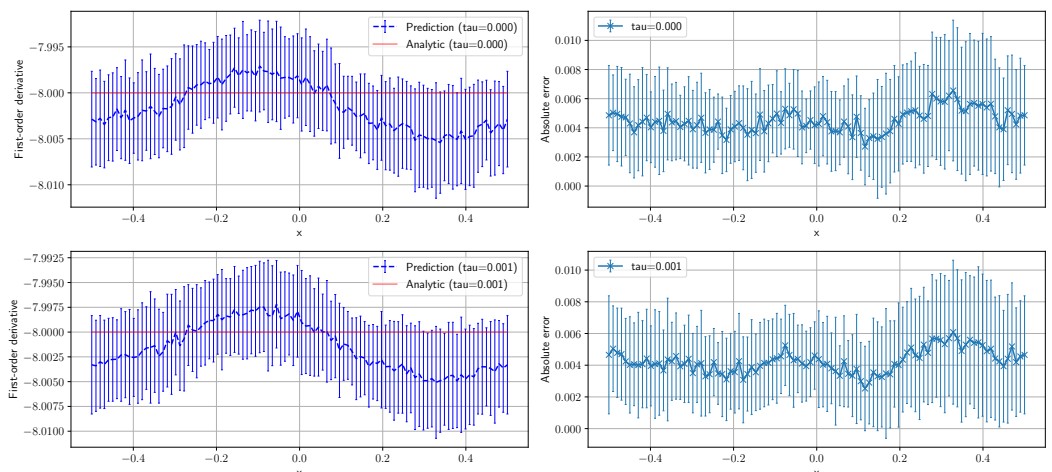

Figure 83: Degree 100.

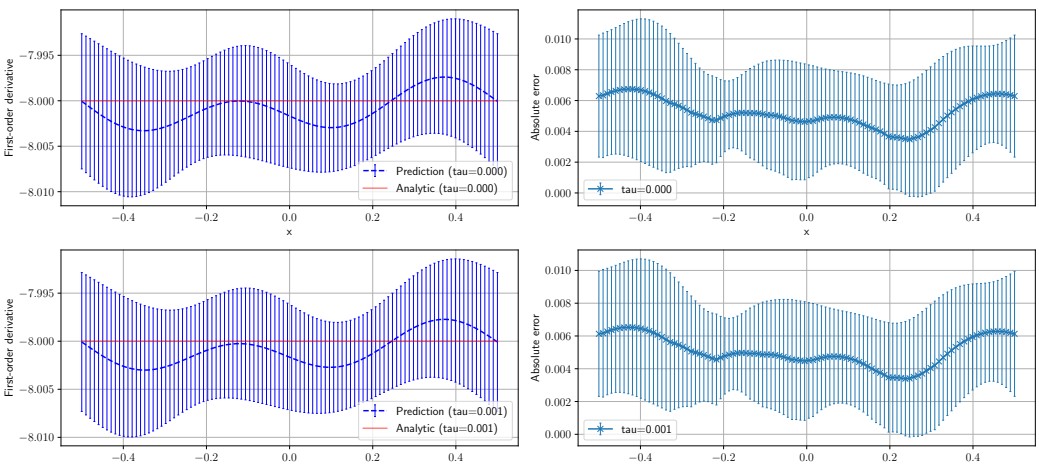

Figure 84: Degree 4. $W([0], \tau) = 0$ is included in the loss function.

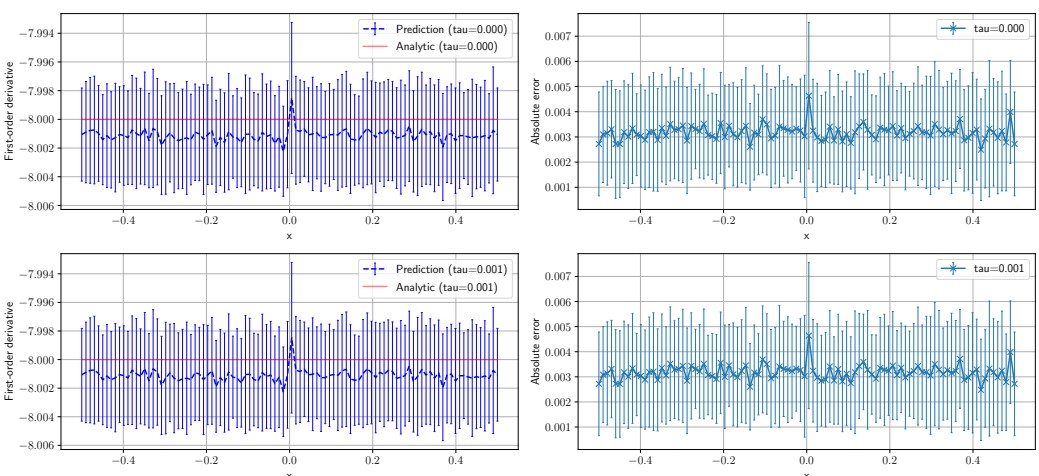

Figure 85: Degree 100. $W([0], \tau) = 0$ is included in the loss function.

### H.9.8 Time-dependent Relative and Absolute Errors at $\theta = 0$: Delta Initial Condition

The time-dependent relative and absolute errors at $\theta = 0$ are provided. The models are trained under the delta initial condition.

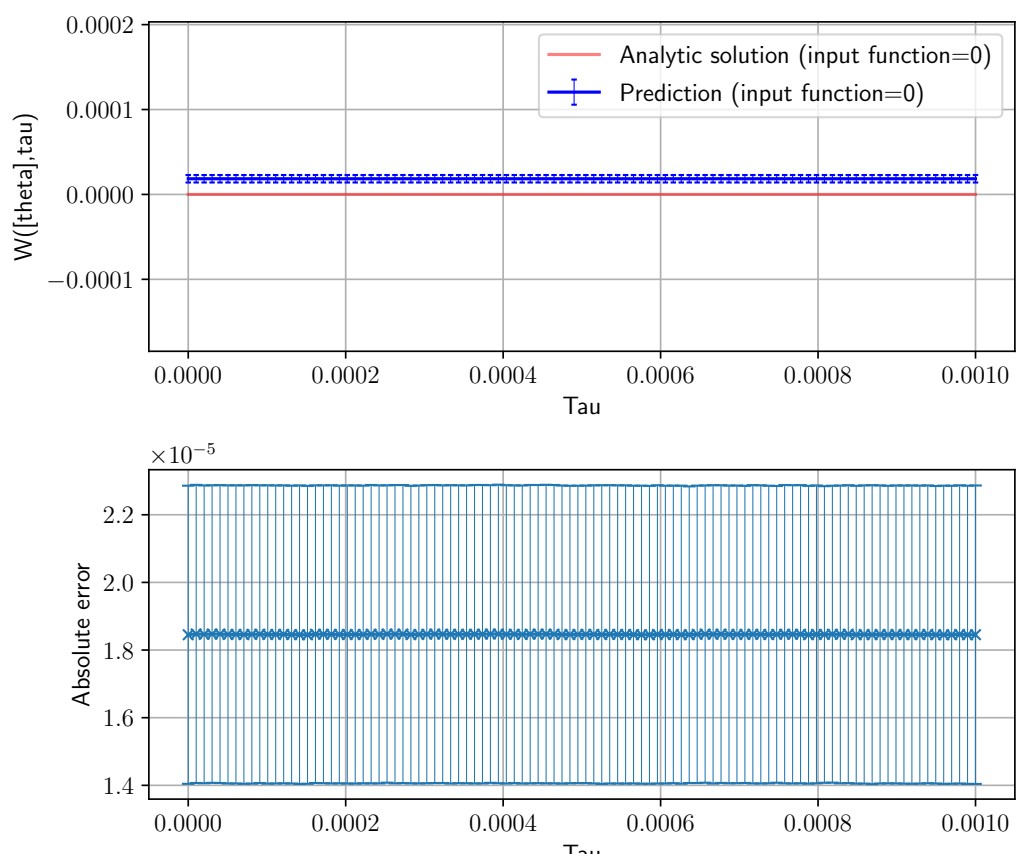

Figure 86: Degree 4.

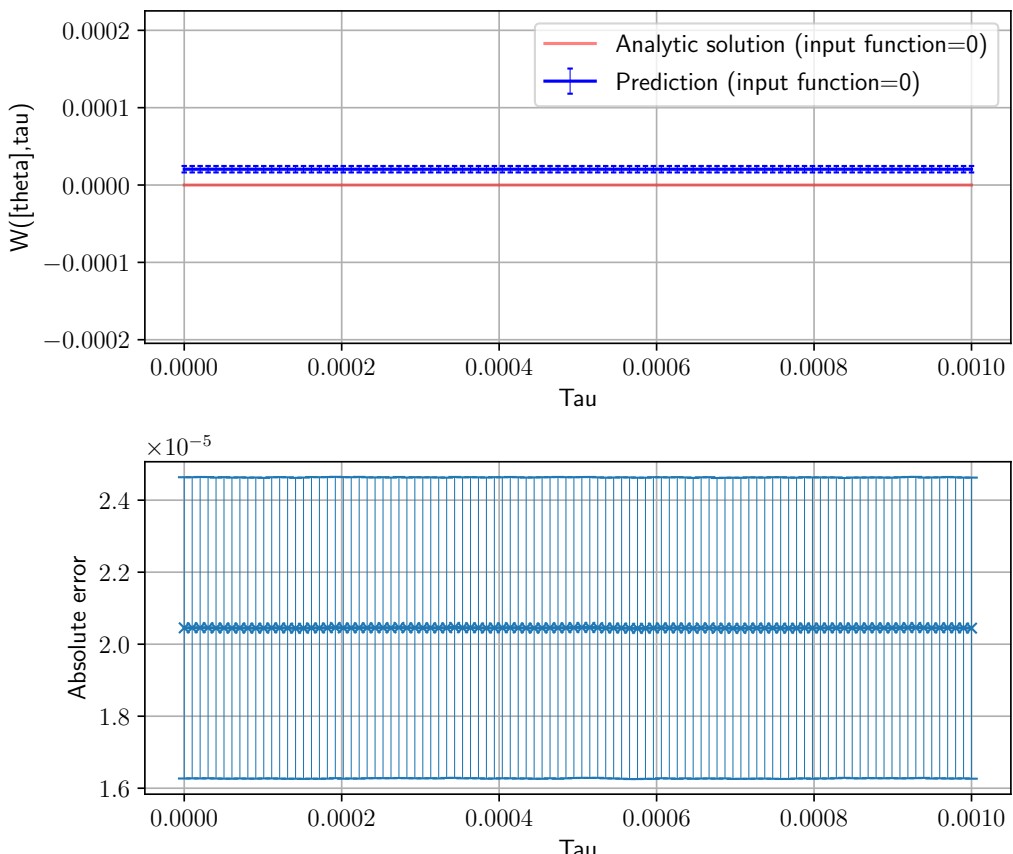

Figure 87: Degree 20.

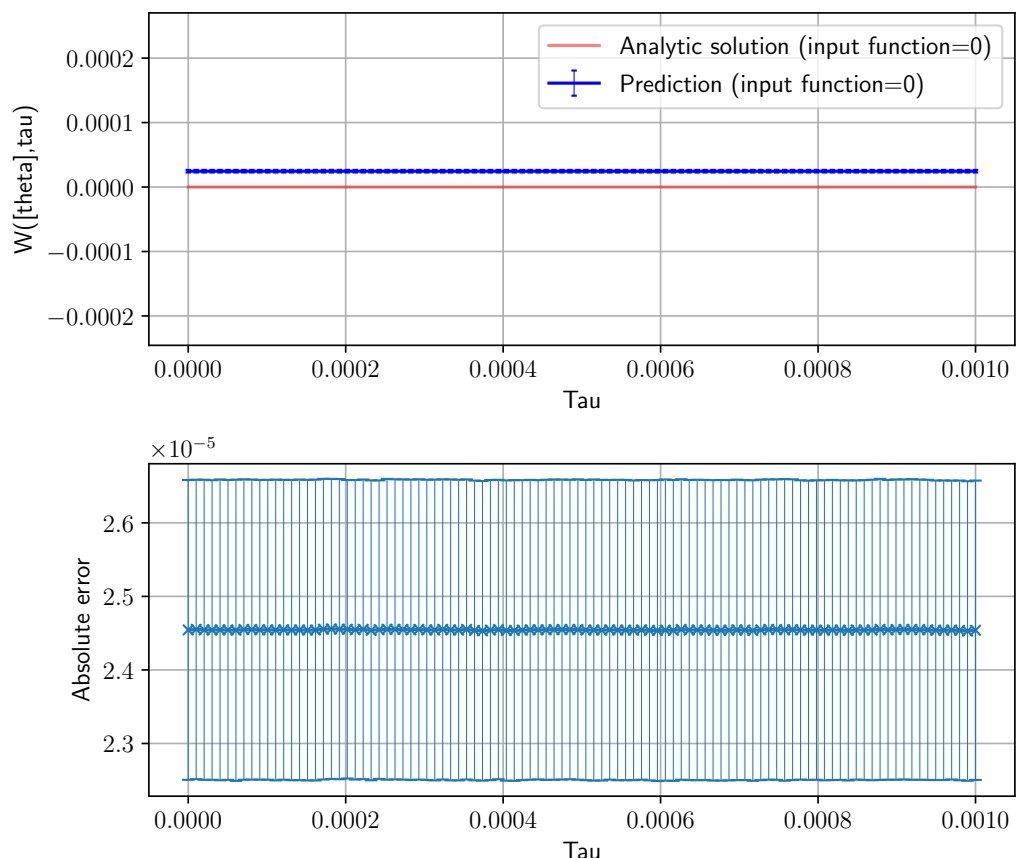

Figure 88: Degree 100.

### H.9.9 Time-dependent Relative and Absolute Errors at $\theta = 0$: Constant Initial Condition

The time-dependent relative and absolute errors at $\theta = 0$ are provided. The models are trained under the constant initial condition.

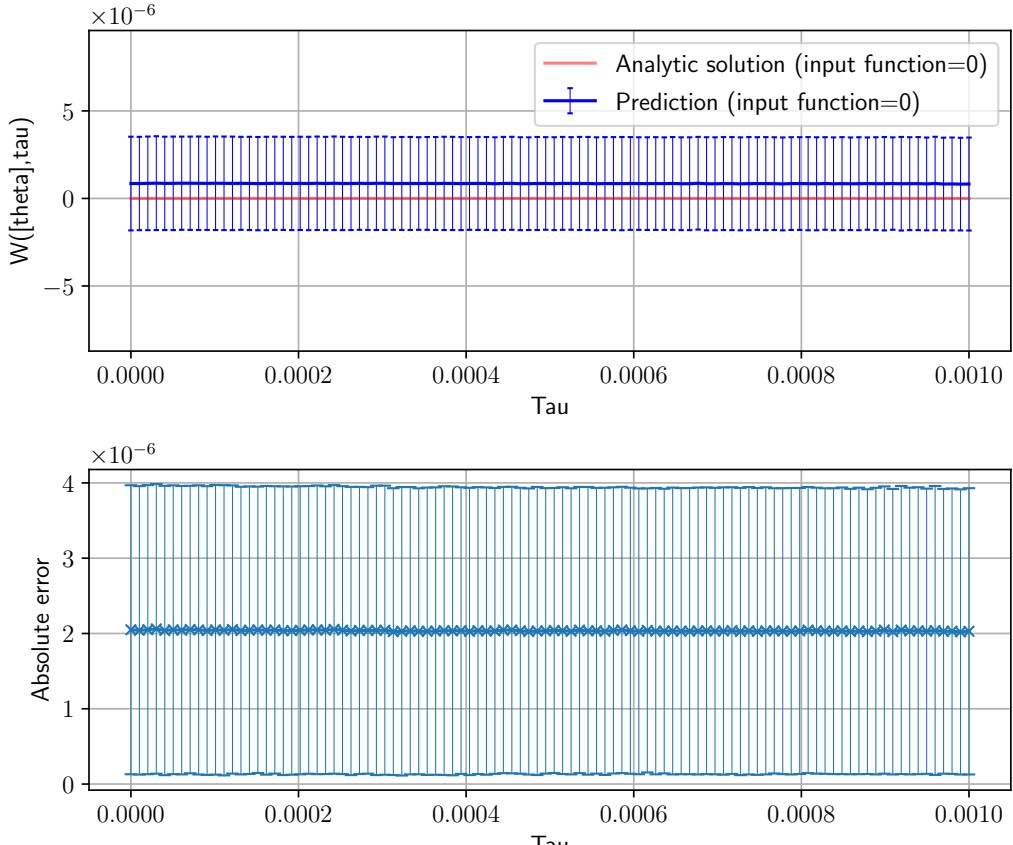

Figure 89: Degree 4.

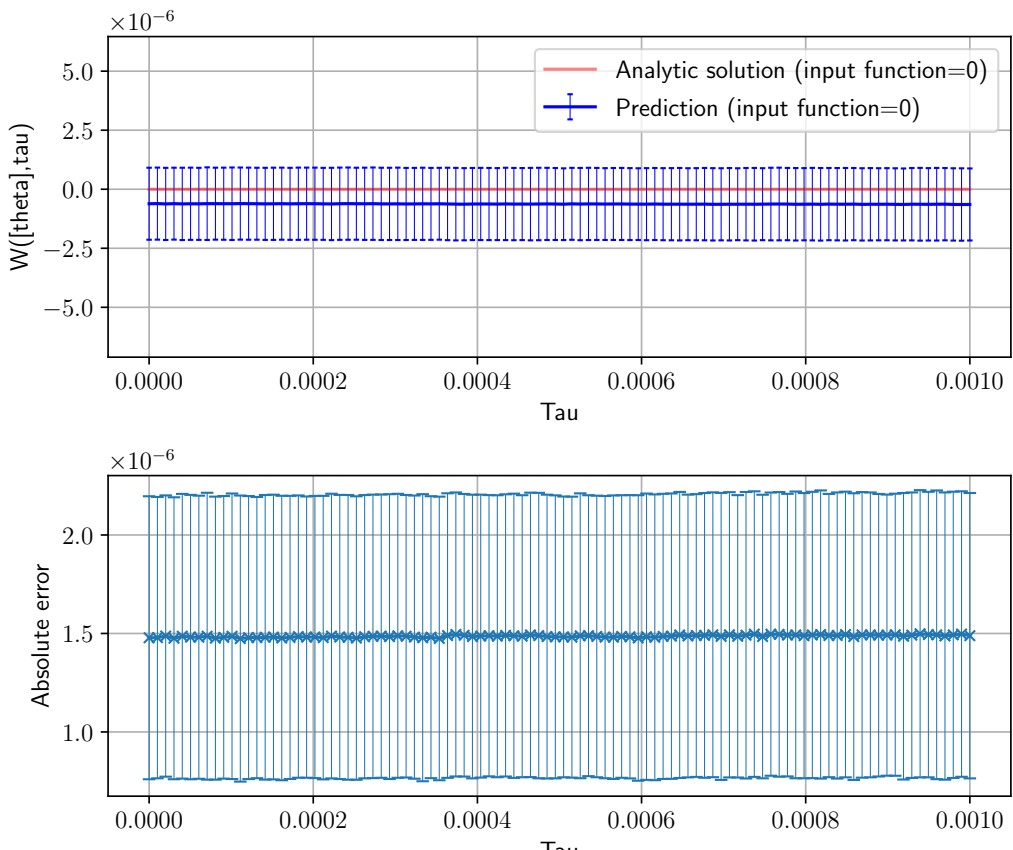

Figure 90: Degree 20.

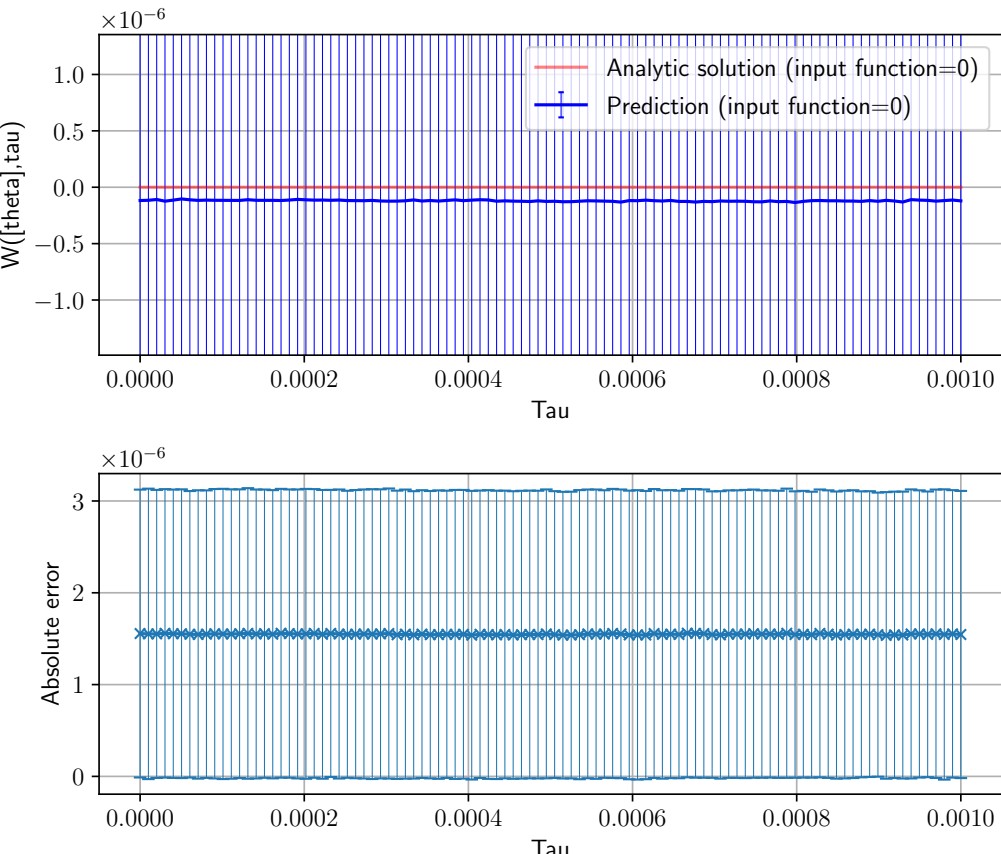

Figure 91: Degree 100.

## H.9.10 Time-dependent Relative and Absolute Errors at $\theta = 0$: Moderate Initial Condition

The time-dependent relative and absolute errors at $\theta = 0$ are provided. The models are trained under the moderate initial condition.

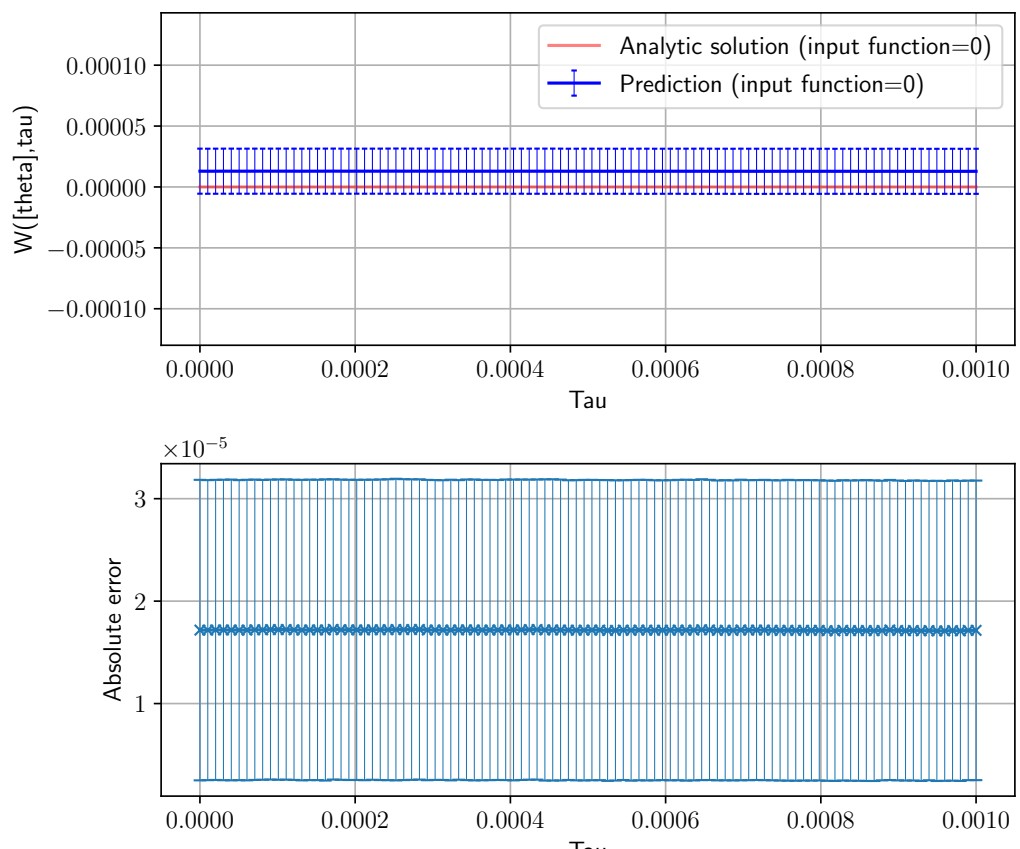

Figure 92: Degree 4.

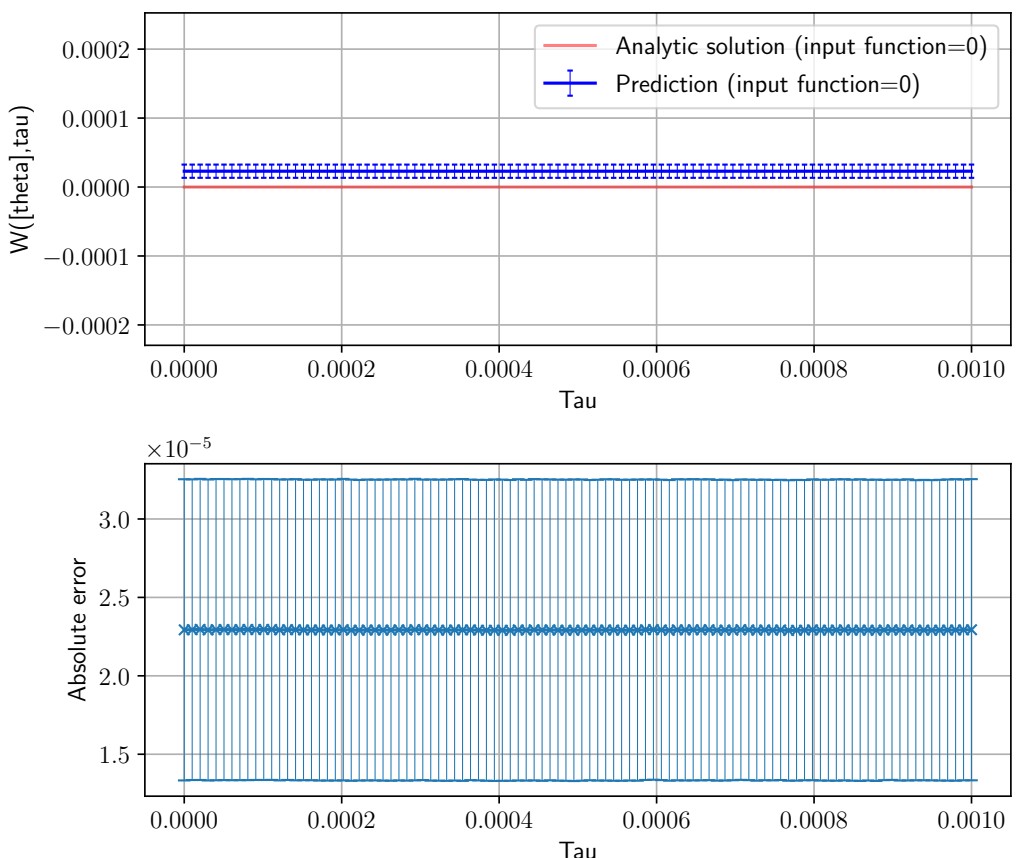

Figure 93: Degree 20.

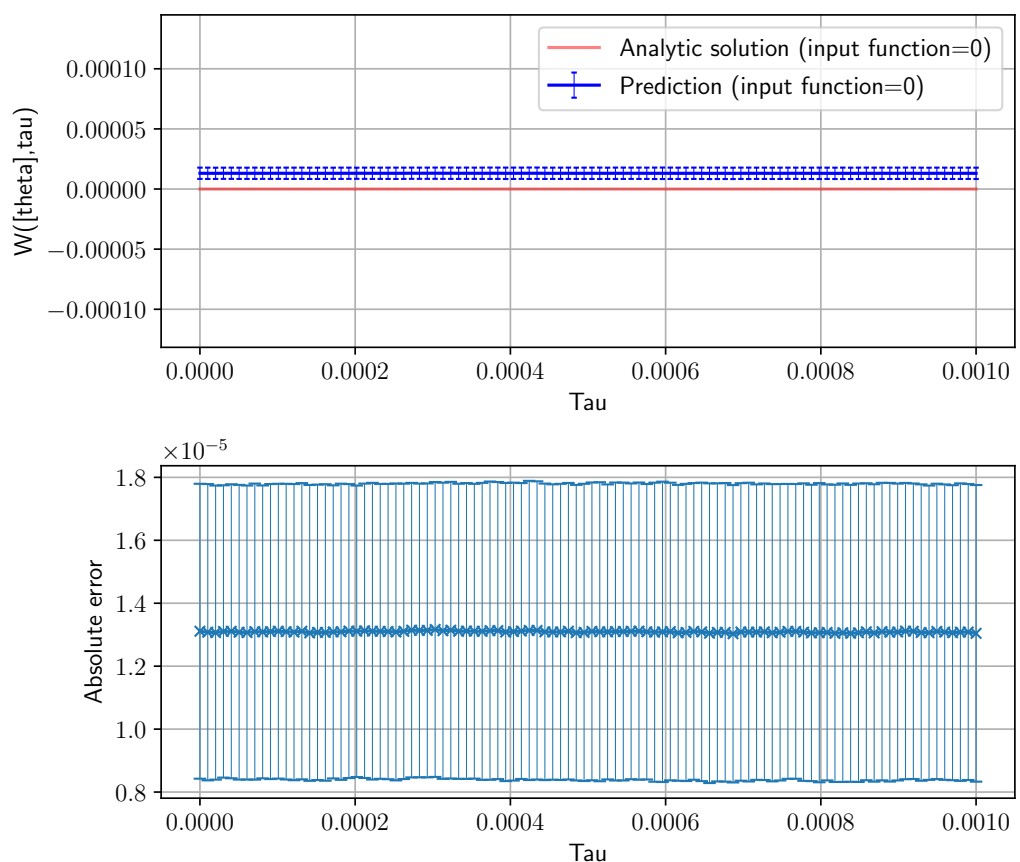

Figure 94: Degree 100.

### H.9.11 Time-averaged Pointwise Absolute Error: Delta Initial Condition

The time-averaged pointwise absolute errors are provided. The models are trained under the delta initial condition.

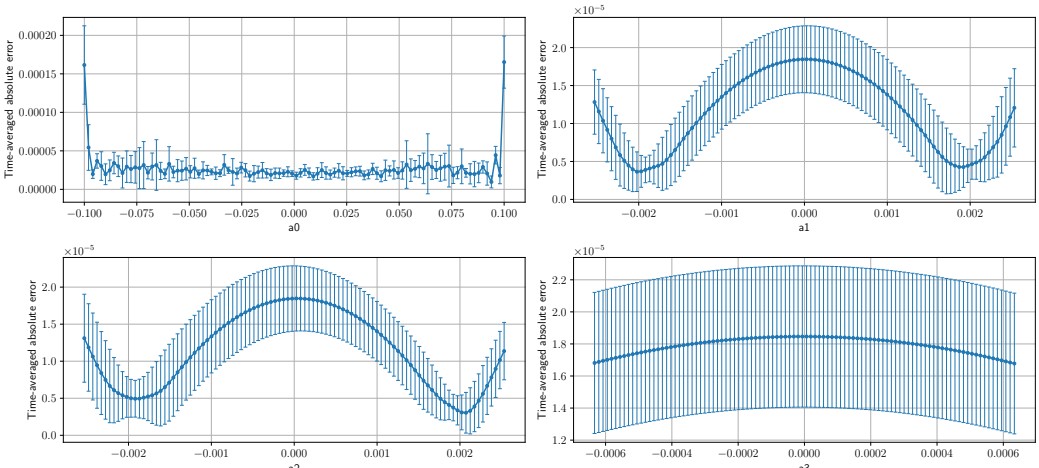

Figure 95: Degree 4.

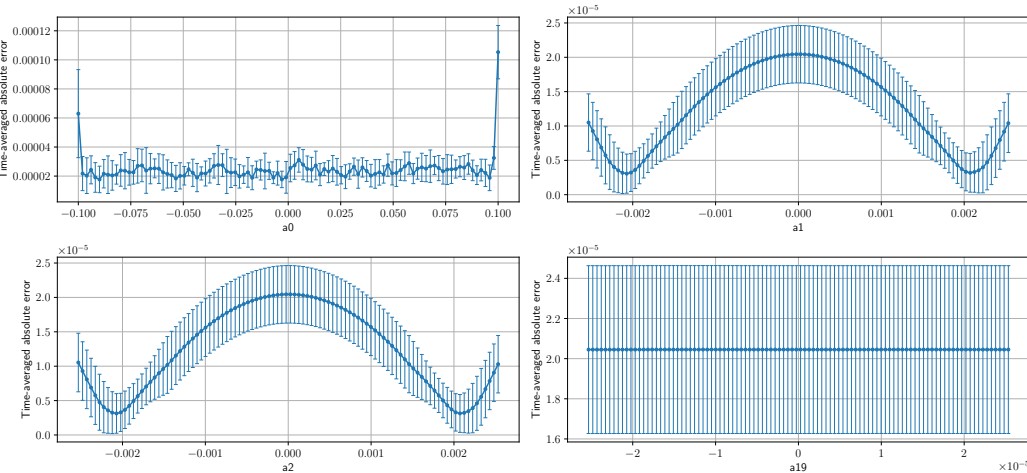

Figure 96: Degree 20.

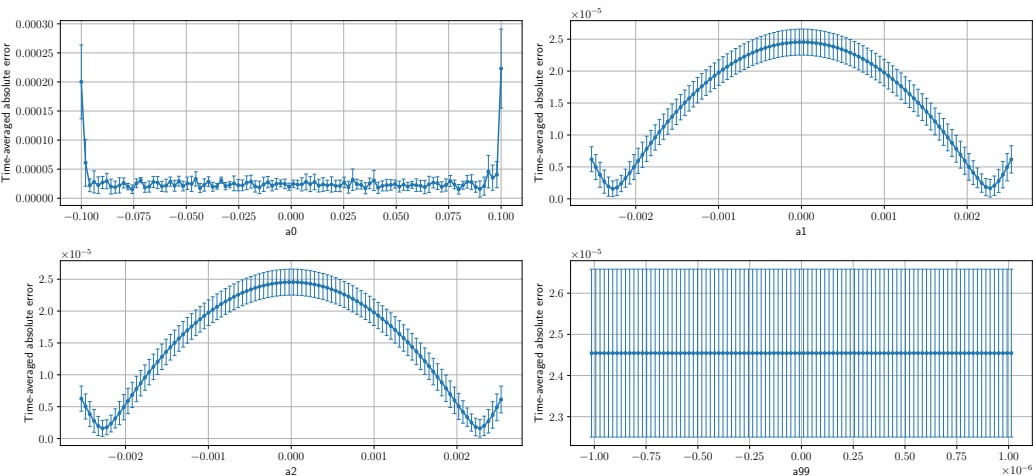

Figure 97: Degree 100.

### H.9.12 Time-averaged Pointwise Absolute Error: Constant Initial Condition

The time-averaged pointwise absolute errors are provided. The models are trained under the constant initial condition.

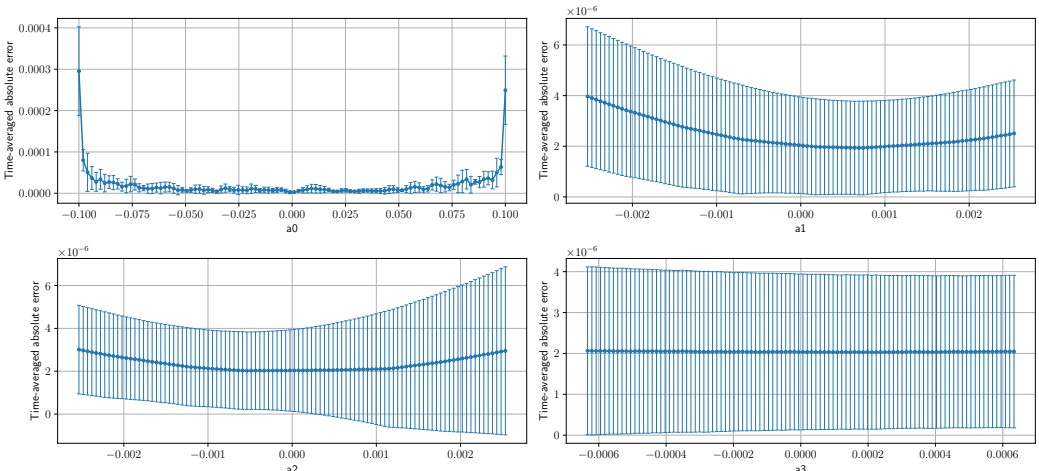

Figure 98: Degree 4.

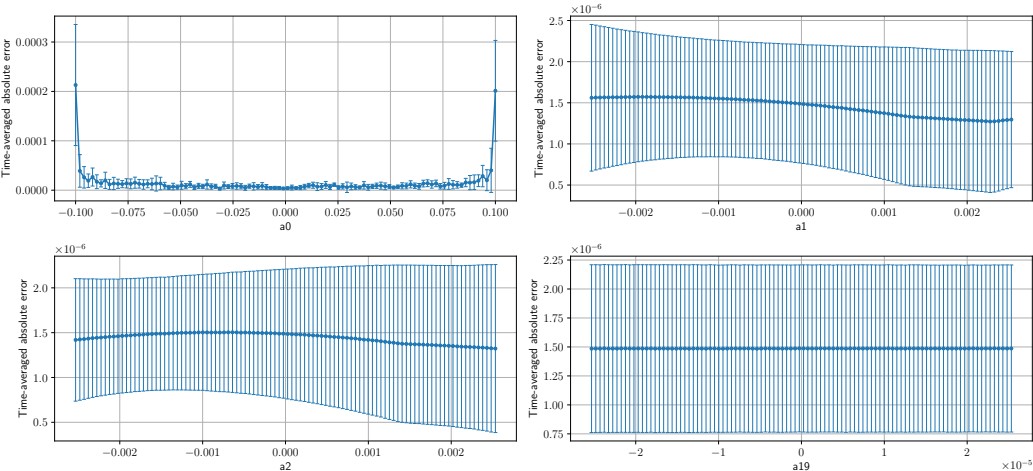

Figure 99: Degree 20.

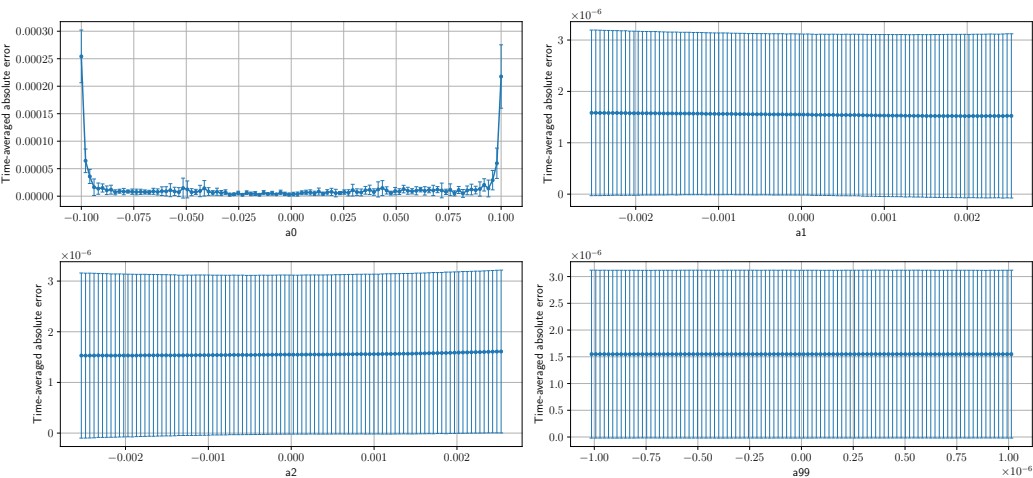

Figure 100: Degree 100.

## H.9.13    Time-averaged Pointwise Absolute Error: Moderate Initial Condition

The time-averaged pointwise absolute errors are provided. The models are trained under the moderate initial condition.

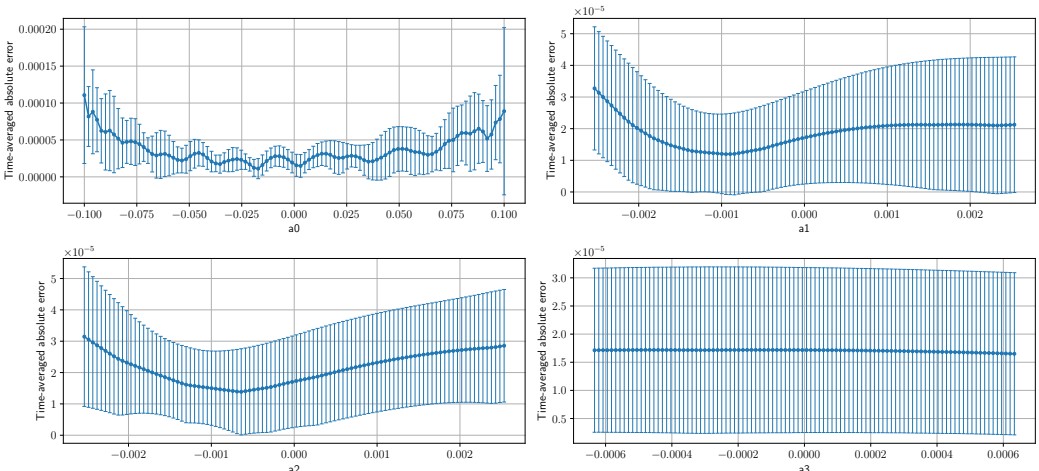

Figure 101: Degree 4.

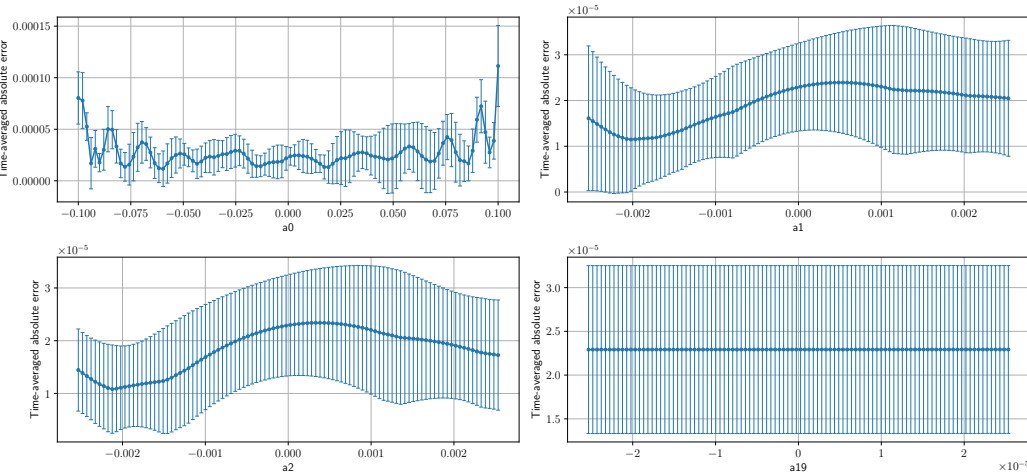

Figure 102: Degree 20.

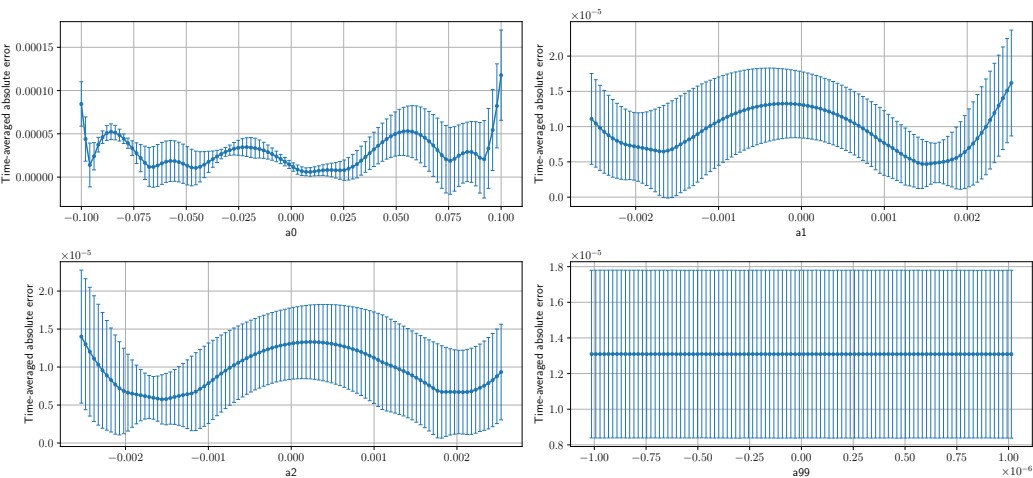

Figure 103: Degree 100.

### H.9.14 Time-averaged Pointwise Relative Error: Delta Initial Condition

The time-averaged pointwise relative errors are provided. The models are trained under the delta initial condition.

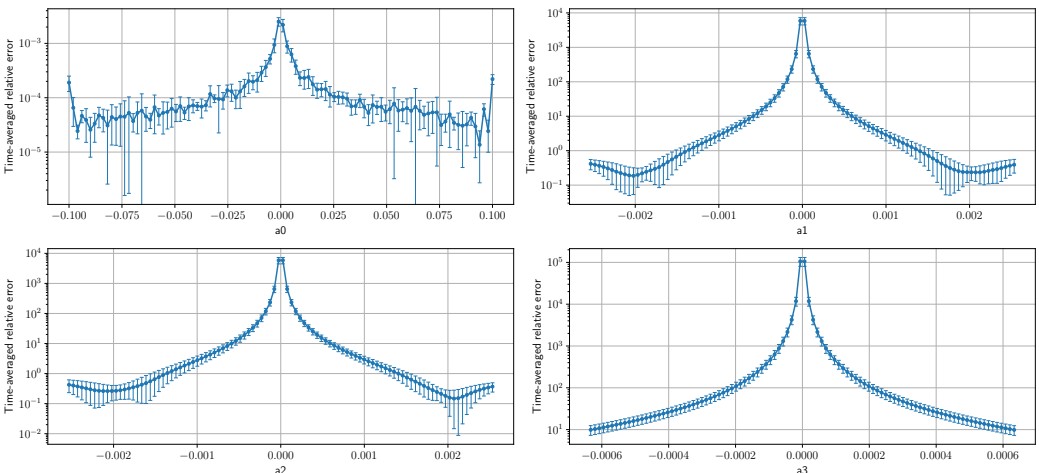

Figure 104: Degree 4.

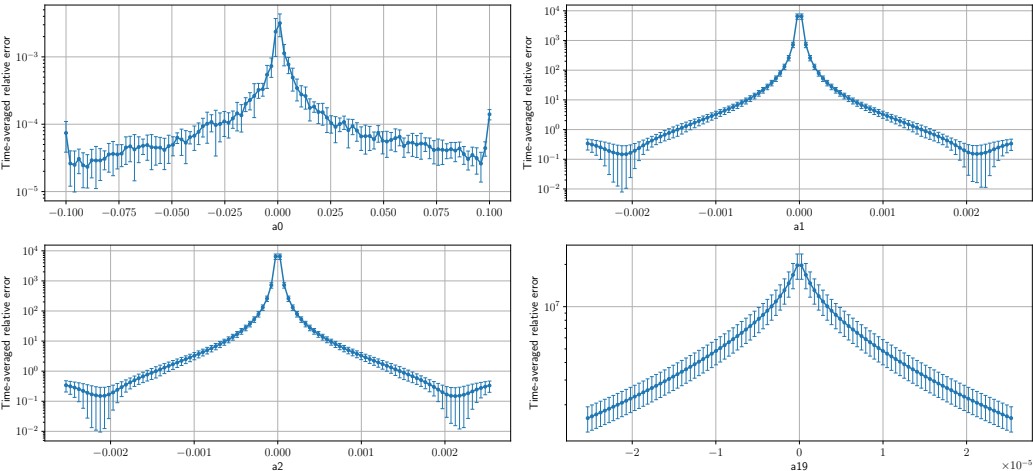

Figure 105: Degree 20.

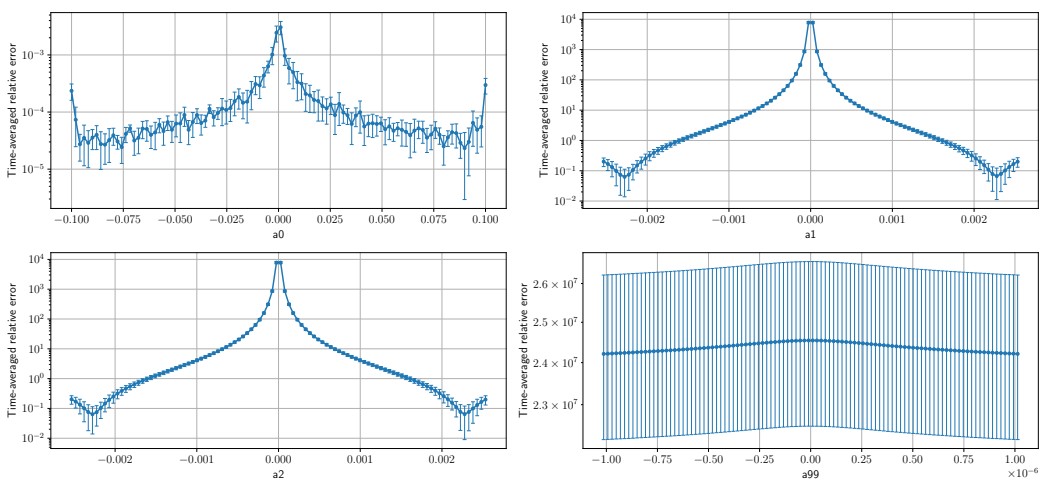

Figure 106: Degree 100.

### H.9.15 Time-averaged Pointwise Relative Error: Constant Initial Condition

The time-averaged pointwise relative errors are provided. The models are trained under the constant initial condition.

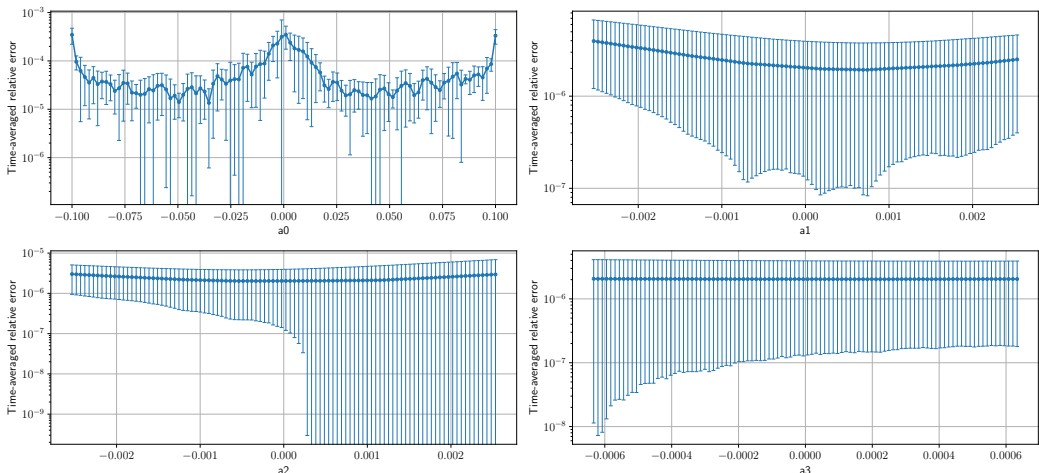

Figure 107: Degree 4.

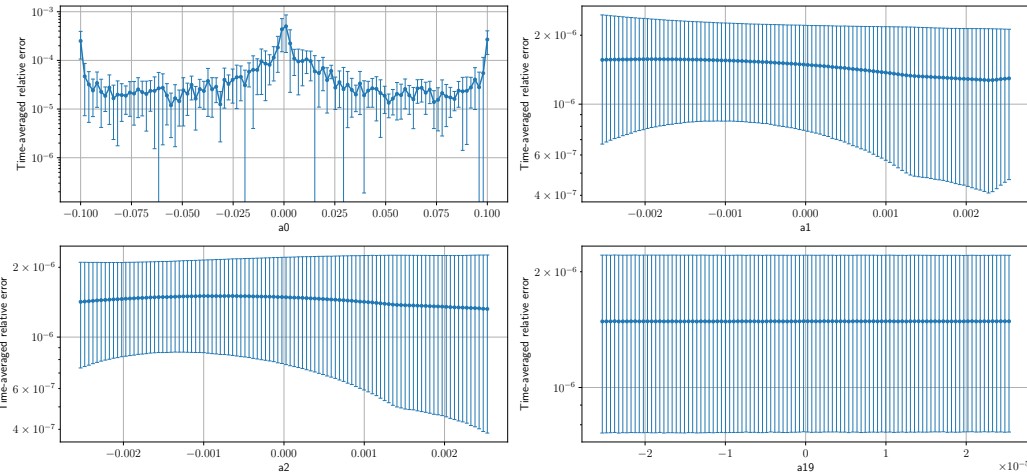

Figure 108: Degree 20.

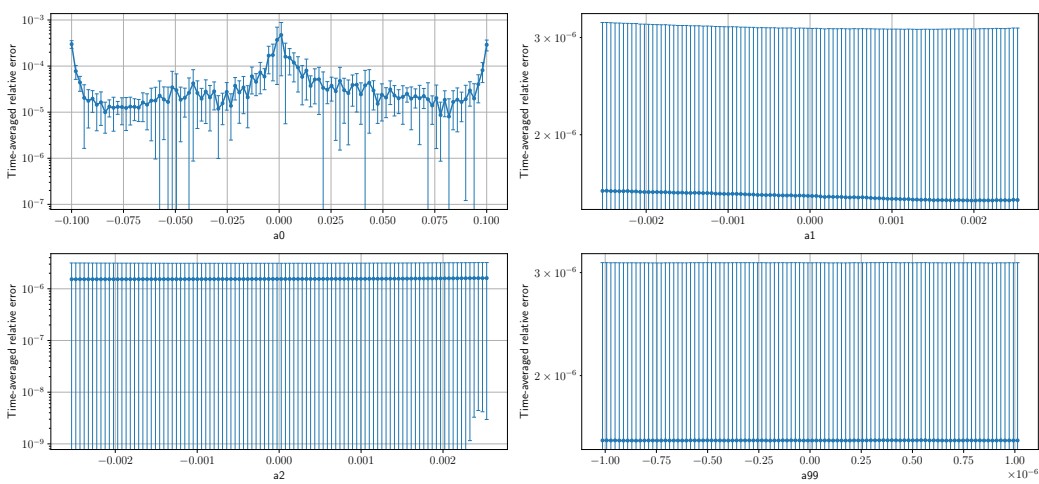

Figure 109: Degree 100.

### H.9.16 Time-averaged Pointwise Relative Error: Moderate Initial Condition

The time-averaged pointwise relative errors are provided. The models are trained under the moderate initial condition.

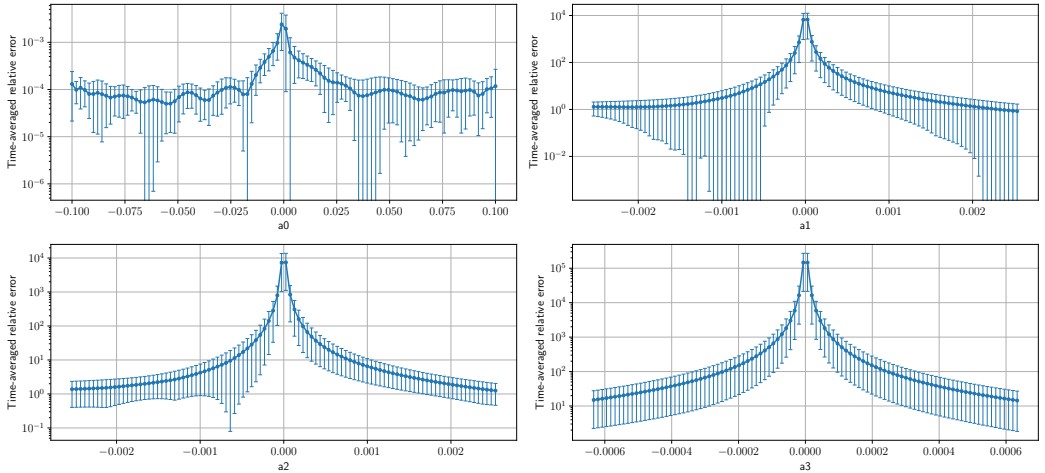

Figure 110: Degree 4.

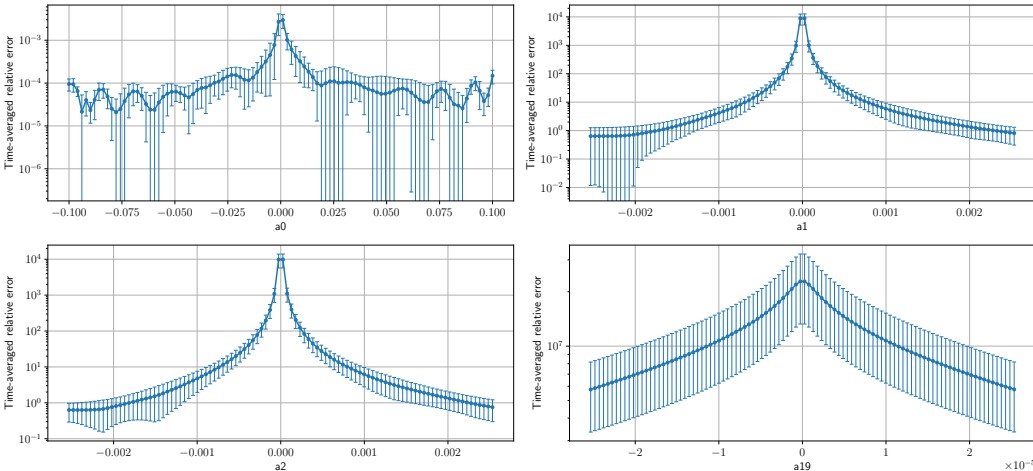

Figure 111: Degree 20.

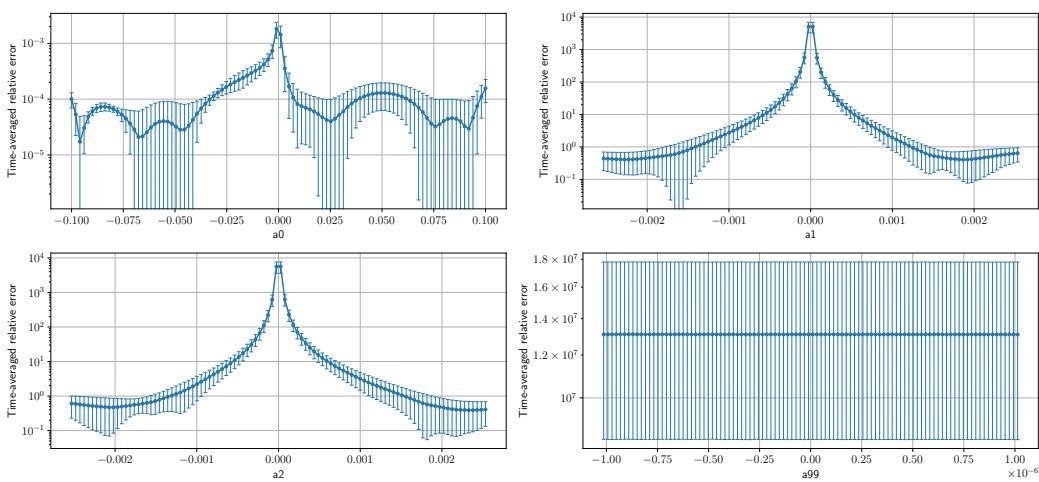

Figure 112: Degree 100.

