# OpenReview forum: "Physics-informed Neural Networks for Functional Differential Equations: Cylindrical Approximation and Its Convergence Guarantees"
_NeurIPS.cc/2024/Conference — NeurIPS 2024 poster_

### Official Review · Reviewer_33cu · 2024-06-26

**Soundness:** 3
**Presentation:** 3
**Contribution:** 3
**Rating:** 6
**Confidence:** 3

**Summary:**

This paper considers solving FDEs (Functional Differential Equations) using neural networks. The difficulty of solving FDEs compared to PDEs (Partial Differential Equations) lies in the fact that the input space is infinite-dimensional. In this case, the author employs Cylindrical Approximation to reduce the infinite-dimensional space to a finite-dimensional space. Then reduce the problem to a PDE case and use PINN to solve it.

**Strengths:**

Consider the challenges of utilizing neural networks in infinite-dimensional spaces. The author's use of Cylindrical Approximation as a method to reduce the infinite-dimensional space to a finite-dimensional one is commendable. The results presented in the paper are interesting and well-written.

**Weaknesses:**

I did not find the main issue of the paper addressed. My concern is that the theoretical part of this paper is somewhat weak. The theorems primarily present asymptotic results, i.e., asking for \(m \to \infty\). In my knowledge, there are some papers considering the functional approximation such as:
1. T. Chen and H. Chen. Approximations of continuous functionals by neural networks with application to dynamic systems. IEEE Transactions on Neural Networks, 4(6):910–918, 1993.
 2. Y. Yang and Y. Xiang. Approximation of Functionals by Neural Network without Curse of Dimensionality.
 3. L. Song et al. Approximation of smooth functionals using deep ReLU networks.

Except for the first paper, the other two papers provide the approximation error of functionals. In my opinion, if you have Cylindrical Approximation rates and PINN approximation rates, the total order of approximation may also be obtainable. Are there any difficulties in obtaining the approximation rate?

**Questions:**

Mentioned in the Weakness.

**Limitations:**

All right.

---

> ### Author Rebuttal · Authors · 2024-08-01
>
> We appreciate Reviewer 33cu for their insightful comments and recognition of our paper's strength.
> Thank you for pointing out the related papers; we enjoyed reading them and will include them in the Related Work section.
> Please note that, just in case, their focus is functional approximation, not functional derivatives or solving FDEs.
>
> > In my opinion, if you have Cylindrical Approximation rates and PINN approximation rates, the total order of approximation may also be obtainable. Are there any difficulties in obtaining the approximation rate?
>
> This is a very interesting question, which we have been discussing in our current research.
> Our response is as follows:
>
> - "the total order of approximation may also be obtainable"
>     - Yes.
>     - The total approximation error rate can be derived by combining the cylindrical approximation error (e.g., lines 1140-1145) with the PINN approximation error; specifically, $|F(\theta) - \hat{f}(\boldsymbol{a})| \leq | F(\theta) - f(\boldsymbol{a}) | + | f(\boldsymbol{a}) - \hat{f}(\boldsymbol{a}) |$, where $F(\theta)$ is the target functional with input $\theta$, $f(\boldsymbol{a})$ is its cylindrical approximation, and $\hat{f}(\boldsymbol{a})$ is the prediction of a PINN after training. Here, $| F(\theta) - f(\boldsymbol{a}) |$ represents the cylindrical approximation error we evaluated in our work, and $| f(\boldsymbol{a}) - \hat{f}(\boldsymbol{a}) |$ is the PINN approximation error. In light of the recent advances in the analysis of PINN approximability, at first glance, [ref1] appears a promising starting point because of its minimal assumptions on the form of PDEs.
>
> - "Are there any difficulties in obtaining the approximation rate?"
>     - Yes.
>     - The challenge is that it may be difficult to observe theoretical convergence of approximation error in experiment: if we successfully derive the combined cylindrical + PINN approximation error, it may be theoretically interesting but practically less relevant, because the optimization error of PINNs often overshadows the approximation error in experiments, as noted in our manuscript (e.g., App. H.6).
>
> Therefore, we conclude that a careful and detailed discussion on this point is required, which warrants a separate paper.
>
> [ref1] Ryck & Mishra. Generic bounds on the approximation error for physics-informed (and) operator learning.

---

> > ### Comment · Reviewer_33cu · 2024-08-07
> >
> > Of course, there should be a gap between the approximation rate and the experiment results, as there are training errors and sample limitations. However, obtaining the approximation error is still meaningful because it demonstrates the approximation ability of neural networks and provides insights into how to design them. Therefore, I believe the results can be improved. Nonetheless, this paper is a commendable first step in this project, and I maintain my score.

---

> > > ### Author Response · Authors · 2024-08-12
> > > **Reply**
> > >
> > > >  However, obtaining the (PINN) approximation error is still meaningful because it demonstrates the approximation ability of neural networks and provides insights into how to design them.
> > >
> > > We fully agree with this point; however, it would require a separate paper, as our focus is on the cylindrical approximation error rather than the PINN approximation error.
> > >
> > > Thank you for your time and support!

---

### Official Review · Reviewer_LXGL · 2024-07-12

**Soundness:** 4
**Presentation:** 4
**Contribution:** 4
**Rating:** 8
**Confidence:** 4

**Summary:**

The power of PINNs is leveraged to solved high-dimensional PDEs which are obtained from FDEs through the cylindrical approximation. FDEs are computationally expensive to learn, and PDEs are more well-studied in the context of learning. This is a novel work in making FDEs more accessible for computation since they have a wide impact across many technical disciplines.

**Strengths:**

1. This is a novel way to circumvent longstanging numerical issues concerning computing FDEs. The idea is easy to follow, at a high level.
2. Computational complexity and expressive power are significantly improved compared to the current SOTA using cylindrical approximations that are shown to converge.
3. Notation and steps are clearly presented.

**Weaknesses:**

The authors presented their weaknesses/limitations in a section.
They could try to expand their suites of experiments to the applications outlined in the appendix e.g. Navier-Stokes modeling, etc.

**Questions:**

I don't have questions

**Limitations:**

The limitations are clearly presented and have been addressed, when possible, reasonably.

---

> ### Author Rebuttal · Authors · 2024-07-31
>
> We greatly appreciate Reviewer LXGL for their time and recognition of our paper's strength.
>
> > The authors presented their weaknesses/limitations in a section. They could try to expand their suites of experiments to the applications outlined in the appendix e.g. Navier-Stokes modeling, etc.
>
> This is indeed our next project, and we are currently conducting the relevant analyses and experiments. We recognize that this work will require a separate paper, likely better suited for physics journals, because the numerical analysis of higher-order FDEs ($r \geq 2$) is particularly important in physics (see Sec. 1 and App. B).
>
> Please let us know if you have any further questions.

---

### Official Review · Reviewer_5Xf5 · 2024-07-13

**Soundness:** 3
**Presentation:** 2
**Contribution:** 1
**Rating:** 4
**Confidence:** 3

**Summary:**

In this paper, they used cylindrical approximation to transform the functional differential equation (FDE) into a higher-dimensional PDE in order to solve it using PINN.

**Strengths:**

This is a relatively new and intriguing topic in the field of functional differential equations (FDEs), aiming to solve them using PINN as an innovative approach. To this end, specific examples of FDEs such as FTE and BHE were provided, accompanied by a clear mathematical explanation of what cylindrical approximation entails.

**Weaknesses:**

It seems that significant improvements are needed in the experimental section. Particularly, Figure 4 and Figure 7 suffer from poor readability, with unclear x-axis and y-axis labels that make it difficult to discern their intended purpose. Moreover, to demonstrate the advantages of the proposed method effectively, it is essential to compare it with existing numerical analysis methods such as CP-ALS. Such comparisons would clarify how the proposed method reduces computational complexity, as mentioned on line 224 of the manuscript, and illustrate how the order varies with respect to $m$ ($m^6$ vs $m^r$) using the experimental results. Therefore, the experimental section appears to be lacking overall.

Furthermore, cylindrical approximation has already been studied in the field of numerical analysis. Clarifying what novelty exists in applying this approach to convert FDEs into high-dimensional PDEs and then using the well-established PINN method would be beneficial. If the author claims that using deep learning to solve FDEs is novel, it is crucial to provide clear comparisons with state-of-the-art numerical methods in terms of experimental results, computation time, computational cost, accuracy, and other relevant metrics. This comparison would ultimately demonstrate how the proposed approach stands out.

**Questions:**

* I'm curious about the range of \(\boldsymbol{a}\) in the experiments. To train with PINN, \(\boldsymbol{a}\)'s range needs to be fixed beforehand for sampling. How was this managed?

* I'm interested in why r is typically "typically 1 or 2," as mentioned in line 229 of the manuscript. A detailed explanation would be helpful.

* The abbreviation BHE in the caption of Figure 2 first appears in line 162. It should be mentioned earlier in the text.

* Aside from FTE and BHE, I'm curious if there are more complex or challenging applications related to actual physical phenomena.

* In Table 1, it seems that the relative error increases as the degree increases. Is this trend correct? I would appreciate a more detailed explanation and analysis.

* Figures 11-16 are labeled as extrapolation. Could you clarify what this means?

**Limitations:**

See above.

---

> ### Author Rebuttal · Authors · 2024-08-01
>
> We greatly appreciate Reviewer 5Xf5 for their time and invaluable comments.
> We will incorporate all the suggestions into our manuscript.
>
> > I'm curious about the range of $\boldsymbol{a}$ in the experiments.
>
> It is provided in lines 1456-1458.
>
> > Aside from FTE and BHE, I'm curious if there are more complex or challenging applications related to actual physical phenomena.
>
> App. B provides an additional introduction to FDEs. For example, the Navier-Stokes-Hopf equation (Eq. (10) in line 805) comprehensively describes the statistical properties of turbulence of realistic 3D fluids, involving complicated differentiation and functional differentiation operations. The Schwinger-Dyson equation (Eq. (13) in line 833) solely provides properties of quantum particles, while solving it is even more challenging because it can involve, for example, third-order functional derivatives.
>
> > I'm interested in why r is "typically 1 or 2," as mentioned in line 229 of the manuscript. A detailed explanation would be helpful.
>
> Major examples of FDEs include the Hopf functional equation, the Fokker-Planck functional equation, and the functional Hamilton-Jacobi equation (lines 22-30 & Sec. 2).
> All of them have $r=1$ or $2$.
> Other less-known FDEs, such as Eq. (16-17) in line 859, also have $r=2$.
>
> A notable example with $r=3$ is the Schwinger-Dyson equation (Eq. (13)).
> To our knowledge, there are no other major significant examples with $r > 2$, although arbitrary toy FDEs can be constructed.
>
> > In Table 1, it seems that the relative error increases as the degree increases. Is this trend correct? I would appreciate a more detailed explanation and analysis.
>
> As noted in the captions of Tables 1 & 2, these tables are not intended to assess the theoretical convergence of the cylindrical approximation; therefore, a decrease in relative error is not necessarily expected. This is because different exact solutions (ground truth functionals) are used for different rows, as is stated in footnote 1 on page 7.
>
> - Firstly, these tables serve as a proof of concept for our proposed approach, demonstrating its capability to learn FDEs effectively.
> - Secondly, the expected decrease of relative error for large $m$ *without* training PINNs is illustrated in Figure 2.
> - Thirdly, the expected decrease of relative error for large $m$ *with* training PINNs is analyzed in App. H.6, where a cross-degree evaluation is performed.
>
> > Figures 11-16 are labeled as extrapolation. Could you clarify what this means?
>
> - Firstly, figures 11-16 shows that no collocation points such that $\| \boldsymbol{a} \| \approx 0$ were included in the training sets.
> - Secondly, Figures 4 & 7 plot the absolute errors of PINN predictions at the collocation points where $\boldsymbol{a} = (0, 0, \dots, 0, a_k, 0, \dots, 0)$ with $k = 0, 1, 2, 19,$ or $99$.
> - Therefore, the collocation points such that $a_k \approx 0$ in Figures 4 & 7 were not included in the training sets, but the errors were as small as those in the region where $a_k \not\approx 0$. We refer to this as "extrapolation."
> - Additionally, we have revised confusing sentences in lines 296-300 and 312-316 for clarity.
>
> > cylindrical approximation has already been studied ... Clarifying what novelty exists in applying this approach ... and then using the well-established PINN method would be beneficial.
>
> - Theoretical Contribution Compared with Previous Studies on the Cylindrical Approximation:
>     - For details, please see lines 97-101 and, for technical details, lines 1016-1019 in App. C.4.2.
>     - In summary, we established convergence theorems of functionals and FDE solutions under a modified cylindrical approximation tailored for practical use (i.e., the cylindrical approximation of functionals without the "tail term", as mentioned in lines 125-127).
>
> - Empirical Contribution, Especially with the Use of PINNs:
>     - In functional analysis, there are several approaches to reduce infinite dimensional function spaces to finite spaces to cut computational costs. Researchers have sought efficient and scalable methods.
>     - Among them, we found that the cylindrical approximation, combined with PINNs, can offer significant scalability, which may seem obvious in hindsight.
>     - Previous approaches using the cylindrical approximation have solely focused on tensor decomposition and finite difference methods (lines 90-92), which limit scalability. Thus, their experiments were confined to the input functions such as polynomials with degrees no higher than $10$. In contrast, our model can treat degrees up to $m \sim 1000$, demonstrating unprecedented expressivity.
>
> > it is essential to compare it with existing numerical analysis methods such as CP-ALS.
>
> > it is crucial to provide clear comparisons with state-of-the-art numerical methods.
>
> For the runtime comparison with the CP-ALS, please see lines 1523-1526 in App. G (Detailed Experimental Settings: Runtime).
>
> For the empirical comparison of computational complexities, note that previous approaches suffer from the severe curse of dimensionality, making it *impossible* to perform experiments with as large as $m \sim 50$ or higher. Besides, the implementations of their complex algorithms are unavailable, further making complicating performance comparisons.
>
> For the error comparison, see lines 337-338 and App. H.4, where we compare the CP-ALS with our model in solving the advection-reaction equation. In summary, the error of the CP-ALS is $\lesssim 10^{-2}$, while ours is $\sim 10^{-1}$. Please note that the following:
> - 1. We further tuned hyperparameters after submission, resulting in even smaller errors for our model. Please see also lines 1327-1332 in App. F.1 ("To further reduce errors").
> - 2. Table 12 contains typos: RELATIVE ERROR, BEST RELATIVE ERROR, and WORST RELATIVE ERROR should be read as WORST RELATIVE ERROR, RELATIVE ERROR, and BEST RELATIVE ERROR, respectively.

---

> > ### Comment · Reviewer_5Xf5 · 2024-08-12
> >
> > Thank you for your response. However, as you mentioned, it seems that many of the critical points I considered important are mostly found in the appendix. I believe these key sections should be brought into the main text and revised in future versions. For example, the explanation for point (a), and the purpose behind figures 4 and 7, which are difficult to understand, should be revised so that their meaning and what they intend to show are clearly visible. Additionally, I still have the following questions:
> >
> > - In the comparison with CP-ALS for the advection-reaction equation, does the proposed method have a higher error? If so, should this be applied to higher-dimensional problems (where m is higher than 50)? In this case, is CP-ALS entirely impossible to perform, or does it just take a long time, or is the error too large? If it’s just a matter of time, then shouldn't the error after running it for a sufficient amount of time be compared in Tables 1 and 2? Or is the table meant to show that the proposed method remains applicable even as the degree increases?
> >
> > - Although I now understand most of my concerns through your responses, many of these points are found in the appendix or are missing from the main text. I believe adding more comparisons with existing methods in the experiments would enhance the novelty of this paper.
> >
> > Therefore, while I think the current version of the paper still has some shortcomings that need to be addressed before it can be accepted, I have resolved several of my concerns through your responses, so I will increase my score by 1 point, bringing it to 4. Thank you once again for addressing my questions.

---

> > > ### Author Response · Authors · 2024-08-13
> > >
> > > Thank you very much for your support.
> > > We are pleased to address your questions and concerns.
> > >
> > >
> > >
> > >
> > >
> > > >  I believe these key sections should be brought into the main text and revised in future versions. For example, the explanation for point (a),
> > >
> > > Thank you for your suggestion. We have now included the range of $\boldsymbol{a}$ in the main text.
> > >
> > > > the purpose behind figures 4 and 7, which are difficult to understand, should be revised so that their meaning and what they intend to show are clearly visible.
> > >
> > > - We will replace figures to improve visibility, adjusting the size of characters and spacing.
> > >
> > > - Our intention: Figures 4 and 7 primarily demonstrate that our proposed approach effectively learns FDEs (FTE and BHE), serving as *proof of concept* (line 262). They are not intended to showcase extrapolation capability, which is a secondary message. We apologize for any confusion. We will clarify this by adding explanations around lines 293 and 320. Thank you for your invaluable feedback, which has enhanced the presentation quality of our paper.
> > >
> > >
> > >
> > > > I believe adding more comparisons with existing methods in the experiments would enhance the novelty of this paper.
> > >
> > > Thank you for the suggestion.
> > > Please note that CP-ALS and hierarchical Tucker (HT) algorithms are the only methods based on the cylindrical approximation for numerically solving FDEs, and HT is slower than CP-ALS (line 692); thus, we chose CP-ALS for comparison.
> > > Additionally, the number of papers focusing on general-purpose numerical FDE solvers is very limited [79, 91], although several papers address numerical functional approximation (not for solving FDEs).
> > >
> > > > In the comparison with CP-ALS for the advection-reaction equation, does the proposed method have a higher error? If so, should this be applied to higher-dimensional problems (where m is higher than 50)? In this case, is CP-ALS entirely impossible to perform, or does it just take a long time, or is the error too large? If it’s just a matter of time, then shouldn't the error after running it for a sufficient amount of time be compared in Tables 1 and 2?
> > >
> > > In the comparison with CP-ALS for the advection-reaction equation, our proposed method indeed has a higher error.
> > > However, performing experiments with CP-ALS (or similar finite difference or element method-based algorithms) in high dimensions is infeasible due to prohibitively long runtimes.
> > >
> > > Specifically, let us consider a state-of-the-art fast PDE solver used in [79] (published in 2024), which reports runtime of approximately 0.75 hours per time step on an Intel i9-7980XE workstation for a 20-dimensional problem.
> > > If CP-ALS had a similar runtime, it would take approximately $0.75 \times \frac{100^6}{20^6} = 0.75 \times 15625 \approx 11719$ hours (over a year) for only a single time step, due to its $\mathcal{O}(m^6)$ complexity.
> > > In contrast, our model can learn FDEs within a few hours, even at dimensions as large as $1000$.
> > >
> > >
> > >
> > >
> > > In addition to this response, we would appreciate it if the reviewer could reevaluate both our
> > >
> > > > Theoretical Contribution
> > >
> > > > Empirical Contribution
> > >
> > > summarized in our previous response or share any additional thoughts or reasons that are contributing to this (borderline) reject assessment. Understanding this would help us in further improving the work.
> > >
> > > Sincerely,
> > >
> > > Authors

---

> > > > ### Comment · Reviewer_5Xf5 · 2024-08-14
> > > >
> > > > Thank you for thoroughly answering my questions. All of my concerns have been resolved. However, in my personal opinion, the current state of the paper still seems insufficient for acceptance, mainly because of the lack of explanation in the main text for the points clarified through my questions and the fact that many important details are in the appendix rather than in the main body of the paper. Therefore, I will maintain my current score. Nonetheless, I appreciate your efforts in addressing my questions and concerns.

---

### Decision · Program_Chairs · 2024-09-25

**Decision:**

Accept (poster)

**Comment:**

The paper considers neural network based approach for solving functional differential equations, and establish a convergence result based on cylindrical approximation. The method is a natural extension of neural network based PDE solvers. Overall the reviewers find the approach interesting and most concerns are addressed during the discussion period, even though more empirical tests are desired. After carefully reading the manuscript and discussion, the meta reviewer would recommend acceptance of the manuscript.